# Towards Optimal Regret in Adversarial Linear MDPs with Bandit Feedback

**Haolin Liu**[1*], **Chen-Yu Wei**[1*], **Julian Zimmert**[2*]

[1] Department of Computer Science, University of Viginia, [2] Google Research
{srs8rh, kfw6en}@virginia.edu
{zimmert}@google.com

## ABSTRACT

We study online reinforcement learning in linear Markov decision processes with adversarial losses and bandit feedback, without prior knowledge on transitions or access to simulators. We introduce two algorithms that achieve improved regret performance compared to existing approaches. The first algorithm, although computationally inefficient, ensures a regret of $\widetilde{\mathcal{O}}(\sqrt{K})$, where $K$ is the number of episodes. This is the first result with the optimal $K$ dependence in the considered setting. The second algorithm, which is based on the policy optimization framework, guarantees a regret of $\widetilde{\mathcal{O}}(K^{3/4})$ and is computationally efficient. Both our results significantly improve over the state-of-the-art: a computationally inefficient algorithm by Kong et al. (2023) with $\widetilde{\mathcal{O}}(K^{4/5} + \text{poly}(1/\lambda_{\min}))$ regret, for some problem-dependent constant $\lambda_{\min}$ that can be arbitrarily close to zero, and a computationally efficient algorithm by Sherman et al. (2023b) with $\widetilde{\mathcal{O}}(K^{6/7})$ regret.

## 1 INTRODUCTION

We study finite-horizon online reinforcement learning in a large state space with adversarial losses amd bandit feedback. We assume the linear Markov decision process (MDP) structure: every state-action pair is equiped with a known feature representation, and both the transitions and the losses can be represented as a linear function of the feature. This problem has received significant attention recently, with fairly complete results when the agent has access to a simulator to query transitions of the MDP (Dai et al., 2023). In the much harder simulator-free setting, the pioneering work of Luo et al. (2021) showed that no-regret ($K^{14/15}$ regret) is possible, where $K$ is the number of episodes. Several followup works have successively improved the $K$ dependence (Dai et al., 2023; Sherman et al., 2023b; Kong et al., 2023), with the state-of-the-art being Kong et al. (2023)'s $K^{4/5} + \text{poly}(1/\lambda_{\min})$ regret through a computationally inefficient algorithm, and Sherman et al. (2023b)'s $K^{6/7}$ regret through a computationally efficient algorithm. Still, there remain significant gaps between the current upper bounds and the $\sqrt{K}$ lower bound. In this work, we push the frontiers both on the information theoretical limits and the achievable bounds under computational constraints: 1) we present the first (computationally inefficient) algorithm that provably obtains $\widetilde{\mathcal{O}}(\sqrt{K})$ regret, showing that this is the minimax $K$ dependence (Section 3); 2) we obtain $\widetilde{\mathcal{O}}(K^{3/4})$ regret with a polynomial-time algorithm (Section 4). Below, we briefly describe the elements in our approaches. The comparison of our algorithms with previous works can be found in Appendix A.

**Inefficient $\sqrt{K}$ algorithm.** We convert the linear MDP problem to a linear bandit problem by mapping each policy to a single $dH$-dimensional feature vector, where $d$ is the ambient dimension of the linear MDP and $H$ is the horizon length. The challenge is that this conversion depends on the transition of the MDP, which is not available to the learner. Therefore, the learner has to estimate the feature of every policy during the learning process. Previous work in this direction (Kong et al., 2023) faced obstacles in controlling the estimation error and was only able to show a $K^{4/5} + \text{poly}(1/\lambda_{\min})$ regret bound assuming there exists an exploratory policy inducing a covariance

---

*The authors are listed in alphabetical order.

matrix $\succeq \lambda_{\min} I$. We addressed the obstacles through 1) state space discretization (Section 3.2), and 2) model-free estimation for the occupancy measure of policies over the discretized state space (Section 3.3). These allow us to emulate the success in the tabular case (Jin et al., 2020a) and obtain the tight $\sqrt{K}$ regret.

**Efficient $K^{3/4}$ algorithm.** The efficient algorithm is based on the policy optimization framework (Luo et al., 2021). Different from previous works that all use exponential weights, we use Follow-the-Regularized-Leader (FTRL) with log-determinant (logdet) barrier regularizer to perform policy updates, which has the benefit of keeping the algorithm more stable (Zimmert and Lattimore, 2022; Liu et al., 2023a). We carefully combine logdet-FTRL with existing algorithmic/analysis techniques to further improve the regret bound. These include 1) an initial exploration phase to control the transition estimation error (Sherman et al., 2023a), 2) optimistic least-square policy evaluation in bonus construction (Sherman et al., 2023b), 3) dilated bonus construction (Luo et al., 2021), and 4) a tighter concentration bound for covariance matrix estimation (Liu et al., 2023a).

We defer detailed comparisons with the literature to Appendix A.

## 2 PRELIMINARIES

**No-Regret Learning in MDPs.** An (episodic) MDP is specified by a tuple $\mathcal{M} = (\mathcal{S}, \mathcal{A}, P)$ where $\mathcal{S}$ is the state space (possibly infinite), $\mathcal{A}$ is the action space (assumed to be finite with size $A = |\mathcal{A}|$), $P \colon \mathcal{S} \times \mathcal{A} \to \Delta(\mathcal{S})$ is the transition kernal. The state space is assumed to be *layered*, i.e., $\mathcal{S} = \mathcal{S}_1 \cup \mathcal{S}_2 \cup \cdots \cup \mathcal{S}_H$ where $\mathcal{S}_h \cap \mathcal{S}_{h'} = \varnothing$ for any $1 \le h < h' \le H$, and transition is only possible from one layer to the next, that is, $P(s' \mid s, a) \ne 0$ only when $s \in \mathcal{S}_h$ and $s' \in \mathcal{S}_{h+1}$. Without loss of generality, we assume $\mathcal{S}_1 = \{s_1\}$.

We consider a process where the learner interact with the MDP for $K$ episodes, each time with a different loss function. Before the game starts, an adversary arbitrarily chooses the loss functions for all episodes $(\ell_k : \mathcal{S} \times \mathcal{A} \to [0, 1])_{k=1}^K$, and does not reveal them to the learner. For each episode $k \in [K]$, the learner starts at state $s_{k,1} = s_1$; for each step $h \in [H]$ within episode $k$, after observing the state $s_{k,h} \in \mathcal{S}_h$, the learner chooses an action $a \in \mathcal{A}$, suffers and observes the loss $\ell_k(s_{k,h}, a_{k,h})$, and transits to a new state $s_{k,h+1}$ sampled from the transition $P(\cdot \mid s_{k,h}, a_{k,h})$.

A policy $\pi$ is a mapping from $\mathcal{S}$ to $\Delta(\mathcal{A})$. The *state-value function* (or V-function in short) $V^\pi(s; \ell)$ is the cumulative loss starting from state $s$, following policy $\pi$ and under loss function $\ell$. This is formally defined as the following for $s \in \mathcal{S}_h$:

$$V^\pi(s; \ell) \triangleq \mathbb{E}\left[\sum_{h'=h}^H \ell(s_{h'}, a_{h'}) \,\middle|\, s_h = s, \ a_{h'} \sim \pi(\cdot \mid s_{h'}), \ s_{h'+1} \sim P(\cdot \mid s_{h'}, a_{h'}), \ \forall h' \ge h\right].$$

The *action-value function* (*a.k.a.* Q-function), on the other hand, is the expected loss suffered by a policy $\pi$ starting from a given state-action pair $(s, a)$. Formally, we define for all $(s, a) \in \mathcal{S} \times \mathcal{A}$:

$$Q^\pi(s, a; \ell) = \ell(s, a) + \mathbb{I}[s \notin \mathcal{S}_H] \cdot \mathbb{E}_{s' \sim P(\cdot \mid s, a)}\left[V^\pi(s'; \ell)\right]. \tag{1}$$

Let $\pi_k$ be the policy used by the learner in episode $k$. The learner aims to minimize the *regret* with respect to the best fixed policy, defined as

**Definition 1** (Regret). $\mathcal{R}_K \triangleq \mathbb{E}\left[\sum_{k=1}^K V^{\pi_k}(s_1; \ell_k)\right] - \min_\pi \sum_{k=1}^K V^\pi(s_1; \ell_k)$.

**Occupancy measures.** For a policy $\pi$ and a state $s$, we define $\mu^\pi(s)$ to be the probability of visiting state $s$ within an episode when following $\pi$, which can be written as $\mu^\pi(s) = V^\pi(s_1; \delta_s)$ with $\delta_s(s', a') = \mathbb{I}\{s' = s\}$. Further define $\mu^\pi(s, a) = \mu^\pi(s)\pi(a|s)$. By definition, we have $V^\pi(s_1; \ell) = \sum_{s \in \mathcal{S}} \sum_{a \in \mathcal{A}} \mu^\pi(s, a)\ell(s, a)$.[1]

### 2.1 LINEAR MDP

Linear MDP is formally defined as follows.

---

[1] For readability, throughout the paper, we use summation over states instead of integration. Technically, all our results hold for case of continuous and infinite state space.

**Definition 2** (Linear MDP). *In a linear MDP, each state-action pair $(s, a)$ is associated with a known feature $\phi(s, a) \in \mathbb{R}^d$ with $\|\phi(s, a)\|_2 \leq 1$. There exists a mapping $\psi \colon \mathcal{S} \to \mathbb{R}^d$ such that the transition can be expressed as*

$$P(s' \mid s, a) = \langle \phi(s, a), \psi(s') \rangle, \quad \forall (s, a, s') \in \bigcup_{h=1}^{H-1} \mathcal{S}_h \times \mathcal{A} \times \mathcal{S}_{h+1}. \tag{2}$$

*Here, $\psi$ is unrevealed to the learner. Moreover, for any episode $k \in [K]$ and any layer $h \in [H]$, there exists a (hidden) vector $\theta_{k,h} \in \mathbb{R}^d$ such that*

$$\ell_k(s, a) = \langle \phi(s, a), \theta_{k,h} \rangle, \quad \forall (s, a) \in \mathcal{S}_h \times \mathcal{A}. \tag{3}$$

*Following previous work, we assume $\| \sum_{s \in \mathcal{S}_h} |\psi(s)| \|_2 \leq \sqrt{d}$ (the absolute value $|\cdot|$ over a vector is element-wise) and $\|\theta_{k,h}\|_2 \leq \sqrt{d}$ for all $k, h, \pi$.*

We also define misspecifeid linear MDPs, which is used in [Section 3](#).

**Definition 3** (Misspecified Linear MDP). *A $\zeta$-misspecified linear MDP follows all the assumptions in [Definition 2](#) except that [Eq. (2)](#) and [Eq. (3)](#) are respectively modified to*

$$\|P(\cdot \mid s, a) - \langle \phi(s, a), \psi(\cdot) \rangle\|_1 \leq \zeta \quad \text{and} \quad |\ell_k(s, a) - \langle \phi(s, a), \theta_{k,h} \rangle| \leq \zeta. \tag{4}$$

## 3 RATE-OPTIMAL ALGORITHM

The aim of this section is to show that there is no statistical barrier to obtaining $\sqrt{K}$ regret for linear MDPs with bandit feedback and adversarial losses. The proposed algorithm is computationally inefficient and it remains an open question if the same can be achieved with an efficient algorithm.

### 3.1 SOLUTION IDEAS

Observe that the expected loss of policy $\pi$ in episode $k$ can be written as $\sum_{s \in \mathcal{S}} \sum_{a \in \mathcal{A}} \mu^\pi(s, a) \ell_k(s, a) = \sum_{h=1}^H \sum_{s \in \mathcal{S}_h} \sum_{a \in \mathcal{A}} \mu^\pi(s, a) \phi(s, a)^\top \theta_{k,h}$. This can be further written as $\langle \phi^\pi, \theta_k \rangle$, where

$$\phi^\pi = (\phi_1^\pi, \ldots, \phi_H^\pi), \quad \theta_k = (\theta_{k,1}, \ldots, \theta_{k,H}), \qquad \text{with } \phi_h^\pi = \sum_{s \in \mathcal{S}_h} \sum_{a \in \mathcal{A}} \mu^\pi(s, a) \phi(s, a).$$

In other words, the adversarial linear MDP problem can be viewed as an adversarial linear bandit problem with $(\phi^\pi)_{\pi \in \Pi}$ as the underlying action set. Therefore, if computation is not an issue (i.e., if we are allowed to run linear bandits over an exponentially large action set), the only additional challenge in linear MDPs is that $(\phi^\pi)_{\pi \in \Pi}$ is not known in advance and the learner must learn the transition to estimate them. This viewpoint has been taken by Kong et al. (2023) to design computationally inefficient algorithms with improved regret bounds. To estimate $(\phi^\pi)_{\pi \in \Pi}$, Kong et al. (2023) use an initial pure exploration phase to estimate $\phi^\pi$ up to an accuracy of $\epsilon$ for all $\pi$, and then run a $\epsilon$-misspecified linear bandit algorithm over policies in the second phase. Their approach gives $K^{4/5} + \text{poly}(1/\lambda_{\min})$ regret.

A natural idea to improve the regret bound is to estimate $(\phi^\pi)_{\pi \in \Pi}$ *on the fly* instead of in a separate initial phase. That is, we directly start a linear bandit algorithm. Then during the learning process, for policies that are more often used by the learner, their $\phi^\pi$ estimation will become more and more accurate, and for others, larger error is allowed. Intuitively, this better balances exploitation and exploration because the learner will not spend too much efforts in estimating $\phi^\pi$ for bad policies. However, there are technical difficulties in doing so. Recall that $\phi_h^\pi = \sum_{s \in \mathcal{S}_h} \sum_{a \in \mathcal{A}} \mu^\pi(s) \pi(a|s) \phi(s, a)$. To estimate this, the learner needs to first estimate $\mu^\pi$. A natural estimator $\hat{\mu}^\pi$ would be defined recursively as $\hat{\mu}^\pi(s') = \sum_{s \in \mathcal{S}_h} \sum_{a \in \mathcal{A}} \hat{\mu}^\pi(s) \pi(a|s) \hat{P}(s'|s, a)$ for $s' \in \mathcal{S}_{h+1}$, with the transition estimator $\hat{P}$ obtained from linear regression: $\hat{P}(s'|s, a) = \phi(s, a)^\top \left( \Lambda_h^{-1} \sum_{(\tilde{s}, \tilde{a}, \tilde{s}') \in \mathcal{D}_h} \phi(\tilde{s}, \tilde{a}) \mathbb{I}\{\tilde{s}' = s'\} \right)$ where $\mathcal{D}_h$ consists of historical data of the form $(s, a, s') \in \mathcal{S}_h \times \mathcal{A} \times \mathcal{S}_{h+1}$ and $\Lambda_h = I + \sum_{(s, a, s') \in \mathcal{D}_h} \phi(s, a) \phi(s, a)^\top$. This is the exact idea

of Kong et al. (2023). Notice that the $\hat{\mu}^\pi$ obtained in this way may not be *valid*, i.e., they may not satisfy $\hat{\mu}^\pi(\cdot) \in \Delta(\mathcal{S})$. Their approach suffers from the issue that it is difficult to control the magnitude of $\hat{\mu}^\pi(s)$ when the amount of data in $\mathcal{D}_h$ is still small. This is why they use an initial phase to explore all directions in the feature space and control the error $\|\hat{\phi}_h^\pi - \phi_h^\pi\|$ uniformly for all policies.

However, "on-the-fly estimation" without the initial phase has been proven to work in the tabular case (Jin et al., 2020a) to get a $\sqrt{K}$ regret. The key difference between the tabular case and the linear case is that the transition estimator $\hat{P}$ in the tabular case is always a valid transition (i.e., $\hat{P}(\cdot|s,a) \in \Delta(\mathcal{S})$), and thus the induced occupancy measure estimator $\hat{\mu}^\pi$ is also always valid. This avoids the aforementioned technical difficulty.

With this observation, we propose to incorporate the constraint that $\hat{\mu}^\pi$ be a valid occupancy measure when dealing with linear MDPs. To find such a $\hat{\mu}^\pi$, we search over the space of valid occupancy measures and pick one that is consistent with the past data. This is different from the approach of Kong et al. (2023), where $\hat{P}$ is obtained via linear regression over the past data first, and then $\hat{\mu}^\pi$ is derived from it, which can fail to be valid.

Since the state space and policy space can both be infinite, in order to get a runnable algorithm for finding $\hat{\mu}^\pi(s)$, we discretize both the state space and the policy space. These are described in the next subsection.

## 3.2  THE DISCRETIZATION PROCEDURES

**Discretization of the state space.**     For linear MDPs, we can assume that a state $s$ is uniquely defined by its action feature set $\mathcal{A}_s = \{\phi(s,a)\,|\,a \in \mathcal{A}\}$. If there are distinct states with identical feature sets, we can collapse them into a single state by combining their $\psi(s)$.

In order to approximate an infinite-state linear MDP as a finite-state MDP, we perform discretization for the entire feature space $\mathbb{B}^d(1)$. To decide the discretization resolution, assume that $\phi(s,a)$ is the true feature and $\phi'(s,a)$ is its approximation, and $\|\phi(s,a) - \phi'(s,a)\|_2 \leq \epsilon$ for all $s,a$. Then we have $\|P(\cdot|s,a) - \langle\phi'(s,a),\psi(\cdot)\rangle\|_1 = \|\langle\phi(s,a) - \phi'(s,a),\psi(\cdot)\rangle\|_1 \leq \sum_{s'}\|\phi(s,a) - \phi'(s,a)\|_2\|\psi(s')\|_2 \leq \epsilon\sum_{s'}\|\psi(s')\|_2 \leq \epsilon\sum_{i=1}^d\sum_{s'}|\psi_i(s')| \leq \epsilon\sqrt{d}\|\sum_{s'}|\psi(s')|\|_2 \leq \epsilon d$ and $|\ell_k(s,a) - \langle\phi'(s,a),\theta_{k,h}\rangle| = |\langle\phi'(s,a) - \phi(s,a),\theta_{k,h}\rangle| \leq \|\phi'(s,a) - \phi(s,a)\|_2\|\theta_{k,h}\|_2 \leq \epsilon\sqrt{d}$ by Definition 2. Thus, the MDP with $\phi'(s,a)$ as the underlying feature is a misspecified linear MDP with misspecification error $\zeta = \epsilon d$ by Definition 3. It turns out that it suffices to set $\epsilon = \frac{1}{K}$ and make the misspecification error $\zeta = \frac{d}{K}$. The number of states after the discretization is upper bounded by (size of $\epsilon$-net of the feature space)$^A = (1/\epsilon)^{\mathcal{O}(dHA)} = K^{\mathcal{O}(dHA)}$.

There is a caveat when working with this discretized state space. Since the true feature space $\Phi = \{\phi(s,a): s \in \mathcal{S}, a \in \mathcal{A}\}$ may not cover the entire $\mathbb{B}^d(1)$, the state space construction above (i.e., by discretizing the whole $\mathbb{B}^d(1)$) may produce states that do not really exist. In fact, there is no problem viewing these non-existing states as part of the state space because their $\psi(s)$ can be set to zero, making them unreachable under the linear MDP assumption. The only thing we have to be careful about is that the assumptions Eq. (2), Eq. (3), Eq. (4), and their implications, such as $-\zeta \leq \langle\phi,\psi(s')\rangle \leq 1+\zeta$ and $|\langle\phi,\theta_{k,h}\rangle| \leq 1+\zeta$, are only guaranteed for $\phi$ in the *true feature space* $\Phi$, but not for the whole feature space $\mathbb{B}^d(1)$. To avoid ambiguity, we use notation $\mathcal{S}$ to denote the set of discretized states from the *true* MDP, and use $\mathcal{X}$ to denote the set of discretized states constructed from the entire $\mathbb{B}^d(1)$. Apparently, $\mathcal{S} \subseteq \mathcal{X}$. We clarify that, 1) the learner knows $\mathcal{X}$, but does not know $\mathcal{S}$ before interacting with the environment, 2) the misspecified linear MDP assumption Eq. (4) is only guaranteed for $\phi(s,a)$ with $s \in \mathcal{S}$, 3) $\mathcal{X} \setminus \mathcal{S}$ are unreachable states and their $\psi(s)$ are set to zero. We use $(\mathcal{X}_h)_{h \in [H]}$ to denote partitions of $\mathcal{X}$ on different layers.

**Discretization of the policy space.**     We consider a discretization of the policy space for Algorithm 2. The policy class is the set of linear policies defined as

$$\Pi = \left\{\pi_\theta: \theta \in \Theta^H, \ \pi_\theta(s) = \operatorname*{argmin}_{a \in \mathcal{A}} \phi(s,a)^\top\theta_h \text{ for } s \in \mathcal{X}_h\right\} \tag{5}$$

where $\Theta$ is an 1-net of $\mathbb{B}^d(K)$. The next lemma shows that this policy set contains a near optimal one. See Appendix B.1 for the proof.

**Algorithm 1** EstOM$(\pi, (\mathcal{D}_h)_{h=1}^H)$ (**Est**imate **O**ccupancy **M**easure)

---

**Input**: target policy $\pi$, historical data $(\mathcal{D}_h)_{h=1}^H$ where $\mathcal{D}_h$ consists of tuples $(s, a, s') \in \mathcal{S}_h \times \mathcal{A} \times \mathcal{S}_{h+1}$ with $s' \sim P(\cdot|s, a)$.

Find $(\hat{\mu}^\pi(s))_{s \in \mathcal{X}} \subset [0, 1]$ and $(\hat{\xi}_{h,f})_{h \in [H], f \in \mathcal{F}^\pi} \subset \mathbb{B}^d(\sqrt{d})$ that satisfy the following for all $h \in [H]$ and all $f \in \mathcal{F}^\pi$ (recall the definition of $\mathcal{F}^\pi$ in Eq. (6), and $\zeta$ in Section 3.2).

$$\sum_{s \in \mathcal{X}_h} \hat{\mu}^\pi(s) = 1, \tag{7}$$

$$\left| \sum_{s' \in \mathcal{X}_{h+1}} \hat{\mu}^\pi(s') f(s') - \sum_{s \in \mathcal{X}_h} \sum_{a \in \mathcal{A}} \hat{\mu}^\pi(s) \pi(a|s) \operatorname{clip}\left[\phi(s, a)^\top \hat{\xi}_{h,f}\right] \right| \leq \zeta \tag{8}$$

$$\sum_{(s,a,s') \in \mathcal{D}_h} \left(f(s') - \phi(s, a)^\top \hat{\xi}_{h,f}\right)^2 - \min_{\xi \in \mathbb{B}^d(\sqrt{d})} \sum_{(s,a,s') \in \mathcal{D}_h} \left(f(s') - \phi(s, a)^\top \xi\right)^2 \leq 16 d^{\frac{5}{2}} \log \frac{18 d^{\frac{3}{2}} K}{\delta} \tag{9}$$

**Output**: $(\hat{\mu}^\pi(s))_{s \in \mathcal{X}}$ (if Eq. (7)-Eq. (9) is not feasible, output any solution that satisfies Eq. (7)).

---

**Lemma 4.** *For any policy $\pi : \mathcal{X} \to \Delta(\mathcal{A})$ and any sequence of losses $(\theta_{k,h})_{h \in [H], k \in [K]}$, there exists a policy $\pi' \in \Pi$ such that $\sum_{k=1}^K \sum_{h=1}^H \sum_{s \in \mathcal{S}_h} \sum_{a \in \mathcal{A}} (\mu^{\pi'}(s, a) - \mu^\pi(s, a)) \phi(s, a)^\top \theta_{k,h} \leq \sqrt{d} H^2$.*

## 3.3 Estimating $\mu^\pi(s)$

With the state space discretized, we are now faced with a finite state problem. To estimate $\mu^\pi$, a potential way is to find a transition estimation $(\hat{P}(s'|s, a))_{s,a,s'}$ which is consistent with the historical data and satisfies the constraint that the $\hat{\mu}^\pi$ induced by $\hat{P}$ is a valid occupancy measure. The issue of this is that since $P(s'|s, a) \approx \phi(s, a)^\top \psi(s')$, this method requires us to estimate $\psi(s')$ for all $s'$, whose complexity will scale with $|\mathcal{S}|$ because $\psi(s')$ for different $s'$ are unrelated. Indeed, as noted by previous works (Foster et al., 2023), the linear MDP model does not allow efficient model-based estimation.

Inspired by previous model-free approaches for linear MDPs (Jin et al., 2020b), instead of estimating $\psi(s')$, we will directly estimate $\sum_{s'} \psi(s') f(s')$ for a class of functions $f$ that is rich enough for our purpose (i.e., to estimate $(\phi^\pi)_{\pi \in \Pi}$ well). This class of functions turns out can be chosen as $\bigcup_{\pi \in \Pi} \mathcal{F}^\pi$ where $\mathcal{F}^\pi = \mathcal{F}_1^\pi \cup \mathcal{F}_2^\pi$ and

$$\mathcal{F}_1^\pi = \left\{ f : \mathcal{X} \to [-1, 1] \ \middle| \ f(s) = \sum_{a \in \mathcal{A}} \pi(a|s) \operatorname{clip}\left[\phi(s, a)^\top \theta\right] \text{ for some } \theta \in \mathbb{B}^d(\sqrt{d}) \right\},$$

$$\mathcal{F}_2^\pi = \left\{ f : \mathcal{X} \to [-1, 1] \ \middle| \ f(s) = \sum_{a \in \mathcal{A}} \pi(a|s) \|\phi(s, a)\|_\Gamma \text{ for some } \Gamma \text{ with } \mathbf{0} \preceq \Gamma \preceq I \right\}, \tag{6}$$

where we define $\operatorname{clip}[a] = \max(\min(a, 1), -1)$. Given historical data $(\mathcal{D}_h)_{h=1}^H$ which consists of $(s, a, s')$ tuples, our way of obtaining $\hat{\mu}^\pi$ is summarized in Algorithm 1. In Algorithm 1, Eq. (7) sets the constraint that $\hat{\mu}^\pi$ is a valid occupancy measure, Eq. (9) requires that $\hat{\xi}_{h,f}$ approximates $\xi_{h,f}^\star = \sum_{s' \in \mathcal{S}_{h+1}} \psi(s') f(s')$ well on the historical data $(\mathcal{D}_h)_{h=1}^H$, and Eq. (8) relates $\hat{\mu}^\pi$ with $\hat{\xi}_{h,f}$ according to their definitions. In the following Lemma 5, we show that Eq. (7)-Eq. (9) is feasible with high probability. Then in Lemma 6, we show the key property that $\hat{\mu}^\pi$ is close to $\mu^\pi$ when evaluated on any $f \in \mathcal{F}^\pi$. The proofs of Lemma 5 and Lemma 6 can be found in Appendix B.2. Below, we define $\hat{\mu}^\pi(s, a) := \hat{\mu}^\pi(s) \pi(a|s)$.

**Lemma 5.** *With probability at least $1 - \frac{\delta}{K}$, Eq. (7)-Eq. (9) is feasible for all $\pi \in \Pi$.*

**Lemma 6.** *Let $(\hat{\mu}^\pi(s))_{s \in \mathcal{X}}$ be the output of Algorithm 1. Then with probability at least $1 - \frac{\delta}{K}$, for any $\pi \in \Pi$ and all $f \in \mathcal{F}^\pi$, $\left| \sum_{s \in \mathcal{X}_h} (\hat{\mu}^\pi(s) - \mu^\pi(s)) f(s) \right|$ is upper bounded by*

$$10d^{\frac{5}{4}} \sqrt{\log \frac{18d^{\frac{3}{2}}K}{\delta}} \times \sum_{h' < h} \min \left\{ \sum_{s \in \mathcal{X}_{h'}} \sum_{a \in \mathcal{A}} \mu^\pi(s,a) \|\phi(s,a)\|_{\Lambda_{h'}^{-1}}, \sum_{s \in \mathcal{X}_{h'}} \sum_{a \in \mathcal{A}} \hat{\mu}^\pi(s,a) \|\phi(s,a)\|_{\Lambda_{h'}^{-1}} \right\} + 2\zeta H$$

*where $\Lambda_h := I + \sum_{(s,a,s') \in \mathcal{D}_h} \phi(s,a)\phi(s,a)^\top$.*

### 3.4 Algorithm: Exponential Weights

From Section 3.3, we know how to obtain the estimation for $(\mu^\pi)_{\pi \in \Pi}$. Now we can use them to construct estimators of $(\phi^\pi)_{\pi \in \Pi}$ via $\hat{\phi}_h^\pi = \sum_{s \in \mathcal{X}_h} \sum_{a \in \mathcal{A}} \hat{\mu}^\pi(s)\pi(a|s)\phi(s,a)$, and run a linear bandit algorithm viewing $(\hat{\phi}^\pi)_{\pi \in \Pi}$ as actions. The algorithm is presented in Algorithm 2. At the beginning of each episode $k$, we call EstOM (Algorithm 1) for all policies with the data up to episode $k-1$ (Line 5). This returns the occupancy measure estimator $\hat{\mu}_k^\pi$ for all $\pi$, which we can use to construct the feature estimator $\hat{\phi}_k^\pi$. Then we use the standard exponential weight together with John's exploration to update the distribution over policies. To deal with the bias induced by the estimation error of $\hat{\phi}_k^\pi$, we incorporate a bonus term $b_k^\pi$ in the update. Similar ideas have also been used in, e.g., Luo et al. (2021); Sherman et al. (2023b); Dai et al. (2023); Kong et al. (2023); Liu et al. (2023a). We defer the regret analysis of this algorithm to Appendix B.3, and only state the final guarantee in the next theorem.

**Theorem 7.** *The regret of Algorithm 2 is bounded by $\mathcal{R}_K \leq \widetilde{\mathcal{O}}(\sqrt{d^7 H^7 K})$.*

---

**Algorithm 2** Exponential Weights

1: Let $\Pi$ be the policy set defined in Eq. (5). Let $\gamma = \min\left\{ d^2 H^{\frac{1}{2}} K^{-\frac{1}{2}}, \frac{1}{2} \right\}, \eta = \frac{\gamma}{2dH}$.
2: For all $h \in [H], \mathcal{D}_{1,h} \leftarrow \emptyset, \Lambda_{1,h} \leftarrow I$.
3: **for** $k = 1, 2, \ldots$ **do**
4:      For all $\pi \in \Pi$, let $\hat{\mu}_k^\pi = \text{EstOM}(\pi, (\mathcal{D}_{k,h})_{h=1}^H)$ (call Algorithm 1).
5:      Define $\hat{\phi}_{k,h}^\pi = \sum_{s \in \mathcal{X}_h} \sum_{a \in \mathcal{A}} \hat{\mu}^\pi(s)\pi(a|s)\phi(s,a)$ and $\hat{\phi}_k^\pi = (\hat{\phi}_{k,1}^\pi, \ldots, \hat{\phi}_{k,H}^\pi)$.
6:      Compute $q_k \in \Delta(\Pi)$ as $q_k(\pi) \propto \exp\left( -\eta \sum_{i=1}^{k-1} \left( \hat{\phi}_i^{\pi\top} \hat{\theta}_i - b_i^\pi \right) \right)$.
7:      Let $q_k' = (1-\gamma)q_k + \gamma J_k$ where $J_k \in \Delta(\Pi)$ is John's exploration over $\{\hat{\phi}_k^\pi\}_{\pi \in \Pi}$.
8:      Sample $\pi_k \sim q_k'$, execute $\pi_k$, and obtain trajectory $(s_{k,1}, a_{k,1}, \ell_{k,1}, \ldots, s_{k,H}, a_{k,H}, \ell_{k,H})$.
9:      Define for $C_{\text{bonus}} = 10d^{\frac{5}{4}} H \sqrt{\log \frac{18d^{\frac{3}{2}}K}{\delta}}$,

$$M_k = \sum_{\pi \in \Pi} q_k'(\pi)\hat{\phi}_k^\pi(\hat{\phi}_k^\pi)^\top, \qquad \hat{\theta}_k = M_k^{-1}\hat{\phi}_k^{\pi_k} L_k, \qquad \text{where } L_k = \sum_{h=1}^H \ell_{k,h},$$

$$b_k^\pi = C_{\text{bonus}} \sum_{h=1}^H \sum_{s \in \mathcal{X}_h} \sum_{a \in \mathcal{A}} \hat{\mu}_k^\pi(s,a) \|\phi(s,a)\|_{\Lambda_{k,h}^{-1}} + \eta \|\hat{\phi}_k^\pi\|_{M_k^{-1}}^2.$$

10:      For all $h \in [H]$,

$$\mathcal{D}_{k+1,h} \leftarrow \mathcal{D}_{k,h} \cup \{(s_{k,h}, a_{k,h}, s_{k,h+1})\}, \quad \Lambda_{k+1,h} \leftarrow \Lambda_{k,h} + \phi(s_{k,h}, a_{k,h})\phi(s_{k,h}, a_{k,h})^\top$$

11: **end for**

---

## 4 Computationally Efficient Policy Optimization Algorithm

In Algorithm 2, we convert the linear MDP problem to a linear bandit problem. It is generally hard to ensure computational efficiency in this paradigm due to the non-linear mapping of policy to occupancy measure and the exponential size of the policy space. A promising alternative is to use

the policy optimization framework (Luo et al., 2021; Dai et al., 2023; Sherman et al., 2023b), which allows to run a Follow-the-Regularized-Leader (FTRL) algorithm over the locally available state-action feature set. An algorithm of this type needs to overcome several hurdles: 1) The algorithm needs to construct loss estimates with carefully controlled bias, which is difficult because the learner does not know the feature covariance matrix under the current policy and has to estimate it. 2) The algorithm needs to inject bonus to ensure sufficient exploration. These bonus terms not only need to compensate the uncertainty in transitions, but also the bias induced in loss estimates. The bonus *itself* also needs to be estimated and induces more bias due to the estimation error.

These challenges are fully exposed in the *adversarial loss*, *bandit feedback*, *unknown transition* setting, because in this case the loss estimators usually have larger magnitudes and necessitate larger bonuses. This make achieving near-optimal bounds difficult, and the current best regret is $\widetilde{\mathcal{O}}(K^{6/7})$ by Sherman et al. (2023b). We successfully improve it to $\widetilde{\mathcal{O}}(K^{3/4})$ by several improved design choices, which we describe in the following.

---

**Algorithm 3** Logdet FTRL with initial exploration

---

1: **Parameters:** $\eta = \frac{1}{3328\sqrt{d}H^2}K^{-\frac{1}{4}}$, $\gamma = 5d\log\left(6dHK^4\right)K^{-\frac{1}{2}}$, $\beta = \sqrt{d}K^{-\frac{1}{4}}$, $\alpha = HK^{\frac{3}{4}}$, $\tau = K^{\frac{1}{2}}$, $\delta = K^{-3}$, $\rho = H^{-\frac{1}{2}}d^{-\frac{1}{4}}K^{-\frac{1}{4}}$, $\epsilon_{\mathrm{cov}} = K^{-\frac{1}{4}}$.

2: **Define:** $\widehat{\mathrm{Cov}}(s,p) = \mathbb{E}_{a\sim p}\begin{bmatrix} \phi(s,a)\phi(s,a)^\top & \phi(s,a) \\ \phi(s,a)^\top & 1 \end{bmatrix}$

3: Run Algorithm 5 with parameters $\delta, \rho, \epsilon_{\mathrm{cov}}$, which ends within $K_0 = \widetilde{\mathcal{O}}(d^{\frac{3}{2}}H^2K^{\frac{3}{4}} + d^4H^4K^{\frac{1}{4}})$ episodes with high probability. Receive outputs $(\mathcal{D}_{0,h})_{h=1}^H$ and $(\mathcal{Z}_h)_{h=1}^H$.

4: **for** $j = 1, \ldots, \lceil(K - K_0)/(2\tau)\rceil$ **do**

5:     For $s \in \mathcal{S}_h$, define

$$\widetilde{\boldsymbol{H}}_j(s) = \underset{\boldsymbol{H}\in\mathcal{H}_s}{\mathrm{argmin}}\left\{\left\langle\boldsymbol{H}, \sum_{i=1}^{j-1}\boldsymbol{\mathcal{L}}_{i,h}\right\rangle + \frac{F(\boldsymbol{H})}{\eta}\right\}, \text{ where } \boldsymbol{\mathcal{L}}_{i,h} = \frac{1}{2\tau}\sum_{k\in T_i}\left(\widehat{\boldsymbol{\Gamma}}_{k,h} - \widehat{\boldsymbol{B}}_{k,h}\right)$$

    where $\mathcal{H}_s = \left\{\widehat{\mathrm{Cov}}(s,p) : p \in \Delta(\mathcal{A})\right\}$ and $F(\boldsymbol{H}) = -\log\det(\boldsymbol{H})$.

6:     Let $\widetilde{\pi}_j(\cdot|s)$ be such that $\widetilde{\boldsymbol{H}}_j(s) = \widehat{\mathrm{Cov}}(s, \widetilde{\pi}_j(\cdot|s))$.

7:     Let $T_j = \{(j-1)\tau + K_0 + 1, \cdots, (j+1)\tau + K_0\}$. Execute $\pi_k = \widetilde{\pi}_j$ for the $2\tau$ episodes $k \in T_j$, and collect $(s_{k,h}, a_{k,h}, \ell_{k,h})_{h\in[H], k\in T_j}$.

8:     Let $T_{j,1}$ and $T_{j,2}$ be the first $\tau$ and the last $\tau$ episodes in $T_j$, respectively. For all $k \in T_j$ and $h \in [H]$, define

$$\mathcal{C}_{k,h} = \begin{cases} \{(s_{k',h}, a_{k',h}, s_{k',h+1})\}_{k'\in T_{j,2}} & \text{if } k \in T_{j,1} \\ \{(s_{k',h}, a_{k',h}, s_{k',h+1})\}_{k'\in T_{j,1}} & \text{if } k \in T_{j,2} \end{cases} \tag{10}$$

$$\widehat{\Sigma}_{k,h} = \gamma I + \frac{1}{\tau}\sum_{(s,a,s')\in\mathcal{C}_{k,h}}\phi(s,a)\phi(s,a)^\top \tag{11}$$

$$\widehat{q}_{k,h} = \widehat{\Sigma}_{k,h}^{-1}\phi(s_{k,h}, a_{k,h})\sum_{t=h}^H\ell_{k,t} \tag{12}$$

$$\widehat{\boldsymbol{\Gamma}}_{k,h} = \begin{bmatrix} 0 & \frac{1}{2}\widehat{q}_{k,h} \\ \frac{1}{2}(\widehat{q}_{k,h})^\top & 0 \end{bmatrix} \tag{13}$$

$$\mathcal{D}_{k,h} = \mathcal{D}_{k-1,h} \cup \{(s_{k,h}, a_{k,h}, s_{k,h+1})\} \tag{14}$$

$$(\widehat{\boldsymbol{B}}_{k,h})_{h=1}^H = \mathrm{OBME}\left((\mathcal{D}_{k,h})_{h=1}^H, (\widehat{\Sigma}_{k,h})_{h=1}^H, (\mathcal{Z}_h)_{h=1}^H\right) \tag{15}$$

    (OBME is presented in Algorithm 4)

9: **end for**

---

Our algorithm (Algorithm 3) starts with an initial pure exploration phase that lasts for $K_0 = \widetilde{\mathcal{O}}(K^{\frac{3}{4}})$ episodes (Line 3), which is crucial in controlling the magnitude of the bonus estimate (will be explained later). In the remaining $K - K_0$ episodes, episodes are divided into $\lceil(K - K_0)/(2\tau)\rceil$ epochs (indexed by $j$), such that in each epoch $j$, a fixed policy $\widetilde{\pi}_j$ is executed for $2\tau$ episodes, and policies are updated only at the end of each epoch. The goal of dividing episodes into epochs is to let the learner collect sufficient samples and create accurate enough loss estimators for each update.

Different from previous work (Luo et al., 2021; Dai et al., 2023; Sherman et al., 2023b) that use exponential weights, we use the Follow-the-Regularized-Leader (FTRL) framework with logdet-barrier as the regularizer for policy updates. Logdet has been recently shown in adversarial linear (contextual) bandit to lead to a more stable update and can handle larger bias for loss estimators (Zimmert and Lattimore, 2022; Liu et al., 2023a). It has similar benefits in our case as well.

Specifically, with logdet-FTRL, the optimization of the policy on state $s$ is over the space of *lifted covariance matrix* $\mathcal{H}_s = \left\{ \widehat{\mathrm{Cov}}(s, p) : p \in \Delta(\mathcal{A}) \right\} \subset \mathbb{R}^{(d+1)\times(d+1)}$, where $\widehat{\mathrm{Cov}}(s, p) = \mathbb{E}_{a \sim p} \begin{bmatrix} \phi(s,a)\phi(s,a)^\top & \phi(s,a) \\ \phi(s,a)^\top & 1 \end{bmatrix}$. In epoch $j$, for state $s$, the FTRL outputs a matrix $\widetilde{\boldsymbol{H}}_j(s) \in \mathcal{H}_s$ (Line 5), and the policy $\widetilde{\pi}_j(\cdot|s)$ is chosen such that $\widetilde{\boldsymbol{H}}_j(s) = \widehat{\mathrm{Cov}}(s, \widetilde{\pi}_j(\cdot|s))$ (Line 6). This policy is then executed for $2\tau$ episodes (Line 7). Then the learner uses the collected samples to construct loss estimators for all episodes $k \in T_j$ (the $\widehat{\mathrm{q}}_{k,h}$ in Eq. (12)), where $T_j$ is the set of episodes in epoch $j$. This follows the standard loss estimator construction for linear bandits, except that in our case, the covariance matrix is unknown and also needs to be estimated using samples (the $\widehat{\Sigma}_{k,h}$ in Eq. (11)). The validity of $\widehat{\mathrm{q}}_{k,h}$ relies on the independence between $\widehat{\Sigma}_{k,h}$ and the loss obtained in episode $k$. To achieve this, we divide the set $T_j$ into two equal parts $T_{j,1}$ and $T_{j,2}$ (Line 8). Then we use samples from $T_{j,2}$ to estimate the covariance matrix when constructing the loss estimator in episode $k \in T_{j,1}$, and vice versa (Eq. (10)-Eq. (12)). In Eq. (13), we further lift the loss estimator $\widehat{\mathrm{q}}_{k,h}$ to $\widehat{\boldsymbol{\Gamma}}_{k,h} \in \mathbb{R}^{(d+1)\times(d+1)}$ to be fed to FTRL. Finally, besides feeding the loss $\widehat{\boldsymbol{\Gamma}}_{k,h}$, we also need to feed the *bonus* $\widehat{\boldsymbol{B}}_{k,h}$ required for sufficient exploration in policy optimization and to compensate the loss estimator bias coming from the estimation error of $\widehat{\Sigma}_{k,h}$. This is explained in the next subsection.

## 4.1 THE EXPLORATION BONUS

Similar to previous work on policy optimization in adversarial linear MDPs (Luo et al., 2021; Dai et al., 2023; Sherman et al., 2023b), we use *exploration bonus* to address the bias in the loss estimator $\widehat{\mathrm{q}}_{k,h}$ and the stability term coming from the FTRL regret analysis. From a high level, the exploration bonus serves a similar purpose as "optimism in the face of uncertainty" as commonly used in the non-adversarial case, but now the sources of uncertainty additionally include the bias and the stability term. From a mathematical analysis perspective, the exploration bonus creates an effect of *change of measure* that prevent the regret to depend on the distribution mismatch coefficient between the optimal policy and the learner's policy. This perspective is best explained in Section 3 of Luo et al. (2021). According to the analysis of Luo et al. (2021), when performing policy update on state $s \in \mathcal{S}_h$, we should incorporate a bonus that is roughly of order $Q^{\pi_k}(s, a; b_t)$ where $b_t(s, a) = \beta \|\phi(s,a)\|_{\widehat{\Sigma}_{k,h}^{-1}}^2$.

Our bonus construction further incorporates the improvement from Sherman et al. (2023b) where an optimistic least-square policy evaluation (OLSPE) is used to fit the bonus (rather than sampling the bonus as in Luo et al. (2021)). This creates another term of $\alpha \|\phi(s,a)\|_{\Lambda_{k,h}^{-1}}^2$ to be incorporated into the bonus to compensate the estimation error of future bonuses. Finally, we further adopt a technique developed in Luo et al. (2021) called *dilated bonus* to simplify our analysis. Overall, the bonus we use for the policy update on state $s \in \mathcal{S}_h$ is defined recursively as

$$B_k(s,a) \approx \left( \beta \|\phi(s,a)\|_{\widehat{\Sigma}_{k,h}^{-1}}^2 + \alpha \|\phi(s,a)\|_{\Lambda_{k,h}^{-1}}^2 \right) + \left( 1 + \frac{1}{H} \right) \mathbb{E}_{s' \sim P(\cdot|s,a)} \mathbb{E}_{a' \sim \pi_k(\cdot|s')} \left[ B_k(s', a') \right].$$

Notice that because of the dilation factor $(1 + \frac{1}{H})$ (Luo et al., 2021), this deviates from a standard Bellman equation. Recall that we run FTRL in the space of covariance matrix, so we would like to write $B_k(s,a)$ as a *linear* function in that space. Fortunately, this is indeed possible because by the linear MDP structure, we can write the above as

$$B_k(s,a) \approx \left\langle \begin{bmatrix} \phi(s,a)\phi(s,a)^\top & \phi(s,a) \\ \phi(s,a)^\top & 1 \end{bmatrix}, \begin{bmatrix} \beta\widehat{\Sigma}_{k,h}^{-1} + \alpha\Lambda_{k,h}^{-1} & \frac{1}{2}w_{k,h} \\ \frac{1}{2}w_{k,h} & 0 \end{bmatrix} \right\rangle \tag{16}$$

where $w_{k,h} = (1 + \frac{1}{H}) \sum_{s' \in \mathcal{S}_{h+1}} \psi(s') \mathbb{E}_{a' \sim \pi_k(\cdot|s')} [B_k(s', a')]$. The purpose of Algorithm 4 is exactly to inductively find an estimator $\widehat{w}_{k,h}$ of $w_{k,h}$ for all $h$. Then, we can form a *bonus matrix* as the second matrix in Eq. (16) (but replacing $w_{k,h}$ by $\widehat{w}_{k,h}$) and feed it to the FTRL algorithm.

There are two technical complications regarding Algorithm 4. First, in order to control the magnitude of $\widehat{w}_{k,h}$, we have to control the magnitude of $\alpha\|\phi(s,a)\|^2_{\Lambda^{-1}_{k,h}}$. This can be done by adding a pure exploration phase in the beginning of the algorithm (Line 3 of Algorithm 3) and form a *known state space* $\mathcal{Z} \subset \mathcal{S}$. Known states are well-explored in the initial phase, and the values of $\|\phi(s,a)\|^2_{\Lambda^{-1}_{k,h}}$ on them are sufficiently small (in our case are of order $1/\sqrt{K}$). On the other hand, unknown states are hard to be reached by any policy (in our case, their probability of being reached is $\leq K^{-\frac{1}{4}}$) and thus can be ignored in the learning phase. The initial exploration phase is inspired by Sherman et al. (2023a), who further built their algorithm on Wagenmaker et al. (2022b)'s reward-free exploration algorithm. We provide the guarantees for the initial exploration phase in Appendix C. The other is that in order to ensure only positive bonuses are propagated over layers under estimation error of $\widehat{w}_{k,h}$, we force the bonus-to-go estimation to be non-negative in Line 8. The additional penalty is related to $\|\widehat{w}_{k,h} - w_{k,h}\|$ and can be well-controlled.

---

**Algorithm 4** OBME$\left((\mathcal{D}_{k,h})_{h=1}^H, (\widehat{\Sigma}_{k,h})_{h=1}^H, (\mathcal{Z}_h)_{h=1}^H\right)$ (**O**ptimistic **B**onus **M**atrix **E**stimation)

---

1: Parameters $\beta, \alpha, \gamma, \rho$ are the same as those in Algorithm 3.
2: **for** $h = H, \ldots, 1$ **do**
3:      $B_h^{\max} = 4H\left(1 + \frac{1}{H}\right)^{2(H-h+1)}\left(\frac{\beta}{\gamma} + \alpha\rho^2\right)$
4:      $\Lambda_{k,h} = I + \sum_{(s,a,s')\in\mathcal{D}_{k,h}}\phi(s,a)\phi(s,a)^\top$
5:      Set $\widehat{w}_{k,h} = \left(1 + \frac{1}{H}\right)\Lambda^{-1}_{k,h}\sum_{(s,a,s')\in\mathcal{D}_{k,h}}\phi(s,a)\widehat{W}_k(s')\mathbb{I}\{s' \in \mathcal{Z}_{h+1}\}$      (if $h = H$, set $\widehat{w}_{k,h} = 0$)
6:      Define $\widehat{\boldsymbol{B}}_{k,h} = \begin{bmatrix} \beta\widehat{\Sigma}^{-1}_{k,h} + \alpha\Lambda^{-1}_{k,h} & \frac{1}{2}\widehat{w}_{k,h} \\ \frac{1}{2}\widehat{w}^\top_{k,h} & 0 \end{bmatrix}$
7:      For $s \in \mathcal{S}_h$, define $\widehat{B}_k(s,a) = \beta\|\phi(s,a)\|^2_{\widehat{\Sigma}^{-1}_{k,h}} + \alpha\|\phi(s,a)\|^2_{\Lambda^{-1}_{k,h}} + \phi(s,a)^\top\widehat{w}_{k,h}$
8:      For $s \in \mathcal{S}_h$, define $\widehat{W}_k(s) = \langle\pi_k(\cdot|s), \widehat{B}^+_k(s,\cdot)\rangle$ where $\widehat{B}^+_k(s,a) = \max\left\{\widehat{B}_k(s,a), 0\right\}$
9: **end for**
10: **return** $(\widehat{\boldsymbol{B}}_{k,h})_{h\in[H]}$

---

## 4.2 REGRET GUARANTEE

We defer the analysis of Algorithm 3 to Appendix D, and only state the final regret bound in the following theorem.

**Theorem 8.** *Algorithm 3 ensures a regret of order* $\mathcal{R}_K = \widetilde{\mathcal{O}}(d^{\frac{3}{2}}H^3K^{\frac{3}{4}})$.

The improvement in our regret primarily stems from two sources. Firstly, we utilize an improved matrix concentration bound from Liu et al. (2023a). This ensures that using $\tau = \frac{1}{\gamma}$ episodes (where $\gamma$ is the parameter in Eq. (11)) is enough to gather data and build a reliable loss estimator. In contrast, previous works require $\tau = \frac{1}{\gamma^2}$ (Dai et al., 2023; Sherman et al., 2023b) or $\tau = \frac{1}{\gamma^3}$ (Luo et al., 2021), thereby consuming excessive episodes to accumulate data for a single policy and consequently slowing down policy updates. Secondly, in previous works (Luo et al., 2021; Dai et al., 2023; Sherman et al., 2023b), the usage of exponential weights requires $\eta$ to be small compared to the magnitude of both loss estimators and exploration bonus. This prevents them from choosing the best $\eta$ in their algorithms. With the help of logdet barrier, in our algorithm, $\eta$ only needs to be small compared to the magnitude of the exploration bonus, which is already small given the initial exploration phase. This gives us more flexibility in choosing $\eta$.

## 5 CONCLUSION

In this work, we obtain the first optimal $\sqrt{K}$ regret bound for adversarial linear MDPs under bandit feedback and unknown transitions without the help of simulators or generative models. We also give a new $K^{3/4}$ regret bound with an efficient policy optimization algorithm. We hope that the techniques and observations in the work could be helpful in developing an algorithm that is both statistically optimal and computationally efficient.

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

## A RELATED WORKS

In this subsection, we review prior works on adversarial MDPs and policy optimization.

**Learning in Adversarial MDPs.** Adversarial MDPs refer to a class of MDP problems where the transition is fixed while the loss function changes over time. Learning adversarial *tabular* MDPs under bandit feedback and unknown transition has been extensively studied (Rosenberg and Mansour, 2019; Jin et al., 2020a; Lee et al., 2020; Jin et al., 2021; Shani et al., 2020; Chen and Luo, 2021; Luo et al., 2021; Dai et al., 2022; Dann et al., 2023a). In this line of work, not only $\sqrt{K}$ regret bounds have been shown, several data-dependent bounds are also established. For adversarial MDPs with a large state space which necessitates the use of function approximation, $\sqrt{K}$ bounds have only been shown under simpler cases such as 1) full-information loss feedback (Cai et al., 2020; He et al., 2022; Sherman et al., 2023a), and 2) known transition or access to generative models / simulators (Neu and Olkhovskaya, 2021; Dai et al., 2023; Foster et al., 2022). Therefore, to our knowledge, we provide the first $\sqrt{K}$ regret for adversarial MDPs with large state spaces under bandit feedback and unknown transitions.[2] For linear MDPs, a series of recent work has made significant progress in improving the regret bound: Luo et al. (2021), Dai et al. (2023), Sherman et al. (2023b) proposed efficient (polynomial-time) algorithms with $K^{14/15}$, $K^{8/9}$, and $K^{6/7}$ regret, respectively, and Kong et al. (2023) proposed an inefficient algorithm with $K^{4/5} + \text{poly}(1/\lambda_{\min})$ regret. Our $\sqrt{K}$ regret through an inefficient algorithm and $K^{3/4}$ regret through an efficient algorithm further push the frontiers. These results are summarized in Table 1.

Table 1: Related works for learning adversarial linear MDPs without a simulator. An algorithm is efficient if its computational complexity is polynomial with action size $|\mathcal{A}|$, dimension $d$, and $K$. For the Type column, PO means the algorithm is based on policy optimization while OM means the algorithm is based on occupancy measure estimation.

| Algorithm | Regret | Efficient | Type | Assumption |
|---|---|---|---|---|
| Luo et al. (2021) | $\widetilde{\mathcal{O}}\left(K^{14/15}\right)$ | ✓ | PO | |
| Dai et al. (2023) | $\widetilde{\mathcal{O}}\left(K^{8/9}\right)$ | ✓ | PO | |
| Sherman et al. (2023b) | $\widetilde{\mathcal{O}}\left(K^{6/7}\right)$ | ✓ | PO | |
| Kong et al. (2023) | $\widetilde{\mathcal{O}}\left(K^{4/5}\right)$ | ✗ | OM | $\exists \pi, \Sigma_\pi \succeq \lambda I$ |
| Algorithm 2 (**ours**) | $\widetilde{\mathcal{O}}\left(\sqrt{K}\right)$ | ✗ | OM | |
| Algorithm 3 (**ours**) | $\widetilde{\mathcal{O}}\left(K^{3/4}\right)$ | ✓ | PO | |

**Policy Optimization with Exploration.** Policy optimization has been regarded as sample inefficient due to its local search nature. Recently, efforts to alleviate this issue have incorporated exploration bonus in policy updates (Agarwal et al., 2020; Shani et al., 2020; Zanette et al., 2021; Luo et al., 2021; Dai et al., 2023; Sherman et al., 2023b; Zhong and Zhang, 2023; Liu et al., 2023b; Sherman et al., 2023a). In the case of linear MDPs with a *fixed* loss function, the state-of-the-art result is by Sherman et al. (2023a), who provide a computationally efficient policy optimization algorithm with a tight $\sqrt{K}$ regret. In the case of linear MDPs with *adversarial* losses, the best existing regret bound is $K^{6/7}$ by Sherman et al. (2023b), while we improve it to $K^{3/4}$ in this paper. Beyond theoretical advancement, exploration in policy optimization has also showcased its potential in addressing real-world challenges, as evidenced by empirical studies (Burda et al., 2018; Pan et al., 2019).

---

[2]Although Zhao et al. (2022) provided a $\sqrt{K}$ regret bound for linear mixture MDPs with bandit feedback and unknown transition, the polynomial dependence on the number of states prohibits its application to MDPs with large state spaces.

# B    OMITTED DETAILS IN SECTION 3

## B.1    POLICY SPACE DISCRETIZATION

*Proof of Lemma 4.* Let $\bar{\theta}_h = \sum_{k=1}^{K} \theta_{k,h}$ and let $\bar{\ell}(s, a) = \langle \phi(s, a), \bar{\theta}_h \rangle$ for $s \in \mathcal{S}_h$ be the loss function under the loss vector $\bar{\theta}$. Under this loss function, the Q-function of a policy $\pi$ can be written as

$$Q^\pi(s, a; \bar{\ell}) = \phi(s, a)^\top \xi_h^\pi \quad \text{for } s \in \mathcal{S}_h,$$

where $\xi_h^\pi$ is recursively defined as

$$\xi_h^\pi = \bar{\theta}_h + \sum_{s' \in \mathcal{S}_{h+1}} \psi(s') \sum_{a' \in \mathcal{A}} \pi(a'|s') \langle \phi(s', a'), \xi_{h+1}^\pi \rangle.$$

Notice that by Definition 2, we have $\|\xi_h^\pi\|_2 \leq H\sqrt{d}K$. Let $\pi^\star$ be the optimal policy under loss function $\bar{\ell}$. Then by Bellman's optimality equation, $\pi^\star$ can be represented as

$$\pi^\star(s) = \operatorname*{argmin}_a \left\{ \phi(s, a)^\top \xi_h^{\pi^\star} \right\}$$

and $\xi_h^{\pi^\star}$ can be found recursively from layer $H$ to layer $1$.

Now, let $\xi_h'$ be the closest element to $\xi_h^{\pi^\star}$ in the $H\sqrt{d}$-net of $\mathbb{B}^d(H\sqrt{d}K)$, and let $\pi'$ be the policy induced by $\xi' = (\xi_1', \ldots, \xi_H')$, i.e.,

$$\pi'(s) = \operatorname*{argmin}_a \left\{ \phi(s, a)^\top \xi_h' \right\}.$$

Then for any $\pi$, we have

$$\sum_{k=1}^{K} \sum_{h=1}^{H} \sum_{s \in \mathcal{S}_h} \sum_{a \in \mathcal{A}} (\mu^{\pi'}(s, a) - \mu^\pi(s, a)) \phi(s, a)^\top \theta_{k,h}$$

$$= \sum_{k=1}^{K} \sum_{h=1}^{H} \sum_{s \in \mathcal{S}_h} \sum_{a \in \mathcal{A}} (\mu^{\pi^\star}(s, a) - \mu^\pi(s, a)) \phi(s, a)^\top \theta_{k,h} + \sum_{k=1}^{K} \sum_{h=1}^{H} \sum_{s \in \mathcal{S}_h} \sum_{a \in \mathcal{A}} (\mu^{\pi'}(s, a) - \mu^{\pi^\star}(s, a)) \phi(s, a)^\top \theta_{k,h}$$

$$= V^{\pi^\star}(s_1; \bar{\ell}) - V^\pi(s_1; \bar{\ell}) + \sum_{h=1}^{H} \sum_{s \in \mathcal{S}_h} \mu^{\pi'}(s) \sum_{a \in \mathcal{A}} (\pi'(a|s) - \pi^\star(a|s)) \phi(s, a)^\top \xi_h^\star$$

$$\text{(by the performance difference lemma)}$$

$$\leq 0 + \sum_{h=1}^{H} \sum_{s \in \mathcal{S}_h} \mu^{\pi'}(s) \sum_{a \in \mathcal{A}} (\pi'(a|s) - \pi^\star(a|s)) \phi(s, a)^\top \xi_h' + H^2\sqrt{d}$$

$$\text{(by the optimality of } \pi^\star \text{ under } \bar{\ell} \text{ and the discretization error)}$$

$$\leq H^2\sqrt{d}$$

where the last inequality is by the fact that $\pi'$ takes the argmin with respect to $\xi_h'$. Finally, notice that policy $\pi'$ belongs to $\Pi$ corresponding to the parameter $\theta_h = \frac{1}{H\sqrt{d}} \xi_h'$.

$\square$

## B.2 FEATURE ESTIMATION

*Proof of Lemma 5.* $\mu^\pi(s)$ satisfies Eq. (7) because $\mu^\pi$ is a valid occupancy measure. To show Eq. (8), notice that

$$
\left| \sum_{s\in\mathcal{X}_h}\sum_{a\in\mathcal{A}} \mu^\pi(s)\pi(a|s)\,\mathrm{clip}\left[\phi(s,a)^\top \xi^\star_{h,f}\right] - \sum_{s\in\mathcal{X}_{h+1}} \mu^\pi(s')f(s') \right|
$$

$$
= \left| \sum_{s\in\mathcal{S}_h}\sum_{a\in\mathcal{A}} \mu^\pi(s)\pi(a|s)\,\mathrm{clip}\left[\phi(s,a)^\top \xi^\star_{h,f}\right] - \sum_{s\in\mathcal{S}_{h+1}} \mu^\pi(s')f(s') \right| \quad (\mu^\pi(s)=0 \text{ for } s\in\mathcal{X}\setminus\mathcal{S})
$$

$$
= \left| \sum_{s\in\mathcal{S}_h}\sum_{a\in\mathcal{A}} \mu^\pi(s)\pi(a|s)\,\mathrm{clip}\left[\phi(s,a)^\top \sum_{s'\in\mathcal{S}_{h+1}} \psi(s')f(s') \right] - \sum_{s\in\mathcal{S}_{h+1}} \mu^\pi(s')f(s') \right|
$$

$$
= \left| \sum_{s\in\mathcal{S}_h}\sum_{a\in\mathcal{A}} \mu^\pi(s)\pi(a|s)\,\mathrm{clip}\left[ \sum_{s'\in\mathcal{S}_{h+1}} P(s'|s,a)f(s') + z \right] - \sum_{s\in\mathcal{S}_{h+1}} \mu^\pi(s')f(s') \right|
$$

$$
\text{(for some } z \text{ such that } |z|\le\zeta \text{ by Definition 3)}
$$

$$
\le \left| \sum_{s\in\mathcal{S}_h}\sum_{a\in\mathcal{A}} \mu^\pi(s)\pi(a|s)\,\mathrm{clip}\left[ \sum_{s'\in\mathcal{S}_{h+1}} P(s'|s,a)f(s') \right] - \sum_{s\in\mathcal{S}_{h+1}} \mu^\pi(s')f(s') \right| + \zeta
$$

$$
= \zeta \tag{17}
$$

Finally, we show Eq. (9). For simplicity, let $\mathcal{D}_h = \{(s_i, a_i, s'_i)\}_{i=1}^n$ and let $\phi_i = \phi(s_i, a_i)$. We first consider a fixed policy $\pi$ and a layer $h$. Let $\epsilon = \frac{1}{K}$, and let $\mathcal{N}_{\epsilon,1}$ be an $\epsilon$-net of $\mathcal{F}^\pi$ on layer $h$ so that for any $f\in\mathcal{F}^\pi$, there exists an $f'\in\mathcal{N}_{\epsilon,1}$ such that $|f'(s)-f(s)|\le\epsilon$ for all $s\in\mathcal{X}_h$. Let $\mathcal{N}_{\epsilon,2}$ be the $\epsilon$-net of $\mathbb{B}^d(\sqrt{d})$. Furthermore, define $|\Pi_h| = (3K)^d$ (whose meaning will be clear later).

Then under this fixed $\pi$, for any $\xi\in\mathcal{N}_{\epsilon,2}$ any $f\in\mathcal{N}_{\epsilon,1}$, with probability at least $1-\frac{\delta}{|\mathcal{N}_{\epsilon,1}||\mathcal{N}_{\epsilon,2}||\Pi_h|K}$,

$$
\sum_{i=1}^n \left(f(s'_i) - \phi_i^\top \xi^\star_{h,f}\right)^2 - \sum_{i=1}^n \left(f(s'_i) - \phi_i^\top \xi\right)^2
$$

$$
= -2\sum_{i=1}^n (f(s'_i) - \phi_i^\top \xi^\star_{h,f})\left(\phi_i^\top \xi^\star_{h,f} - \phi_i^\top \xi\right) - \sum_{i=1}^n \left(\phi_i^\top \xi^\star_{h,f} - \phi_i^\top \xi\right)^2
$$

$$
\le -2\sum_{i=1}^n (f(s'_i) - \mathbb{E}_{s'\sim P(\cdot|s_i,a_i)}[f(s')])\left(\phi_i^\top \xi^\star_{h,f} - \phi_i^\top \xi\right) - \sum_{i=1}^n \left(\phi_i^\top \xi^\star_{h,f} - \phi_i^\top \xi\right)^2 + 2\sqrt{d}n\zeta
$$

$$
\le 6\sqrt{\sum_{i=1}^n \left(\phi_i^\top \xi^\star_{h,f} - \phi_i^\top \xi\right)^2 \log\frac{|\mathcal{N}_{\epsilon,1}||\mathcal{N}_{\epsilon,2}||\Pi_h|K}{\delta}} + 2\sqrt{d}\log\frac{|\mathcal{N}_{\epsilon,1}||\mathcal{N}_{\epsilon,2}||\Pi_h|K}{\delta}
$$

$$
- \sum_{i=1}^n \left(\phi_i^\top \xi^\star_{h,f} - \phi_i^\top \xi\right)^2 + 2\sqrt{d}n\zeta \qquad \text{(Freedman's inequality)}
$$

$$
\le 7\sqrt{d}\log\frac{|\mathcal{N}_{\epsilon,1}||\mathcal{N}_{\epsilon,2}||\Pi_h|K}{\delta} + 2\sqrt{d}n\zeta. \qquad \text{(AM-GM)}
$$

Below, we take a union bound over $f\in\mathcal{N}_{\epsilon,1}$, $\xi\in\mathcal{N}_{\epsilon,2}$, and $\pi\in|\Pi|$. Notice that although the size of the policy set is $|\Pi|\le(3K)^{dH}$ (a product of $H$ $\frac{1}{K}$-net for $\mathbb{B}^d(1)$), when considering the policies over layer $h$, the total number of different policies is only $|\Pi_h|\le(3K)^d$. Therefore, a union bound over policies require only a size of $|\Pi_h|$. Bounding the distance between the full sets and $\epsilon$-nets, we

conclude that with probability at least $\frac{\delta}{K}$, for all $\xi \in \mathbb{B}^d(\sqrt{d})$, all $\pi \in \Pi$, and all $f \in \mathcal{F}^\pi$,

$$\sum_{i=1}^{n} \left(f(s_i') - \phi_i^\top \xi_{h,f}^\star\right)^2 - \sum_{i=1}^{n} \left(f(s_i') - \phi_i^\top \xi\right)^2 \leq 7\sqrt{d} \log \frac{|\mathcal{N}_{\epsilon,1}||\mathcal{N}_{\epsilon,2}||\Pi_h|K}{\delta} + 2\sqrt{d}n\zeta + \sqrt{d}n\epsilon.$$
(18)

By our choice of $\zeta$ and $\epsilon$, the second and third terms above are both negligible compared to the first term. Finally, we bound $|\mathcal{N}_{\epsilon,1}|$ and $|\mathcal{N}_{\epsilon,2}|$ via Lattimore and Szepesvári (2020) (Exercise 27.6). $|\mathcal{N}_{\epsilon,2}|$ is the size of the $\epsilon$-net of $\mathbb{B}^d(\sqrt{d})$, equivalently the $(\epsilon/\sqrt{d})$-net of $\mathbb{B}^d(1)$, which is upper bounded by $(3\sqrt{d}/\epsilon)^d$. By the definition of $\mathcal{F}^\pi$, the $\epsilon$-net of $\mathcal{F}^\pi$ would be the union of the $\epsilon$-nets of $\{\theta : \theta \in \mathbb{B}^d(\sqrt{d})\}$ and $\{\Gamma \in \mathbb{R}^{d \times d} : \mathbf{0} \preceq \Gamma \preceq I\}$. Thus $|\mathcal{N}_{\epsilon,1}| = (6d^{\frac{3}{2}}/\epsilon)^{d+d^2}$. Using these in Eq. (18) concludes the proof.

$$7\sqrt{d} \log \frac{|\mathcal{N}_{\epsilon,1}||\mathcal{N}_{\epsilon,2}||\Pi_h|K}{\delta} + 2\sqrt{d}n\zeta + \sqrt{d}n\epsilon$$
$$\leq 8\sqrt{d} \log \frac{|\mathcal{N}_{\epsilon,1}||\mathcal{N}_{\epsilon,2}||\Pi_h|K}{\delta}$$
$$\leq 16d^{\frac{5}{2}} \log \frac{18d^{\frac{3}{2}}K}{\delta} .$$

$\square$

**Lemma 9.** *Fix* $\pi \in \Pi, h \in [H], f \in \mathcal{F}^\pi$. *Let* $\xi_1$ *and* $\xi_2$ *be two solutions for the* $\hat{\xi}_{h,f}$ *in Eq. (9).* *Then* $\|\xi_1 - \xi_2\|_{\Lambda_h} \leq \frac{C_{\text{bonus}}}{H}$. *($C_{\text{bonus}}$ is defined in Algorithm 2)*

*Proof.* Let $\mathcal{D}_h = \{(s_i, a_i, s_i')\}_{i=1}^n$ and denote $\phi_i = \phi(s_i, a_i)$. Let $\xi_{\min} := \operatorname{argmin}_{\xi \in \mathbb{B}^d(\sqrt{d})} \sum_{i=1}^{n} \left(f(s_i') - \phi_i^\top \xi\right)^2$, where $\phi_i := \phi(s_i, a_i)$. By the first-order optimality condition,

$$\sum_{i=1}^{n} \left(f(s_i') - \phi_i^\top \xi_{\min}\right) \left(\phi_i^\top \xi_1 - \phi_i^\top \xi_{\min}\right) \leq 0.$$
(19)

By the fact that $\xi_1$ satisfies Eq. (9),

$$16d^{\frac{5}{2}} \log \frac{18d^{\frac{3}{2}}K}{\delta} \geq \sum_{i=1}^{n} \left(f(s_i') - \phi_i^\top \xi_1\right)^2 - \sum_{i=1}^{n} \left(f(s_i') - \phi_i^\top \xi_{\min}\right)^2$$
$$= 2\sum_{i=1}^{n} \left(f(s_i') - \phi_i^\top \xi_{\min}\right) \left(\phi_i^\top \xi_{\min} - \phi_i^\top \xi_1\right) + \sum_{i=1}^{n} \left(\phi_i^\top (\xi_1 - \xi_{\min})\right)^2$$
$$\geq \sum_{i=1}^{n} \left(\phi_i^\top (\xi_1 - \xi_{\min})\right)^2 \qquad \text{(using Eq. (19))}$$
$$= \|\xi_1 - \xi_{\min}\|_{\Lambda_h}^2 - \|\xi_1 - \xi_{\min}\|_2^2 \qquad \text{(by the definition of } \Lambda_h)$$
$$\geq \|\xi_1 - \xi_{\min}\|_{\Lambda_h}^2 - 4d,$$

which gives $\|\xi_1 - \xi_{\min}\|_{\Lambda_h}^2 \leq \frac{C_{\text{bonus}}^2}{4H^2}$ (recall $C_{\text{bonus}} = 10d^{\frac{5}{4}}H\sqrt{\log \frac{18d^{\frac{3}{2}}K}{\delta}}$. Similarly, $\|\xi_2 - \xi_{\min}\|_{\Lambda_h}^2 \leq \frac{C_{\text{bonus}}^2}{4H^2}$. Combining them proves the lemma. $\square$

*Proof of Lemma 6.*

$$\sum_{s'\in\mathcal{X}_{h+1}}(\hat{\mu}^\pi(s')-\mu^\pi(s'))f(s')$$

$$\leq \sum_{s\in\mathcal{X}_h}\sum_{a\in\mathcal{A}}\hat{\mu}^\pi(s,a)\,\mathrm{clip}\left[\phi(s,a)^\top\hat{\xi}_{h,f}\right] - \sum_{s\in\mathcal{X}_h}\sum_{a\in\mathcal{A}}\mu^\pi(s,a)\,\mathrm{clip}\left[\phi(s,a)^\top\xi_{h,f}^\star\right] + 2\zeta$$

(by Eq. (8) and the same calculation as Eq. (17))

$$= \sum_{s\in\mathcal{X}_h}\sum_{a\in\mathcal{A}}\mu^\pi(s,a)\left(\mathrm{clip}\left[\phi(s,a)^\top\hat{\xi}_{h,f}\right] - \mathrm{clip}\left[\phi(s,a)^\top\xi_{h,f}^\star\right]\right)$$

$$+ \sum_{s\in\mathcal{X}_h}\sum_{a\in\mathcal{A}}(\hat{\mu}^\pi(s,a)-\mu^\pi(s,a))\,\mathrm{clip}\left[\phi(s,a)^\top\hat{\xi}_{h,f}\right] + 2\zeta$$

$$\leq \sum_{s\in\mathcal{X}_h}\sum_{a\in\mathcal{A}}\mu^\pi(s,a)\|\phi(s,a)\|_{\Lambda_h^{-1}}\|\hat{\xi}_{h,f}-\xi_{h,f}^\star\|_{\Lambda_h} + \sum_{s\in\mathcal{X}_h}(\hat{\mu}^\pi(s)-\mu^\pi(s))\tilde{f}(s) + 2\zeta$$

$$\leq \frac{C_{\text{bonus}}}{H}\times\sum_{s\in\mathcal{X}_h}\sum_{a\in\mathcal{A}}\mu^\pi(s,a)\|\phi(s,a)\|_{\Lambda_h^{-1}} + \sum_{s\in\mathcal{X}_h}(\hat{\mu}^\pi(s)-\mu^\pi(s))\tilde{f}(s) + 2\zeta \quad \text{(by Lemma 9)}$$

where $\tilde{f}(s) := \sum_{a\in\mathcal{A}}\pi(a|s)\,\mathrm{clip}\left[\phi(s,a)^\top\hat{\xi}_{h,f}\right]$, which again belongs to $\mathcal{F}^\pi$. Recursively applying the inequality proves the first inequality in the lemma. To obtain the second inequality in the lemma, with slightly different decomposition in the second step above, we get

$$\sum_{s\in\mathcal{X}_h}\sum_{a\in\mathcal{A}}\hat{\mu}^\pi(s,a)\left(\mathrm{clip}\left[\phi(s,a)^\top\hat{\xi}_{h,f}\right] - \mathrm{clip}\left[\phi(s,a)^\top\xi_{h,f}^\star\right]\right)$$

$$+ \sum_{s\in\mathcal{X}_h}\sum_{a\in\mathcal{A}}(\hat{\mu}^\pi(s,a)-\mu^\pi(s,a))\,\mathrm{clip}\left[\phi(s,a)^\top\xi_{h,f}^\star\right] + 2\zeta$$

$$\leq \sum_{s\in\mathcal{X}_h}\sum_{a\in\mathcal{A}}\hat{\mu}^\pi(s,a)\|\phi(s,a)\|_{\Lambda_h^{-1}}\|\hat{\xi}_{h,f}-\xi_{h,f}^\star\|_{\Lambda_h} + \sum_{s\in\mathcal{X}_h}(\hat{\mu}^\pi(s)-\mu^\pi(s))\tilde{f}'(s) + 2\zeta$$

$$\leq \frac{C_{\text{bonus}}}{H}\times\sum_{s\in\mathcal{X}_h}\sum_{a\in\mathcal{A}}\hat{\mu}^\pi(s,a)\|\phi(s,a)\|_{\Lambda_h^{-1}} + \sum_{s\in\mathcal{X}_h}(\hat{\mu}^\pi(s)-\mu^\pi(s))\tilde{f}'(s) + 2\zeta$$

where $\tilde{f}'(s) := \sum_{a\in\mathcal{A}}\pi(a|s)\,\mathrm{clip}\left[\phi(s,a)^\top\xi_{h,f}^\star\right]$. Following the same argument proves the second inequality.

$\square$

### B.3   REGRET ANALYSIS

$$\mathbb{E}\left[\mathcal{R}_K\right]$$

$$= \mathbb{E}\left[\sum_{k=1}^K\sum_{h=1}^H\sum_{\pi\in\Pi}q_k'(\pi)(\phi_h^\pi)^\top\theta_{k,h} - \sum_{k=1}^K\sum_{h=1}^H(\phi_h^{\pi^\star})^\top\theta_{k,h}\right]$$

$$= \mathbb{E}\left[\sum_{k=1}^K\sum_{h=1}^H\sum_{\pi\in\Pi}q_k(\pi)(\phi_h^\pi)^\top\theta_{k,h} - \sum_{k=1}^K\sum_{h=1}^H(\phi_h^{\pi^\star})^\top\theta_{k,h} + \underbrace{\sum_{k=1}^K\sum_{h=1}^H(q_k'(\pi)-q_k(\pi))(\phi_h^\pi)^\top\theta_{k,h}}_{\leq\eta HK}\right]$$

$$\leq \mathbb{E}\left[\sum_{k=1}^K\sum_{\pi\in\Pi}q_k(\pi)(\hat{\phi}_k^\pi)^\top\hat{\theta}_k - \sum_{k=1}^K(\hat{\phi}_k^{\pi^\star})^\top\hat{\theta}_k + \underbrace{\sum_{k=1}^K\sum_{h=1}^H\sum_{\pi\in\Pi}q_k(\pi)\left((\phi_h^\pi-\phi_h^{\pi^\star})^\top\theta_{k,h}-(\hat{\phi}_{k,h}^\pi-\hat{\phi}_{k,h}^{\pi^\star})^\top\hat{\theta}_{k,h}\right)}_{\textbf{bias}}\right] + \eta HK$$

$$= \mathbb{E}\left[\underbrace{\sum_{k=1}^K\sum_{\pi\in\Pi}q_k(\pi)\left((\hat{\phi}_k^\pi)^\top\hat{\theta}_k - b_k^\pi\right) - \sum_{k=1}^K\left((\hat{\phi}_k^{\pi^\star})^\top\hat{\theta}_k - b_k^{\pi^\star}\right)}_{\textbf{ftrl}} + \underbrace{\sum_{k=1}^K\sum_{\pi\in\Pi}q_k(\pi)b_k^\pi - \sum_{k=1}^K b_k^{\pi^\star}}_{\textbf{bonus}} + \textbf{bias}\right] + \eta HK$$

We bound the terms individually in Lemma 10, Lemma 11 and Lemma 12. The potentially unbounded bias term is offset by a negative contribution in the bonus term.

### B.3.1 BOUNDING THE BIAS

**Lemma 10.**

$$\mathbf{bias} \leq \mathbb{E}\Bigg[C_{\mathrm{bonus}} \sum_{k=1}^{K}\sum_{h=1}^{H}\sum_{s\in\mathcal{X}_h}\sum_{a\in\mathcal{A}} \hat{\mu}_k^{\pi^\star}(s,a)\|\phi(s,a)\|_{\Lambda_{k,h}^{-1}} + \eta\sum_{k=1}^{K}\|\hat{\phi}_k^{\pi^\star}\|_{M_k^{-1}}^2\Bigg]$$
$$+ \widetilde{\mathcal{O}}\left(\frac{d^{\frac{9}{2}}H^3}{\eta} + \eta dHK + d^3 H^3\sqrt{K}\right).$$

*Proof.* The bias of any policy $\pi$ at episode $k$ and stage $h$ can be calculated as the following:

$$(\phi_h^\pi)^\top\theta_{k,h} - (\hat{\phi}_{k,h}^\pi)^\top\mathbb{E}[\hat{\theta}_{k,h}] \leq \underbrace{\left|(\phi_h^\pi - \hat{\phi}_{k,h}^\pi)^\top\theta_{k,h}\right|}_{\mathbf{bias}_{k,h,1}^\pi} + \underbrace{\left|(\hat{\phi}_{k,h}^\pi)^\top(\theta_{k,h} - \mathbb{E}[\hat{\theta}_{k,h}])\right|}_{\mathbf{bias}_{k,h,2}^\pi}.$$

Set

$$f(s) = \sum_{a\in\mathcal{A}}\pi(a|s)\phi(s,a)^\top\theta_{k,h} = \sum_{a\in\mathcal{A}}\pi(a|s)\,\mathrm{clip}\left[\phi(s,a)^\top\theta_{k,h}\right]\in\mathcal{F}_1^\pi,\quad (|\phi(s,a)^\top\theta_{k,h}|\leq 1)$$

then the first term is by Lemma 6

$$\mathbf{bias}_{k,h,1}^\pi = \left|\sum_{s\in\mathcal{X}_h}(\mu^\pi(s) - \hat{\mu}_k^\pi(s))f(s)\right| \leq \frac{C_{\mathrm{bonus}}}{H}\times\sum_{h'<h}\sum_{s\in\mathcal{X}_{h'}}\sum_{a\in\mathcal{A}}\hat{\mu}_k^\pi(s,a)\|\phi(s,a)\|_{\Lambda_{k,h'}^{-1}} + 2\zeta H.$$

Define $M_{k,h} = \sum_{\pi\in\Pi}q_k'(\pi)\hat{\phi}_{k,h}^\pi(\hat{\phi}_{k,h}^\pi)^\top$. Then the second term is

$$\mathbf{bias}_{k,h,2}^\pi \leq \|\hat{\phi}_{k,h}^\pi\|_{M_{k,h}^{-1}}\left\|\theta_{k,h} - M_{k,h}^{-1}\sum_{\pi'}q_k'(\pi')\hat{\phi}_{k,h}^{\pi'}(\phi_h^{\pi'})^\top\theta_{k,h}\right\|_{M_{k,h}}$$

$$= \|\hat{\phi}_{k,h}^\pi\|_{M_{k,h}^{-1}}\left\|M_{k,h}^{-1}\sum_{\pi'}q_k'(\pi')\hat{\phi}_{k,h}^{\pi'}(\hat{\phi}_{k,h}^{\pi'} - \phi_h^{\pi'})^\top\theta_{k,h}\right\|_{M_{k,h}}$$

$$= \|\hat{\phi}_{k,h}^\pi\|_{M_{k,h}^{-1}}\left\|\sum_{\pi'}q_k'(\pi')\hat{\phi}_{k,h}^{\pi'}(\hat{\phi}_{k,h}^{\pi'} - \phi_h^{\pi'})^\top\theta_{k,h}\right\|_{M_{k,h}^{-1}}$$

$$\leq \eta\|\hat{\phi}_{k,h}^\pi\|_{M_{k,h}^{-1}}^2 + \frac{1}{\eta}\left\|\sum_{\pi'}q_k'(\pi')\hat{\phi}_{k,h}^{\pi'}(\hat{\phi}_{k,h}^{\pi'} - \phi_h^{\pi'})^\top\theta_{k,h}\right\|_{M_{k,h}^{-1}}^2$$

$$\leq \eta\|\hat{\phi}_{k,h}^\pi\|_{M_{k,h}^{-1}}^2 + \frac{1}{\eta}\left(\sum_{\pi'}q_k'(\pi')\left\|\hat{\phi}_{k,h}^{\pi'}\right\|_{M_{k,h}^{-1}}^2\right)\left(\sum_{\pi'}q_k'(\pi')((\hat{\phi}_{k,h}^{\pi'} - \phi_h^{\pi'})^\top\theta_{k,h})^2\right)$$
$$\text{(by Lemma 47)}$$

$$\leq \eta\|\hat{\phi}_{k,h}^\pi\|_{M_{k,h}^{-1}}^2 + \frac{d}{\eta}\sum_{\pi'}q_k'(\pi')\left(\widetilde{\mathcal{O}}(d^{\frac{5}{4}})\times\sum_{h'<h}\sum_{s\in\mathcal{X}_{h'}}\sum_{a\in\mathcal{A}}\mu^{\pi'}(s,a)\|\phi(s,a)\|_{\Lambda_{k,h'}^{-1}} + 2\zeta H\right)^2$$
$$\text{(by Lemma 6)}$$

$$\leq \eta\|\hat{\phi}_{k,h}^\pi\|_{M_{k,h}^{-1}}^2 + \frac{\widetilde{\mathcal{O}}(d^{\frac{7}{2}})}{\eta}\times\sum_{\pi'}q_k'(\pi')\left(\sum_{h'<h}\sum_{s\in\mathcal{X}_{h'}}\sum_{a\in\mathcal{A}}\mu^{\pi'}(s,a)\right)\left(\sum_{h'<h}\sum_{s\in\mathcal{X}_{h'}}\sum_{a\in\mathcal{A}}\mu^{\pi'}(s,a)\|\phi(s,a)\|_{\Lambda_{k,h'}^{-1}}^2\right)$$
$$\text{(Cauchy-Schwarz)}$$

$$+ \mathcal{O}\left(\frac{d\zeta^2 H^2}{\eta}\right)$$

$$\leq \eta\|\hat{\phi}_{k,h}^{\pi}\|_{M_{k,h}^{-1}}^2 + \frac{\widetilde{\mathcal{O}}(d^{\frac{7}{2}}H)}{\eta} \sum_{h'<h} \beta_{k,h'} + \mathcal{O}\left(\frac{d\zeta^2 H^2}{\eta}\right)$$

where $\beta_{k,h} = \sum_{\pi} \sum_{s\in\mathcal{S}_h, a\in\mathcal{A}} q_k'(\pi)\mu^{\pi}(s,a)\|\phi(s,a)\|_{\Lambda_{k,h}^{-1}}^2$. We have

$$\mathbb{E}\left[\sum_{k=1}^{K}\sum_{h=1}^{H}\beta_{k,h}\right] = \mathbb{E}\left[\sum_{k=1}^{K}\sum_{h=1}^{H}\mathbb{E}\left[\|\phi(s_{k,h}, a_{k,h})\|_{\Lambda_{k,h}^{-1}}^2 \mid \mathcal{D}_{k-1}\right]\right]$$

$$= \mathbb{E}\left[\sum_{k=1}^{K}\sum_{h=1}^{H}\|\phi(s_{k,h}, a_{k,h})\|_{\Lambda_{k,h}^{-1}}^2\right] \leq \mathcal{O}(dH\log(K)).$$

Thus, for any $\pi$,

$$\mathbb{E}\left[\sum_{k=1}^{K}\sum_{h=1}^{H}\mathbf{bias}_{k,h,2}^{\pi}\right] = \mathbb{E}\left[\sum_{k=1}^{K}\sum_{h=1}^{H}\eta\|\hat{\phi}_{k,h}^{\pi}\|_{M_{k,h}^{-1}}^2 + \frac{\widetilde{\mathcal{O}}(d^{\frac{7}{2}}H)}{\eta} \times \sum_{k=1}^{K}\sum_{h=1}^{H}\sum_{h'<h}\beta_{k,h'}\right] + \mathcal{O}\left(\frac{d\zeta^2 H^3 K}{\eta}\right)$$

$$\leq \mathbb{E}\left[\sum_{k=1}^{K}\eta\|\hat{\phi}_k^{\pi}\|_{M_k^{-1}}^2\right] + \frac{\widetilde{\mathcal{O}}(d^{\frac{9}{2}}H^3)}{\eta}. \qquad (\zeta = \tfrac{d}{K})$$

Overall,

$$\mathbf{bias} \leq \mathbb{E}\left[\sum_{k=1}^{K}\sum_{h=1}^{H}\left(\mathbf{bias}_{k,h,1}^{\pi^{\star}} + \sum_{\pi}q_k(\pi)\mathbf{bias}_{k,h,1}^{\pi}\right) + \sum_{k=1}^{K}\sum_{h=1}^{H}\left(\mathbf{bias}_{k,h,2}^{\pi^{\star}} + \sum_{\pi}q_k(\pi)\mathbf{bias}_{k,h,2}^{\pi}\right)\right]$$

$$\leq \mathbb{E}\left[\sum_{k=1}^{K}\sum_{h=1}^{H}\left(\frac{C_{\text{bonus}}}{H} \times \sum_{h'<h}\sum_{s\in\mathcal{X}_{h'}}\sum_{a\in\mathcal{A}}\left(\hat{\mu}_k^{\pi^{\star}}(s,a) + \sum_{\pi}q_k(\pi)\hat{\mu}_k^{\pi}(s,a)\right)\|\phi(s,a)\|_{\Lambda_{k,h'}^{-1}}\right)\right.$$

$$\left. + \sum_{k=1}^{K}\left(\eta\|\hat{\phi}_k^{\pi^{\star}}\|_{M_k^{-1}}^2 + \eta\sum_{\pi}q_k(\pi)\|\hat{\phi}_k^{\pi}\|_{M_k^{-1}}^2\right)\right] + \frac{\widetilde{\mathcal{O}}(d^{\frac{9}{2}}H^3)}{\eta}$$

$$\leq \mathbb{E}\left[C_{\text{bonus}}\sum_{k=1}^{K}\sum_{h=1}^{H}\sum_{s\in\mathcal{X}_h}\sum_{a\in\mathcal{A}}\left(\hat{\mu}_k^{\pi^{\star}}(s,a) + 2\sum_{\pi}q_k'(\pi)\hat{\mu}_k^{\pi}(s,a)\right)\|\phi(s,a)\|_{\Lambda_{k,h}^{-1}}\right.$$

$$\left. + \sum_{k=1}^{K}\left(\eta\|\hat{\phi}_k^{\pi^{\star}}\|_{M_k^{-1}}^2 + 2\eta\sum_{\pi}q_k'(\pi)\|\hat{\phi}_k^{\pi}\|_{M_k^{-1}}^2\right)\right] + \frac{\widetilde{\mathcal{O}}(d^{\frac{9}{2}}H^3)}{\eta}$$

$$\leq \mathbb{E}\left[C_{\text{bonus}}\sum_{k=1}^{K}\sum_{h=1}^{H}\sum_{s\in\mathcal{X}_h}\sum_{a\in\mathcal{A}}\left(\hat{\mu}_k^{\pi^{\star}}(s,a) + 2\sum_{\pi}q_k'(\pi)\mu^{\pi}(s,a)\right)\|\phi(s,a)\|_{\Lambda_{k,h}^{-1}}\right.$$

$$+ 2C_{\text{bonus}}\sum_{k=1}^{K}\sum_{h=1}^{H}\sum_{s\in\mathcal{X}_h}\sum_{a\in\mathcal{A}}\sum_{\pi}q_k'(\pi)(\hat{\mu}_k^{\pi}(s,a) - \mu^{\pi}(s,a))\|\phi(s,a)\|_{\Lambda_{k,h}^{-1}}$$

$$\left. + \eta\sum_{k=1}^{K}\|\hat{\phi}_k^{\pi^{\star}}\|_{M_k^{-1}}^2 + 2\eta dHK\right] + \frac{\widetilde{\mathcal{O}}(d^{\frac{9}{2}}H^3)}{\eta}$$

$$\leq \mathbb{E}\left[C_{\text{bonus}}\sum_{k=1}^{K}\sum_{h=1}^{H}\sum_{s\in\mathcal{X}_h}\sum_{a\in\mathcal{A}}\hat{\mu}_k^{\pi^{\star}}(s,a)\|\phi(s,a)\|_{\Lambda_{k,h}^{-1}} + \widetilde{\mathcal{O}}(C_{\text{bonus}}H\sqrt{dK})\right. \qquad (*)$$

$$+ 2C_{\text{bonus}}\sum_{k=1}^{K}\sum_{h=1}^{H}\sum_{\pi}q_k'(\pi)\left(\frac{C_{\text{bonus}}}{H}\sum_{h'<h}\sum_{s\in\mathcal{X}_{h'}}\sum_{a\in\mathcal{A}}\mu^{\pi}(s,a)\|\phi(s,a)\|_{\Lambda_{k,h'}^{-1}}\right)$$

(by Lemma 6)

$$\left. + \eta\sum_{k=1}^{K}\|\hat{\phi}_k^{\pi^{\star}}\|_{M_k^{-1}}^2 + 2\eta dHK\right] + \frac{\widetilde{\mathcal{O}}(d^{\frac{9}{2}}H^3)}{\eta}$$

$$\leq \mathbb{E}\left[ C_{\text{bonus}} \sum_{k=1}^{K} \sum_{h=1}^{H} \sum_{s\in\mathcal{X}_h} \sum_{a\in\mathcal{A}} \hat{\mu}_k^{\pi^\star}(s,a)\|\phi(s,a)\|_{\Lambda_{k,h}^{-1}} + \eta \sum_{k=1}^{K} \|\hat{\phi}_k^{\pi^\star}\|_{M_k^{-1}}^2 \right.$$

$$\left. + 2C_{\text{bonus}}^2 \sum_{k=1}^{K} \sum_{h=1}^{H} \sum_{s\in\mathcal{X}_h} \sum_{a\in\mathcal{A}} \sum_{\pi} q_k'(\pi)\mu^\pi(s,a)\|\phi(s,a)\|_{\Lambda_{k,h}^{-1}} \right]$$

$$+ \widetilde{\mathcal{O}}\left( \frac{d^{\frac{9}{2}}H^3}{\eta} + \eta dHK + C_{\text{bonus}}H\sqrt{dK} \right)$$

$$\leq \mathbb{E}\left[ C_{\text{bonus}} \sum_{k=1}^{K} \sum_{h=1}^{H} \sum_{s\in\mathcal{X}_h} \sum_{a\in\mathcal{A}} \hat{\mu}_k^{\pi^\star}(s,a)\|\phi(s,a)\|_{\Lambda_{k,h}^{-1}} + \eta \sum_{k=1}^{K} \|\hat{\phi}_k^{\pi^\star}\|_{M_k^{-1}}^2 \right]$$

$$+ \widetilde{\mathcal{O}}\left( \frac{d^{\frac{9}{2}}H^3}{\eta} + \eta dHK + C_{\text{bonus}}H\sqrt{dK} + C_{\text{bonus}}^2 H\sqrt{dK} \right) \tag{*}$$

where in the two (*) places we use

$$\mathbb{E}\left[ \sum_{k=1}^{K} \sum_{h=1}^{H} \sum_{s\in\mathcal{X}_h} \sum_{a\in\mathcal{A}} \sum_{\pi} q_k'(\pi)\mu^\pi(s,a)\|\phi(s,a)\|_{\Lambda_{k,h}^{-1}} \right]$$

$$\leq \mathbb{E}\left[ \sqrt{\sum_{k=1}^{K} \sum_{h=1}^{H} \sum_{s\in\mathcal{X}_h} \sum_{a\in\mathcal{A}} \sum_{\pi} q_k'(\pi)\mu^\pi(s,a)} \sqrt{\sum_{k=1}^{K} \sum_{h=1}^{H} \sum_{s\in\mathcal{X}_h} \sum_{a\in\mathcal{A}} \sum_{\pi} q_k'(\pi)\mu^\pi(s,a)\|\phi(s,a)\|_{\Lambda_{k,h}^{-1}}^2} \right]$$

$$\leq \sqrt{HK\mathbb{E}\left[ \sum_{k=1}^{K} \sum_{h=1}^{H} \beta_{k,h} \right]}$$

$$\leq \widetilde{\mathcal{O}}(H\sqrt{dK}).$$

Finally, plugging in the definition of $C_{\text{bonus}} = \widetilde{\mathcal{O}}(d^{\frac{5}{4}}H)$ gives the desired bound.

$\square$

### B.3.2 BOUNDING THE FTRL REGRET

**Lemma 11.**

$$\mathbf{ftrl} \leq \widetilde{\mathcal{O}}\left( \eta d^2 H^4 K + \frac{\eta^3 H^2}{\gamma^2}K + \gamma HK \right).$$

*Proof.* The magnitude of the loss is bounded by

$$|\hat{\phi}_k^{\pi^\top}\hat{\theta}_k - b_k^\pi| \leq \left| \hat{\phi}_k^{\pi^\top} M_k^{-1}\hat{\phi}_k^{\pi_k} L_k \right| + C_{\text{bonus}} \sum_{h=1}^{H} \sum_{s\in\mathcal{X}_h} \sum_{a\in\mathcal{A}} \hat{\mu}_k^\pi(s,a)\|\phi(s,a)\|_{\Lambda_{k,h}^{-1}} + \eta\|\hat{\phi}_k^\pi\|_{M_k^{-1}}^2$$

$$\leq \left\|\hat{\phi}^\pi\right\|_{M_k^{-1}} \left\|\hat{\phi}_k^{\pi_k}\right\|_{M_k^{-1}} H + C_{\text{bonus}}H + \frac{\eta dH}{\gamma}$$

$$\leq \frac{dH}{\gamma} + C_{\text{bonus}}H + \frac{\eta dH}{\gamma} \leq \frac{2dH}{\gamma} + C_{\text{bonus}}H.$$

If $\eta \leq \frac{1}{\frac{4dH}{\gamma} + 2C_{\text{bonus}}H}$, then we have $\eta|\hat{\phi}_k^{\pi^\top}\hat{\theta}_k - b_k^\pi| \leq \frac{1}{2}$ and we can use the standard FTRL regret bound of exponential weights (Lattimore and Szepesvári, 2020, Equation (27.2, 27.3)):

$$\mathbf{ftrl} \leq \underbrace{\gamma KH}_{\text{John's exploration}} + \frac{\ln|\Pi|}{\eta} + \eta \sum_{k=1}^{K} \mathbb{E}\left[ \mathbb{E}_{\pi_k\sim q_k'}\left[ \sum_{\pi\in\Pi} q_k(\pi)(2(\hat{\phi}_k^{\pi^\top}\hat{\theta}_k)^2 + 2(b_k^\pi)^2) \right] \right].$$

Since $M_k = \mathbb{E}_{\pi \sim q'_k}[\hat{\phi}_k^\pi \hat{\phi}_k^{\pi\top}]$, we have $M_k^{-1} \preceq \frac{1}{1-\gamma}\left(\mathbb{E}_{\pi \sim q_k}[\hat{\phi}_k^\pi \hat{\phi}_k^{\pi\top}]\right)^{-1}$, and thus

$$\mathbb{E}_{\pi_k \sim q'_k}\left[\sum_{\pi \in \Pi} q_k(\pi)(\hat{\phi}_k^{\pi\top} M_k^{-1} \hat{\phi}_k^{\pi_k} L_k)^2\right] \leq H^2 \frac{1}{(1-\gamma)^2} \operatorname{Tr}\left(M_k M_k^{-1} M_k M_k^{-1}\right) = \mathcal{O}(dH^3).$$

For the final term, we have

$$\sum_{k=1}^K \eta \sum_\pi q_k(\pi)(b_k^\pi)^2 \leq \eta C_{\text{bonus}}^2 H^2 K + \frac{\eta^3 d^2 H^2}{\gamma^2} K = \widetilde{\mathcal{O}}\left(\eta d^{\frac{5}{2}} H^4 K + \frac{\eta^3 d^2 H^2}{\gamma^2} K\right).$$

$\square$

### B.3.3 Bounding the bonus

**Lemma 12.**

$$\mathbf{bonus} \leq -\mathbb{E}\left[\sum_{k=1}^K \eta \|\hat{\phi}_k^{\pi^\star}\|_{M_k^{-1}}^2 + C_{\text{bonus}} \sum_{h=1}^H \sum_{s \in \mathcal{X}_h} \sum_{a \in \mathcal{A}} \hat{\mu}_k^{\pi^\star}(s,a) \|\phi(s,a)\|_{\Lambda_h^{-1}}\right]$$
$$+ \widetilde{\mathcal{O}}\left(\frac{d^{\frac{9}{2}} H^3}{\eta} + \eta dHK + d^3 H^3 \sqrt{K}\right).$$

*Proof.*

$$\mathbf{bonus} \leq \mathbb{E}\left[\sum_{k=1}^K \eta \sum_\pi q'_k(\pi) \|\hat{\phi}_k^\pi\|_{M_k^{-1}}^2 + C_{\text{bonus}} \sum_{h=1}^H \sum_{s \in \mathcal{X}_h} \sum_{a \in \mathcal{A}} \sum_\pi q'_k(\pi) \hat{\mu}_k^\pi(s,a) \|\phi(s,a)\|_{\Lambda_h^{-1}}\right.$$
$$\left. - \eta \|\hat{\phi}_k^{\pi^\star}\|_{M_k^{-1}}^2 - C_{\text{bonus}} \sum_{h=1}^H \sum_{s \in \mathcal{X}_h} \sum_{a \in \mathcal{A}} \hat{\mu}_k^{\pi^\star}(s,a) \|\phi(s,a)\|_{\Lambda_h^{-1}}\right]$$

The first and the second term above have been handled in the proof of Lemma 10. Following the analysis there, we can bound their sum by $\widetilde{\mathcal{O}}\left(\frac{d^{\frac{9}{2}} H^3}{\eta} + \eta dHK + d^3 H^3 \sqrt{K}\right)$. $\square$

### B.3.4 Finishing up

*Proof of Theorem 7.* Combining the bounds in Lemma 10, Lemma 11, and Lemma 12, we bound the regret as

$$\mathbb{E}[\mathcal{R}_K] \leq \widetilde{\mathcal{O}}\left(\eta d^{\frac{5}{2}} H^4 K + \frac{\eta^3 d^2 H^2}{\gamma^2} K + \gamma HK + \frac{d^{\frac{9}{2}} H^3}{\eta} + d^3 H^3 \sqrt{K}\right)$$
$$= \widetilde{\mathcal{O}}\left(\eta d^{\frac{5}{2}} H^4 K + \frac{d^{\frac{9}{2}} H^3}{\eta} + d^3 H^3 \sqrt{K}\right) \qquad (\gamma = \Theta(\eta dH))$$
$$= \widetilde{\mathcal{O}}(d^{\frac{7}{2}} H^{\frac{7}{2}} \sqrt{K}). \qquad (\eta = \Theta(d/\sqrt{HK}))$$

$\square$

## C   INITIAL PURE EXPLORATION PHASE

---

**Algorithm 5** Initial Pure Exploration (Algorithm 2 of Sherman et al. (2023a))

---

**input:** $\delta, \rho, \epsilon_{\text{cov}}$
Set $m = \lceil \log \frac{1}{\epsilon_{\text{cov}}} \rceil$
Set $\forall i \in [m], \ \rho_i = \rho$
**for** $h = H, \ldots, 1$ **do**
    $\left\{ \widetilde{\mathcal{X}}_{h,i}, \widetilde{\mathcal{D}}_{h,i}, \widetilde{\Lambda}_{h,i} \right\}_{i=1}^{m} \leftarrow \text{CoverTraj}(h, \frac{\delta}{H}, \{\rho_i\}_{i=1}^{m}, m)$
    $\mathcal{D}_h \leftarrow \bigcup_i \widetilde{\mathcal{D}}_{h,i}$
    $\Lambda_h \leftarrow I + \sum_{(s,a,s') \in \mathcal{D}_h} \phi(s,a)\phi(s,a)^\top$
    $\mathcal{Z}_h \leftarrow \left\{ s \in \mathcal{S}_h : \ \forall a \in \mathcal{A}, \ \|\phi(s,a)\|_{\Lambda_h^{-1}} \le \rho \right\}$
**end for**
**return** $(\mathcal{D}_h, \mathcal{Z}_h)_{h=1}^{H}$

---

**Theorem 13** (Theorem 2 in Sherman et al. (2023a)). *The* CoverTraj *algorithm (Wagenmaker et al., 2022b, Algorithm 4) when instantiated with* Force *(Wagenmaker et al., 2022a, Algorithm 1) enjoys the following guarantee for linear MDPs. Given a sequence of tolerance parameters $\rho_1, \ldots, \rho_m > 0$ and $h \in [H]$, the algorithm interacts with the environment for $T$ steps, where*

$$T \le T_{\max} \triangleq C \sum_{i=1}^{m} 2^i \max \left\{ \frac{d}{\rho_i^2} \log \frac{2^i}{\rho_i^2}, d^4 H^3 m^3 \log^{7/2} \frac{1}{\delta} \right\}, \quad C > 0 \ \text{is a logarithmic term,}$$

*and outputs $\left\{ \widetilde{\mathcal{X}}_{h,i}, \widetilde{\mathcal{D}}_{h,i}, \widetilde{\Lambda}_{h,i} \right\}_{i=1}^{m}$ such that $\left\{ \widetilde{\mathcal{X}}_{h,i} \right\}_{i=1}^{m+1}$ forms a partition for the unit Euclidean ball, $\widetilde{\Lambda}_{h,i} = I + \sum_{(s,a,s') \in \widetilde{\mathcal{D}}_{h,i}} \phi(s,a)\phi(s,a)^\top$, and with probability $1 - \delta$, it holds that:*

$$\forall i \in [m], \quad \phi^\top \widetilde{\Lambda}_{h,i}^{-1} \phi \le \rho_i^2, \quad \forall \phi \in \widetilde{\mathcal{X}}_{h,i};$$

$$\text{and} \ \ \forall i \in [m+1], \quad \sup_\pi \left\{ \sum_{s \in \mathcal{S}_h} \sum_{a \in \mathcal{A}} \mathbb{I}\left\{ \phi(s,a) \in \widetilde{\mathcal{X}}_{h,i} \right\} \mu^\pi(s,a) \right\} \le 2^{-i+1}.$$

**Lemma 14** (Lemma 15 in Sherman et al. (2023a)). *Assume $h \in [H], \epsilon_{\text{cov}} > 0, \delta > 0, m = \lceil \log(1/\epsilon_{\text{cov}}) \rceil, \rho_m \ge \cdots \ge \rho_1 > 0$, and let $\left\{ \widetilde{\Lambda}_{h,i} \right\}_{i \in [m]}$ be the covariance matrices returned from* CoverTraj$(h, \frac{\delta}{H}, \{\rho_i\}_{i=1}^{m}, m)$. *Then under the assumption that the event from Theorem 13 holds, we have for any policy $\pi$ and $i \in [m]$:*

$$\sum_{s \in \mathcal{S}_h} \mu^\pi(s) \mathbb{I}\left\{ \exists a \ \text{s.t.} \ \|\phi(s,a)\|_{\widetilde{\Lambda}_{h,i}^{-1}} > \rho_m \right\} \le \epsilon_{\text{cov}}.$$

**Lemma 15.** *For linear MDPs, with inputs $\delta \in (0,1)$, $\rho > 0$, $\epsilon_{\text{cov}} > 0$, Algorithm 5 will terminate in $T = \widetilde{\Theta}\left( \frac{dH/\rho^2 + d^4 H^4}{\epsilon_{\text{cov}}} \text{polylog}\left( \frac{1}{\delta}, \frac{1}{\rho}, \frac{1}{\epsilon_{\text{cov}}}, d, H \right) \right)$ episodes, and output $H$ datasets $\{\mathcal{D}_h\}_{h=1}^{H}$ where $\mathcal{D}_h \subset \mathcal{S}_h \times \mathcal{A} \times \mathcal{S}_{h+1}$ such that with probability $\ge 1 - \delta$,*

$$\forall h, \forall \pi, \quad \sum_{s \in \mathcal{S}_h} \mu^\pi(s) \mathbb{I}\{s \notin \mathcal{Z}_h\} \le \epsilon_{\text{cov}}, \quad \text{where} \ \ \mathcal{Z}_h \triangleq \left\{ s \in \mathcal{S}_h : \ \forall a \in \mathcal{A}, \ \|\phi(s,a)\|_{\Lambda_h^{-1}} \le \rho \right\}$$

*with $\Lambda_h \triangleq I + \sum_{(s,a,s') \in \mathcal{D}_h} \phi(s,a)\phi(s,a)^\top$.*

*Proof of Lemma 15.* Let $T_h$ denote the number of episodes run by CoverTraj, by Theorem 13,

$$\begin{aligned}
T_h &\le C \sum_{i=1}^{m} 2^i \max \left\{ \frac{d}{\rho_i^2} \log \frac{2^i}{\rho_i^2}, d^4 H^3 m^3 \log^{7/2} \frac{1}{\delta} \right\} \\
&\le \widetilde{\mathcal{O}} \left( m 2^m \left( \frac{d}{\rho^2} \log \left( \frac{2^m}{\rho^2} \right) + d^4 H^3 m^3 \log^{7/2} \frac{1}{\delta} \right) \right) \\
&\le \widetilde{\mathcal{O}} \left( \frac{d/\rho^2 + d^4 H^3}{\epsilon_{\text{cov}}} \text{polylog} \left( \frac{1}{\delta}, \frac{1}{\epsilon_{\text{cov}}}, \frac{1}{\rho}, d, H \right) \right).
\end{aligned}$$

Given that Algorithm 5 executes COVERTRAJ $H$ times, the claim follows. For the claim on the un-reachability of $\mathcal{S}_h \setminus \mathcal{Z}_h$, fix $h \in [H]$, and observe that by Lemma 14, w.p. $1 - \delta/H$, for any $\pi$;

$$\sum_{s \in \mathcal{S}_h} \mu^\pi(s) \mathbb{I}\left\{\exists a \text{ s.t. } \|\phi(s,a)\|_{\Lambda_h^{-1}} > \rho_m\right\} \leq \epsilon_{\text{cov}},$$

where in the inequality we use that $\widetilde{\Lambda}_{h,i} \preceq \Lambda_h$. The proof is complete by a union bound over $h$. $\qquad\square$

## D OMITTED DETAILS IN SECTION 4

We will be using several additional notations in the analysis.

**Definition 16** ($\mu_h^\pi$, $\mu_h^k$, $\mu_h^\star$). *Define $\mu_h^\pi(s) = \mu^\pi(s)\mathbb{I}\{s \in \mathcal{S}_h\}$. By the definition of $\mu^\pi(s)$, we know that $\mu_h^\pi$ is a distribution over $\mathcal{S}$ that is supported on $\mathcal{S}_h$. Define $\mu_h^k = \mu_h^{\pi_k}$ and $\mu_h^\star = \mu_h^{\pi^\star}$.*

**Definition 17** ($T_h^\pi$, $\mathbb{E}_h^\pi$, $\mathbb{E}_h^\star$). *We define $T_h^\pi$ be the distribution over trajectories $\{(s_i, a_i)\}_{i=1}^h$ for the first $h$ steps generated by policy $\pi$ and transition $P$. Then we define*

$$\mathbb{E}_h^\pi[\cdot] = \mathbb{E}_{(s_i,a_i)_{i=1}^{h-1} \sim T_{h-1}^\pi} \mathbb{E}_{s \sim P(\cdot|s_{h-1},a_{h-1})}[\cdot],$$

*where $[\cdot]$ can be a function of $(s_1, a_1, \ldots, s_{h-1}, a_{h-1}, s)$.*

*In the analysis, we will mainly consider the optimal policy $\pi^\star$. For notation simplicity, we write $\mathbb{E}_h^\star[\cdot] = \mathbb{E}_h^{\pi^\star}[\cdot]$.*

**Definition 18** (Good trajectory). *For any trajectory $t = \{(s_h, a_h, s_{h+1})\}_{h=i}^j$ where $1 \leq i \leq j \leq H$, if $s_h \in \mathcal{Z}_h$ for any $h$, then we say $t$ is a good trajectory.*

**Definition 19** ($Q_k$). *Define $Q_k(s,a) = Q^{\pi_k}(s,a;\ell_k)$.*

### D.1 REGRET DECOMPOSITION AND DILATED BONUS LEMMA

**Lemma 20.** *For any trajectory $t = \{(s_h, a_h, s_{h+1})\}_{h=i}^j$ with $1 \leq i \leq j \leq H$ generated by any policy, we have*

$$\Pr(t \text{ is not a good trajectroy}) \leq HK^{-\frac{1}{4}}$$

*Proof.* From Lemma 15, since we choose $\epsilon_{cov} = K^{-\frac{1}{4}}$, for any $h$ and $s_h$ generated by any policy, we have $P(t \notin \mathcal{Z}_h) \leq K^{-\frac{1}{4}}$. By union bound, we have

$$\Pr(t \text{ is not a good trajectory}) = \Pr\left(\bigcup_{i \leq h \leq j} s_h \notin \mathcal{Z}_h\right) \leq HK^{-\frac{1}{4}}$$

$\qquad\square$

In the regret decomposition below, we use the notation $\mathbb{E}_h^\star[\cdot]$ defined in Definition 17 to denote the expectation over trajectories $(s_1, a_1, \ldots, s_{h-1}, a_{h-1}, s_h = s)$ drawn from $\pi^\star$, and use $\mathcal{E}_h$ to denote the event that $\forall h' \leq h, s_{h'} \in \mathcal{Z}_{h'}$. By Lemma 20, we have $\mathbb{E}_h^\star[\mathbb{I}\{\mathcal{E}_h\}] \geq 1 - HK^{-\frac{1}{4}}$ for any $h$. By

performance difference lemma ([Kakade and Langford, 2002](#)), we have

$$\mathbb{E}[\mathcal{R}_K]$$

$$= \mathbb{E}\left[\sum_{k=1}^{K}\sum_{h=1}^{H}\mathbb{E}_{s\sim\mu_h^\star}\left[\langle Q_k(s,\cdot), \pi_k(\cdot|s) - \pi^\star(\cdot|s)\rangle\right]\right]$$

$$= \mathbb{E}\left[\sum_{k=1}^{K}\sum_{h=1}^{H}\mathbb{E}_h^\star\left[\langle Q_k(s,\cdot), \pi_k(\cdot|s) - \pi^\star(\cdot|s)\rangle\right]\right]$$

$$= \mathbb{E}\left[\sum_{k=1}^{K}\sum_{h=1}^{H}\mathbb{E}_h^\star\left[\langle Q_k(s,\cdot), \pi_k(\cdot|s) - \pi^\star(\cdot|s)\rangle\,\mathbb{I}\{\mathcal{E}_h\}\right]\right] + \mathbb{E}\left[\sum_{k=1}^{K}\sum_{h=1}^{H}\mathbb{E}_h^\star\left[\langle Q_k(s,\cdot), \pi_k(\cdot|s) - \pi^\star(\cdot|s)\rangle\,\mathbb{I}\{\overline{\mathcal{E}_h}\}\right]\right]$$

$$\leq \underbrace{\mathbb{E}\left[\sum_{k=1}^{K}\sum_{h=1}^{H}\mathbb{E}_h^\star\left[\langle Q_k(s,\cdot), \pi_k(\cdot|s) - \pi^\star(\cdot|s)\rangle\,\mathbb{I}\{\mathcal{E}_h\}\right]\right]}_{\textbf{reg-term}} + H^3 K^{\frac{3}{4}} \tag{20}$$

where the last step comes from [Lemma 20](#) and $Q_k(s,a) \leq H$ for any $k, h, s, a$.

To handle **reg-term**, we utilize the dilated bonus technique proposed in [Luo et al. (2021)](#). We summarize the technique in [Lemma 21](#), with slight modification to make it align with our settings.

**Lemma 21** (Adaptation of Lemma 3.1 in [Luo et al. (2021)](#)). *Suppose that for some bonus functions* $b_k(s,a)$, $B_k(s,a)$ *and some constants* $f, g$, *we have for all* $s \in \mathcal{S}_h$,

$$B_k(s,a) \geq b_k(s,a) + \left(1 + \frac{1}{H}\right)\mathbb{E}_{s'\sim P(\cdot|s,a)}\mathbb{E}_{a'\sim\pi_k(\cdot|s')}\left[B_k(s',a')\mathbb{I}\{s' \in \mathcal{Z}_{h+1}\}\right] - f, \tag{21}$$

*and suppose that our algorithm guarantees*

$$\mathbb{E}\left[\sum_{k=1}^{K}\sum_{h=1}^{H}\mathbb{E}_h^\star\left[\langle Q_k(s,\cdot) - B_k(s,a), \pi_k(\cdot|s) - \pi^\star(\cdot|s)\rangle\,\mathbb{I}\{\mathcal{E}_h\}\right]\right]$$

$$\leq g + \mathbb{E}\left[\sum_{k=1}^{K}\sum_{h=1}^{H}\mathbb{E}_h^\star\mathbb{E}_{a\sim\pi^\star(\cdot|s)}\left[b_k(s,a)\mathbb{I}\{\mathcal{E}_h\}\right]\right] + \frac{1}{H}\mathbb{E}\left[\sum_{k=1}^{K}\sum_{h=1}^{H}\mathbb{E}_h^\star\mathbb{E}_{a\sim\pi_k(\cdot|s)}\left[B_k(s,a)\mathbb{I}\{\mathcal{E}_h\}\right]\right]. \tag{22}$$

*Then, we have (recall the **reg-term** defined in the proof of [Eq. (20)](#))*

$$\textbf{reg-term} \leq g + fHK + \left(1 + \frac{1}{H}\right)\mathbb{E}\left[\sum_{k=1}^{K}\mathbb{E}_{a\sim\pi_k(\cdot|s_1)}\left[B_k(s_1,a)\right]\right].$$

*Proof.* Notice that for any function $X$ of $(s_1, a_1, \ldots, s_H, a_H)$, it holds that

$$\mathbb{E}_h^\star\mathbb{E}_{a\sim\pi^\star(\cdot|s)}\mathbb{E}_{s'\sim P(\cdot|s,a)}\left[X\mathbb{I}\{\mathcal{E}_h\}\mathbb{I}\{s' \in \mathcal{Z}_{h+1}\}\right] = \mathbb{E}_{h+1}^\star\left[X\mathbb{I}\{\mathcal{E}_{h+1}\}\right]. \tag{23}$$

By the definition of **reg-term**, we have

**reg-term**

$$= \mathbb{E}\left[\sum_{k=1}^{K}\sum_{h=1}^{H}\mathbb{E}_h^\star\left[\langle Q_k(s,\cdot), \pi_k(\cdot|s) - \pi^\star(\cdot|s)\rangle\,\mathbb{I}\{\mathcal{E}_h\}\right]\right]$$

$$\leq g + \mathbb{E}\left[\sum_{k=1}^{K}\sum_{h=1}^{H}\mathbb{E}_h^\star\mathbb{E}_{a\sim\pi^\star(\cdot|s)}\left[b_k(s,a)\mathbb{I}\{\mathcal{E}_h\}\right]\right] + \frac{1}{H}\mathbb{E}\left[\sum_{k=1}^{K}\sum_{h=1}^{H}\mathbb{E}_h^\star\mathbb{E}_{a\sim\pi_k(\cdot|s)}\left[B_k(s,a)\mathbb{I}\{\mathcal{E}_h\}\right]\right]$$

$$+ \mathbb{E}\left[\sum_{k=1}^{K}\sum_{h=1}^{H}\mathbb{E}_h^\star\left[\langle B_k(s,\cdot), \pi_k(\cdot|s)\rangle\,\mathbb{I}\{\mathcal{E}_h\}\right]\right] - \mathbb{E}\left[\sum_{k=1}^{K}\sum_{h=1}^{H}\mathbb{E}_h^\star\left[\langle B_k(s,\cdot), \pi^\star(\cdot|s)\rangle\,\mathbb{I}\{\mathcal{E}_h\}\right]\right]$$

$$\text{(by [Eq. (22)](#))}$$

$$\leq g + fHK + \mathbb{E}\left[\sum_{k=1}^{K}\sum_{h=1}^{H}\mathbb{E}_h^\star\mathbb{E}_{a\sim\pi^\star(\cdot|s)}\left[b_k(s,a)\mathbb{I}\{\mathcal{E}_h\}\right]\right]$$

$$+ \left(1+\frac{1}{H}\right)\mathbb{E}\left[\sum_{k=1}^{K}\sum_{h=1}^{H}\mathbb{E}_h^\star\mathbb{E}_{a\sim\pi_k(\cdot|s)}\left[B_k(s,a)\mathbb{I}\{\mathcal{E}_h\}\right]\right] - \mathbb{E}\left[\sum_{k=1}^{K}\sum_{h=1}^{H}\mathbb{E}_h^\star\mathbb{E}_{a\sim\pi^\star(\cdot|s)}\left[b_k(s,a)\mathbb{I}\{\mathcal{E}_h\}\right]\right]$$

$$- \left(1+\frac{1}{H}\right)\mathbb{E}\left[\sum_{k=1}^{K}\sum_{h=1}^{H}\mathbb{E}_h^\star\mathbb{E}_{a\sim\pi^\star(\cdot|s)}\mathbb{E}_{s'\sim P(\cdot|s,a)}\mathbb{E}_{a'\sim\pi_k(\cdot|s')}\left[B_k(s',a')\mathbb{I}\{\mathcal{E}_h\}\mathbb{I}\{s'\in\mathcal{Z}_{h+1}\}\right]\right]$$

(by Eq. (21))

$$= g + fHK + \left(1+\frac{1}{H}\right)\mathbb{E}\left[\sum_{k=1}^{K}\sum_{h=1}^{H}\mathbb{E}_h^\star\mathbb{E}_{a\sim\pi_k(\cdot|s)}\left[B_k(s,a)\mathbb{I}\{\mathcal{E}_h\}\right]\right]$$

$$- \left(1+\frac{1}{H}\right)\mathbb{E}\left[\sum_{k=1}^{K}\sum_{h=1}^{H}\mathbb{E}_{h+1}^\star\mathbb{E}_{a\sim\pi^\star(\cdot|s)}\left[B_k(s,a)\mathbb{I}\{\mathcal{E}_{h+1}\}\right]\right]$$

(by Eq. (23))

$$= g + fHK + \left(1+\frac{1}{H}\right)\mathbb{E}\left[\sum_{k=1}^{K}\mathbb{E}_1^\star\mathbb{E}_{a\sim\pi_k(\cdot|s)}\left[B_k(s,a)\mathbb{I}\{\mathcal{E}_1\}\right]\right]$$

(telescoping)

$$= g + fHK + \left(1+\frac{1}{H}\right)\mathbb{E}\left[\sum_{k=1}^{K}\mathbb{E}_{a\sim\pi_k(\cdot|s_1)}\left[B_k(s_1,a)\right]\right]. \qquad (\mathcal{S}_1 = \{s_1\} \text{ and } s_1\in\mathcal{Z}_1)$$

□

In the following Appendix D.2 and Appendix D.3, we aim to show that our Algorithm 3 and Algorithm 4 could induce bonus functions $b_k(s,a)$, $B_k(s,a)$ that satisfy the condition of Lemma 21. This allows us to directly apply it and get the desired regret bound in Appendix D.4. Our choices of $B_k(s,a)$ and $b_k(s,a)$ are the following:

---

For $s\in\mathcal{S}_h$, $a\in\mathcal{A}$,

$$b_k(s,a) = \beta\|\phi(s,a)\|_{\widehat{\Sigma}_{k,h}^{-1}}^2 + \left(1-\frac{1}{4H}\right)\alpha\|\phi(s,a)\|_{\Lambda_{k,h}^{-1}}^2 \tag{24}$$

$$B_k(s,a) = b_k(s,a) + \phi(s,a)^\top w_{k,h} \tag{25}$$

where

$$w_{k,h} = \left(1+\frac{1}{H}\right)\sum_{s'\in\mathcal{S}_{h+1}}\psi(s')\widehat{W}_k(s')\mathbb{I}\{s'\in\mathcal{Z}_{h+1}\} \qquad (w_{k,H}\triangleq 0) \tag{26}$$

with the $\widehat{W}_k(s')$ defined in Algorithm 4.

---

## D.2 CONSTRUCTION OF DILATED BONUS (ACHIEVING EQ. (21) USING ALGORITHM 4)

In the linear regression (Line 5) of Algorithm 4, the $\widehat{w}_{k,h}$ is an estimation of $w_{k,h}$ defined in Eq. (26), where for $s'\in\mathcal{S}_{h+1}$,

$$\widehat{W}_k(s') = \mathbb{E}_{a'\sim\pi_k(\cdot|s')}\left[\left[\beta\|\phi(s',a')\|_{\widehat{\Sigma}_{k,h+1}^{-1}}^2 + \alpha\|\phi(s',a')\|_{\Lambda_{k,h+1}^{-1}}^2 + \phi(s',a')^\top\widehat{w}_{k,h+1}\right]^+\right], \tag{27}$$

with $[x]^+$ denoting $\max\{x,0\}$.

The next Lemma 22 is a key lemma that 1) bounds the error between $\widehat{w}_{k,h}$ and $w_{k,h}$, and 2) bounds the magnitude of $\widehat{w}_{k,h}$ and $w_{k,h}$ for all $h\in[H]$.

**Lemma 22.** *Let $C_\iota = 15\sqrt{\log\left(\frac{12dK}{\delta}\right)}$ and suppose that $B_h^{\max} \leq \frac{\alpha}{C_\iota^2 H d^2}$. Then with probability at least $1 - \delta$, the following inequalities hold for all $k \in [K]$, $h \in [H]$, and all $s \in \mathcal{S}_h$:*

$$\|w_{k,h}\|_2 \leq \sqrt{d}B_h^{\max}, \tag{28}$$

$$\left|\phi(s,a)^\top \widehat{w}_{k,h} - \phi(s,a)^\top w_{k,h}\right| \leq C_\iota d B_h^{\max}\|\phi(s,a)\|_{\Lambda_{k,h}^{-1}}, \tag{29}$$

$$|\phi(s,a)^\top \widehat{w}_{k,h}|\mathbb{I}\{s \in \mathcal{Z}_h\} \leq \left(1 + \frac{1}{2H}\right)B_h^{\max}. \tag{30}$$

*Proof.* We use induction to prove these three inequalities. For the base case $h = H$, we have $w_{k,H} = \mathbf{0}$ and $\widehat{w}_{k,H} = \mathbf{0}$, so all three inequalities holds.

Suppose that all three inequalities holds for the case of $h + 1$. Below, we show that that also holds for $h$.

**Showing Eq. (28).** Observe that for any $s' \in \mathcal{S}_{h+1}$,

$$\left(1 + \frac{1}{H}\right)\widehat{W}_k(s')\mathbb{I}\{s' \in \mathcal{Z}_{h+1}\}$$

$$\leq \max_{a' \in \mathcal{A}}\left(1 + \frac{1}{H}\right)\left(\beta\|\phi(s',a')\|_{\widehat{\Sigma}_{k,h+1}^{-1}}^2 + \alpha\|\phi(s',a')\|_{\Lambda_{k,h+1}^{-1}}^2 + |\phi(s',a')^\top \widehat{w}_{k,h+1}|\right)\mathbb{I}\{s' \in \mathcal{Z}_{h+1}\}$$

$$\leq \left(1 + \frac{1}{H}\right)\left(\frac{\beta}{\gamma} + \alpha\rho^2\right) + \left(1 + \frac{1}{H}\right)\left(1 + \frac{1}{2H}\right)B_{h+1}^{\max}$$

$(\|\phi(s',a')\|_{\Lambda_{k,h+1}^{-1}} \leq \rho$ for $s' \in \mathcal{Z}_{h+1}$ by Algorithm 5; using induction hypothesis Eq. (30) for $h + 1$)

$$\leq \left(1 + \frac{1}{H}\right)\frac{1}{2H}B_{h+1}^{\max} + \left(1 + \frac{1}{H}\right)\left(1 + \frac{1}{2H}\right)B_{h+1}^{\max} \qquad \text{(by the definition of } B_{h+1}^{\max})$$

$$\leq \left(1 + \frac{1}{H}\right)^2 B_{h+1}^{\max}$$

$$= B_h^{\max}. \tag{31}$$

Thus,

$$\|w_{k,h}\|_2 = \left\|\left(1 + \frac{1}{H}\right)\sum_{s' \in \mathcal{S}_{h+1}}\psi(s')\widehat{W}_k(s')\mathbb{I}\{s' \in \mathcal{Z}_{h+1}\}\right\|_2 \leq B_h^{\max}\left\|\sum_{s' \in \mathcal{S}_{h+1}}\psi(s')\right\|_2 \leq \sqrt{d}B_h^{\max}$$

where in the last inequality we use the linear MDP assumption (Definition 2).

**Showing Eq. (29).**

$$\left|\phi(s,a)^\top \widehat{w}_{k,h} - \phi(s,a)^\top w_{k,h}\right| \leq \|\phi(s,a)\|_{\Lambda_{k,h}^{-1}}\|\widehat{w}_{k,h} - w_{k,h}\|_{\Lambda_{k,h}}. \tag{32}$$

By Lemma 44 and $\|w_{k,h}\| \leq \sqrt{d}B_h^{\max}$ (which we just proved), it holds that

$$\|\widehat{w}_{k,h} - w_{k,h}\|_{\Lambda_{k,h}}$$

$$\leq \left\|\sum_{(s,a,s') \in \mathcal{D}_{k,h}}\phi(s,a)\left(\left(1 + \frac{1}{H}\right)\widehat{W}_k(s')\mathbb{I}\{s' \in \mathcal{Z}_{h+1}\} - \phi(s,a)^\top w_{k,h}\right)\right\|_{\Lambda_{k,h}^{-1}} + \sqrt{d}B_h^{\max}. \tag{33}$$

By Lemma 43, the first term above can be upper bounded by

$$\sqrt{4(B_h^{\max})^2\left(\frac{d}{2}\log K + \log\frac{\mathcal{N}_\epsilon(\mathcal{V}_h)}{\delta}\right) + 8K^2\epsilon^2}. \tag{34}$$

where $\mathcal{V}_h$ is the function class where $\left(1 + \frac{1}{H}\right)\widehat{W}_k(s')\mathbb{I}\{s' \in \mathcal{Z}_{h+1}\}$ lies, and $\mathcal{N}_\epsilon(\mathcal{V}_h)$ is its $\epsilon$-covering number. By the form of $\widehat{W}_k(s')$ given in Eq. (27), $\mathcal{V}_h$ can be chosen as the that defined in Definition 39. Then by Lemma 42 with $\epsilon = \frac{1}{K}$ and $\frac{\beta}{\gamma} + 2\alpha \leq K^2$, we have

$$\log\left(\mathcal{N}_\epsilon\left(\mathcal{V}_h\right)\right) \leq 4(d+1)^2 \log\left(400(d+1)^2 K^3\right) \leq 48d^2 \log\left(12dK\right)$$

Combining this with Eq. (33) and Eq. (34), we get

$$\|\widehat{w}_{k,h} - w_{k,h}\|_{\Lambda_{k,h}} \leq 15dB_h^{\max}\sqrt{\log\left(\frac{12dK}{\delta}\right)}.$$

Further combining this with Eq. (32) proves Eq. (29).

**Showing Eq. (30).**

$$\left|\phi(s,a)^\top \widehat{w}_{k,h}\right|\mathbb{I}\{s \in \mathcal{Z}_h\}$$
$$\leq \left|\phi(s,a)^\top w_{k,h}\right|\mathbb{I}\{s \in \mathcal{Z}_h\} + \left|\phi(s,a)^\top\left(\widehat{w}_{k,h} - w_{k,h}\right)\right|\mathbb{I}\{s \in \mathcal{Z}_h\}$$
$$\leq \left(1 + \frac{1}{H}\right)\sup_{s' \in \mathcal{S}_{h+1}}\widehat{W}_k(s')\mathbb{I}\{s' \in \mathcal{Z}_{h+1}\} + C_\iota dB_h^{\max}\|\phi(s,a)\|_{\Lambda_{k,h}^{-1}}$$
$$\text{(by the definition of } w_{k,h} \text{ and Eq. (29))}$$
$$\leq B_h^{\max} + \left(\frac{(C_\iota dB_h^{\max})^2}{4\alpha} + \alpha\|\phi(s,a)\|_{\Lambda_{k,h}^{-1}}^2\right)\mathbb{I}\{s \in \mathcal{Z}_h\} \quad \text{(by Eq. (31) and AM-GM inequality)}$$
$$\leq B_h^{\max} + \left(\frac{1}{4H}B_h^{\max} + \alpha\rho^2\right)$$
$$\text{(by the condition specified in the lemma and that } \|\phi(s,a)\|_{\Lambda_{k,h}^{-1}} \leq \rho \text{ for } s \in \mathcal{Z}_h)$$
$$\leq B_h^{\max} + \frac{1}{2H}B_h^{\max} \qquad \text{(by the definition of } B_h^{\max})$$

This proves Eq. (30). $\qquad\square$

**Lemma 23.** *With the definition of Eq. (24) and Eq. (25), any $s \in \mathcal{S}_h$, we have*

$$B_k(s,a) \geq b_k(s,a) + \left(1 + \frac{1}{H}\right)\mathbb{E}_{s' \sim P(\cdot|s,a)}\mathbb{E}_{a' \sim \pi_k(\cdot|s')}\left[B_k(s',a')\mathbb{I}\{s' \in \mathcal{Z}_{h+1}\}\right] - \frac{(C_\iota dB^{\max})^2}{\alpha}.$$

*where $B^{\max} \triangleq \max_{h \in [H]} B_h^{\max}$ and $C_\iota$ is a logarithmic term defined in Lemma 22.*

*Proof.* Recall the definition of $w_{k,h}$ in Eq. (26), from the definition of linear MDP, for all $k, h, s, a$, we have

$$\phi(s,a)^\top w_{k,h}$$
$$= \left(1 + \frac{1}{H}\right)\mathbb{E}_{s' \sim P(\cdot|s,a)}\left[\widehat{W}(s')\mathbb{I}\{s' \in \mathcal{Z}_{h+1}\}\right]$$
$$= \left(1 + \frac{1}{H}\right)\mathbb{E}_{s' \sim P(\cdot|s,a)}\mathbb{E}_{a' \sim \pi_k(\cdot|s)}\left[\widehat{B}_k^+(s',a')\mathbb{I}\{s' \in \mathcal{Z}_{h+1}\}\right]$$
$$\geq \left(1 + \frac{1}{H}\right)\mathbb{E}_{s' \sim P(\cdot|s,a)}\mathbb{E}_{a' \sim \pi_k(\cdot|s)}\left[\widehat{B}_k(s',a')\mathbb{I}\{s' \in \mathcal{Z}_{h+1}\}\right]$$
$$= \left(1 + \frac{1}{H}\right)\mathbb{E}_{s' \sim P(\cdot|s,a)}\mathbb{E}_{a' \sim \pi_k(\cdot|s)}\left[\left(B_k(s',a') + \frac{\alpha}{4H}\|\phi(s',a')\|_{\Lambda_{k,h+1}^{-1}}^2 + \phi(s',a')^\top\left(\widehat{w}_{k,h+1} - w_{k,h+1}\right)\right)\mathbb{I}\{s' \in \mathcal{Z}_{h+1}\}\right]$$
$$\text{(by the definition of } \widehat{B}_k(s',a') \text{ in Line 7 and } B_k(s',a') \text{ in Eq. (25))}$$
$$\geq \left(1 + \frac{1}{H}\right)\mathbb{E}_{s' \sim P(\cdot|s,a)}\mathbb{E}_{a' \sim \pi_k(\cdot|s')}\left[B_k(s',a')\mathbb{I}\{s' \in \mathcal{Z}_{h+1}\}\right] - \frac{(C_\iota dB^{\max})^2}{\alpha}$$
$$\text{(Eq. (29) and AM-GM inequity)}$$

Thus, we have

$$B_k(s,a)$$

$$= b_k(s,a) + \phi(s,a)^\top w_{k,h}$$
$$\geq b_k(s,a) + \left(1 + \frac{1}{H}\right) \mathbb{E}_{s' \sim P(\cdot|s,a)} \mathbb{E}_{a' \sim \pi_k(\cdot|s')} \left[B_k(s',a') \mathbb{I}\{s' \in \mathcal{Z}_{h+1}\}\right] - \frac{(C_\iota d B^{\max})^2}{\alpha}.$$

$\square$

## D.3 Regret Analysis (achieving Eq. (22) using Algorithm 3)

The goal of this subsection is to prove Eq. (22) for the definitions of $b_k(s,a)$ and $B_k(s,a)$ in Eq. (24) and Eq. (25). We first decompose the left-hand side of Eq. (22).

$$\mathbb{E}\left[\sum_{k=1}^{K}\sum_{h=1}^{H} \mathbb{E}_h^\star \left[\langle Q_k(s,\cdot) - B_k(s,a), \pi_k(\cdot|s) - \pi^\star(\cdot|s)\rangle \mathbb{I}\{\mathcal{E}_h\}\right]\right]$$

$$\leq \mathbb{E}\left[\sum_{k=1}^{K}\sum_{h=1}^{H} \mathbb{E}_h^\star \left[\left\langle Q_k(s,\cdot) - \widehat{Q}_k(s,\cdot), \pi_k(\cdot|s)\right\rangle \mathbb{I}\{\mathcal{E}_h\}\right]\right]}_{\textbf{bias-1}}$$

$$+ \mathbb{E}\left[\sum_{k=1}^{K}\sum_{h=1}^{H} \mathbb{E}_h^\star \left[\left\langle \widehat{Q}_k(s,\cdot) - Q_k(s,\cdot), \pi^\star(\cdot|s)\right\rangle \mathbb{I}\{\mathcal{E}_h\}\right]\right]}_{\textbf{bias-2}}$$

$$+ \mathbb{E}\left[\sum_{k=1}^{K}\sum_{h=1}^{H} \mathbb{E}_h^\star \left[\left\langle \widehat{\mathbf{\Gamma}}_{k,h} - \widehat{\boldsymbol{B}}_{k,h}, \boldsymbol{H}_k(s) - \boldsymbol{H}_\star(s)\right\rangle \mathbb{I}\{\mathcal{E}_h\}\right]\right]}_{\textbf{ftrl}}$$

$$+ \mathbb{E}\left[\sum_{k=1}^{K}\sum_{h=1}^{H} \mathbb{E}_h^\star \left[\left\langle \widehat{B}_k(s,\cdot) - B_k(s,\cdot), \pi_k(\cdot|s) - \pi^\star(\cdot|s)\right\rangle \mathbb{I}\{\mathcal{E}_h\}\right]\right]}_{\textbf{bias-3}} \tag{35}$$

where we use that for $s \in \mathcal{S}_h$, $\mathbb{E}_{a \sim \pi(\cdot|s)} \widehat{Q}_k(s,a) = \langle \widehat{\text{Cov}}(s, \pi(\cdot|s)), \widehat{\mathbf{\Gamma}}_{k,h}\rangle$ and $\mathbb{E}_{a \sim \pi(\cdot|s)} \widehat{B}_k(s,a) = \langle \widehat{\text{Cov}}(s, \pi(\cdot|s)), \widehat{\boldsymbol{B}}_{k,h}\rangle$, and we define $\boldsymbol{H}_k(s) = \widehat{\text{Cov}}(s, \pi_k(\cdot|s))$, $\boldsymbol{H}_\star(s) = \widehat{\text{Cov}}(s, \pi^\star(\cdot|s))$.

We further deal with the **ftrl** term. This term is analyzed through the standard FTRL analysis. In order to deal with the issue that $F$ can be unbounded on the boundary of $\mathcal{H}_s$, we define the following auxiliary comparator:

$$\overline{\boldsymbol{H}}_\star(s) = \left(1 - \frac{1}{K^3}\right) \boldsymbol{H}_\star(s) + \frac{1}{K^3} \boldsymbol{H}_{\min}(s)$$

where $\boldsymbol{H}_{\min}(s) = \underset{\boldsymbol{H} \in \mathcal{H}_s}{\arg\min} F(\boldsymbol{H})$

Applying Lemma 46 for logdet FTRL, we have

$$\textbf{ftrl} = \mathbb{E}\left[\sum_{k=1}^{K}\sum_{h=1}^{H} \mathbb{E}_h^\star \left[\left\langle \widehat{\mathbf{\Gamma}}_{k,h} - \widehat{\boldsymbol{B}}_{k,h}, \boldsymbol{H}_k(s) - \boldsymbol{H}_\star(s)\right\rangle \mathbb{I}\{\mathcal{E}_h\}\right]\right]$$

$$= \mathbb{E}\left[\sum_{k=1}^{K}\sum_{h=1}^{H} \mathbb{E}_h^\star \left[\left\langle \widehat{\mathbf{\Gamma}}_{k,h} - \widehat{\boldsymbol{B}}_{k,h}, \boldsymbol{H}_k(s) - \overline{\boldsymbol{H}}_\star(s)\right\rangle \mathbb{I}\{\mathcal{E}_h\}\right]\right]$$

$$+ \mathbb{E}\left[\sum_{k=1}^{K}\sum_{h=1}^{H} \mathbb{E}_h^\star \left[\left\langle \widehat{\mathbf{\Gamma}}_{k,h} - \widehat{\boldsymbol{B}}_{k,h}, \overline{\boldsymbol{H}}_\star(s) - \boldsymbol{H}_\star(s)\right\rangle \mathbb{I}\{\mathcal{E}_h\}\right]\right]$$

$$\leq \mathbb{E}_h^\star \left[\frac{\tau\left(F\left(\overline{\boldsymbol{H}}_\star(s)\right) - \min_{\boldsymbol{H} \in \mathcal{H}_s} F(\boldsymbol{H})\right)}{\eta} \mathbb{I}\{\mathcal{E}_h\}\right]}_{\textbf{penalty}}$$

$$+ \underbrace{\mathbb{E}\left[\sum_{k=1}^{K}\sum_{h=1}^{H}\mathbb{E}_h^{\star}\left[\left(\max_{\boldsymbol{H}\in\mathcal{H}_s}\langle\boldsymbol{H}_k(s)-\boldsymbol{H},\widehat{\boldsymbol{\Gamma}}_{k,h}\rangle-\frac{D_F(\boldsymbol{H},\boldsymbol{H}_k(s))}{2\eta}\right)\mathbb{I}\{\mathcal{E}_h\}\right]\right]}_{\textbf{stability-1}}$$

$$+ \underbrace{\mathbb{E}\left[\sum_{k=1}^{K}\sum_{h=1}^{H}\mathbb{E}_h^{\star}\left[\left(\max_{\boldsymbol{H}\in\mathcal{H}_s}\langle\boldsymbol{H}_k(s)-\boldsymbol{H},-\widehat{\boldsymbol{B}}_{k,h}\rangle-\frac{D_F(\boldsymbol{H},\boldsymbol{H}_k(s))}{2\eta}\right)\mathbb{I}\{\mathcal{E}_h\}\right]\right]}_{\textbf{stability-2}}$$

$$+ \underbrace{\mathbb{E}\left[\sum_{k=1}^{K}\sum_{h=1}^{H}\mathbb{E}_h^{\star}\left[\left\langle\widehat{\boldsymbol{\Gamma}}_{k,h}-\widehat{\boldsymbol{B}}_{k,h},\overline{\boldsymbol{H}}_{\star}(s)-\boldsymbol{H}_{\star}(s)\right\rangle\mathbb{I}\{\mathcal{E}_h\}\right]\right]}_{\textbf{error}} \tag{36}$$

Below, we further bound the individual terms in Eq. (35) and Eq. (36).

### D.3.1    Bound **bias-1**, **bias-2**, **bias-3** in Eq. (35)

**Lemma 24.** *For any policy $\pi_k$, there exists a $\mathrm{q}_{k,h}$ such that for any $s \in \mathcal{S}_h$, $Q_k(s,a) = \phi(s,a)^{\top}\mathrm{q}_{k,h}$. Moreover, $\|\mathrm{q}_{k,h}\|_2 \leq H\sqrt{d}$.*

*Proof.* Define $\mathrm{q}_{k,h} = \theta_{k,h} + \sum_{s'\in\mathcal{S}_{h+1}}\psi(s')\mathbb{E}_{a'\sim\pi_k(\cdot|s')}\left[Q_k(s',a')\right]$, we have

$$Q_k(s,a) = Q^{\pi_k}(s,a;\ell_k) = \ell_k(s,a) + \mathbb{E}_{s'\sim P(\cdot|s,a)}\mathbb{E}_{a'\sim\pi_k(\cdot|s')}\left[Q_k(s',a')\right]$$

$$= \phi(s,a)^{\top}\left(\theta_{k,h} + \sum_{s'\in\mathcal{S}_{h+1}}\psi(s')\mathbb{E}_{a'\sim\pi_k(\cdot|s')}\left[Q_k(s',a')\right]\right)$$

$$= \phi(s,a)^{\top}\mathrm{q}_{k,h}.$$

Moreover,

$$\|\mathrm{q}_{k,h}\|_2 = \left\|\theta_{k,h} + \sum_{s'\in\mathcal{S}_{h+1}}\psi(s')\mathbb{E}_{a'\sim\pi_k(\cdot|s')}\left[Q_k(s',a')\right]\right\|_2 \leq \sqrt{d} + \sqrt{d}(H-1) = \sqrt{d}H.$$

$\square$

**Lemma 25.** *Let $\Sigma_{k,h} = \mathbb{E}_{s\sim\mu_h^k}\mathbb{E}_{a\sim\pi_k(\cdot|s)}\left[\phi(s,a)\phi(s,a)^{\top}\right]$. If $\gamma = \frac{5d\log(6dHK/\delta)}{\tau}$, then with probability of $1-\delta$, for all $k,h$,*

$$\left\|\left(\widehat{\Sigma}_{k,h}-\Sigma_{k,h}\right)\mathrm{q}_{k,h}\right\|_{\widehat{\Sigma}_{k,h}^{-1}}^2 \leq \mathcal{O}\left(\frac{d^2H^2\log\left(dHK/\delta\right)}{\tau}\right)$$

*Proof.* This follows the fact the $\|\mathrm{q}_{k,h}\|_2 \leq H\sqrt{d}$ given in Lemma 24 and the matrix concentration bound in Lemma 14 of Liu et al. (2023a) with a union bound over $k,h$. Taking a union bound for all $k,h$ finishes the proof. $\square$

**Lemma 26.** *If $\gamma = \frac{5d\log(6dHK/\delta)}{\tau}$, then*

$$\textbf{bias-1} \leq \widetilde{\mathcal{O}}\left(\frac{d^2H^3}{\tau\beta}K\right) + \frac{\beta}{4H}\mathbb{E}\left[\sum_{k=1}^{K}\sum_{h=1}^{H}\mathbb{E}_h^{\star}\mathbb{E}_{a\sim\pi_k(\cdot|s)}\left[\|\phi(s,a)\|_{\widehat{\Sigma}_{k,h}^{-1}}^2\mathbb{I}\{\mathcal{E}_h\}\right]\right]$$

$$\textbf{bias-2} \leq \widetilde{\mathcal{O}}\left(\frac{d^2H^3}{\tau\beta}K\right) + \frac{\beta}{4H}\mathbb{E}\left[\sum_{k=1}^{K}\sum_{h=1}^{H}\mathbb{E}_h^{\star}\mathbb{E}_{a\sim\pi^{\star}(\cdot|s)}\left[\|\phi(s,a)\|_{\widehat{\Sigma}_{k,h}^{-1}}^2\mathbb{I}\{\mathcal{E}_h\}\right]\right].$$

*Proof.* Let $\mathbb{E}_k [\cdot]$ be the expectation conditioned on history up to episode $k-1$. We have

$$\mathbb{E}_k \left[ \sum_{t=h}^{H} \ell_{k,t} \right] = \mathbb{E}_k \left[ Q_k(s_{k,h}, a_{k,h}) \right] = \mathbb{E}_k \left[ \phi(s_{k,h}, a_{k,h})^\top \mathrm{q}_{k,h} \right].$$

Therefore,

$$\mathbb{E}_k \left[ \widehat{\mathrm{q}}_{k,h} \right] = \mathbb{E}_k \left[ \widehat{\Sigma}_{k,h}^{-1} \phi(s_{k,h}, a_{k,h}) \phi(s_{k,h}, a_{k,h})^\top \mathrm{q}_{k,h} \right] = \widehat{\Sigma}_{k,h}^{-1} \Sigma_{k,h} \mathrm{q}_{k,h},$$

and for $s \in \mathcal{S}_h$,

$$\begin{aligned}
\mathbb{E}_k \left[ Q_k(s,a) - \widehat{Q}_k(s,a) \right] &= \mathbb{E}_k \left[ \phi(s,a)^\top \mathrm{q}_{k,h} - \phi(s,a)^\top \widehat{\mathrm{q}}_{k,h} \right] \\
&= \phi(s,a)^\top \left( I - \widehat{\Sigma}_{k,h}^{-1} \Sigma_{k,h} \right) \mathrm{q}_{k,h} \\
&= \phi(s,a)^\top \widehat{\Sigma}_{k,h}^{-1} \left( \widehat{\Sigma}_{k,h} - \Sigma_{k,h} \right) \mathrm{q}_{k,h} \\
&\leq \| \phi(s,a) \|_{\widehat{\Sigma}_{k,h}^{-1}} \left\| \left( \widehat{\Sigma}_{k,h} - \Sigma_{k,h} \right) \mathrm{q}_{k,h} \right\|_{\widehat{\Sigma}_{k,h}^{-1}} \quad \text{(Cauchy-Schwarz)} \\
&\leq \mathcal{O} \left( \sqrt{\frac{d^2 H^2 \log (dK/\delta)}{\tau}} \| \phi(s,a) \|_{\widehat{\Sigma}_{k,h}^{-1}} \right) \quad \text{(Lemma 25)} \\
&\leq \mathcal{O} \left( \frac{d^2 H^3 \log (dK/\delta)}{\tau \beta} \right) + \frac{\beta}{4H} \| \phi(s,a) \|_{\widehat{\Sigma}_{k,h}^{-1}}^2 . \\
&\qquad\qquad\qquad\qquad\qquad\qquad\qquad \text{(AM-GM inequality)}
\end{aligned}$$

Thus,

$$\begin{aligned}
\textbf{bias-1} &= \mathbb{E} \left[ \sum_{k=1}^{K} \sum_{h=1}^{H} \mathbb{E}_h^\star \left[ \left\langle Q_k(s,\cdot) - \widehat{Q}_k(s,\cdot), \pi_k(\cdot|s) \right\rangle \mathbb{I}\{\mathcal{E}_h\} \right] \right] \\
&\leq \widetilde{\mathcal{O}} \left( \frac{d^2 H^3}{\tau \beta} K \right) + \frac{\beta}{4H} \mathbb{E} \left[ \sum_{k=1}^{K} \sum_{h=1}^{H} \mathbb{E}_h^\star \mathbb{E}_{a \sim \pi_k(\cdot|s)} \left[ \| \phi(s,a) \|_{\widehat{\Sigma}_{k,h}^{-1}}^2 \mathbb{I}\{\mathcal{E}_h\} \right] \right]
\end{aligned}$$

Similarly, we can prove

$$\textbf{bias-2} \leq \widetilde{\mathcal{O}} \left( \frac{d^2 H^3}{\tau \beta} K \right) + \frac{\beta}{4H} \mathbb{E} \left[ \sum_{k=1}^{K} \sum_{h=1}^{H} \mathbb{E}_h^\star \mathbb{E}_{a \sim \pi^\star(\cdot|s)} \left[ \| \phi(s,a) \|_{\widehat{\Sigma}_{k,h}^{-1}}^2 \mathbb{I}\{\mathcal{E}_h\} \right] \right]$$

$\square$

**Lemma 27.** *Suppose that $B_h^{\max} \leq \frac{\alpha}{C_\iota^2 H d^2}$ where $C_\iota = 15 \sqrt{\log \left( \frac{12dK}{\delta} \right)}$. Then*

$$\textbf{bias-3} \leq \widetilde{\mathcal{O}} \left( \frac{H^2 d^2 (B^{\max})^2}{\alpha} K \right) + \frac{\alpha}{2H} \mathbb{E} \left[ \sum_{k=1}^{K} \sum_{h=1}^{H} \mathbb{E}_h^\star \mathbb{E}_{a \sim \pi_k(\cdot|s)} \left[ \| \phi(s,a) \|_{\Lambda_{k,h}^{-1}}^2 \mathbb{I}\{\mathcal{E}_h\} \right] \right].$$

*Proof.* By Eq. (29) and AM-GM inequality, we have that with probability at least $1 - \delta$, for all $k, h, s, a$, $\left| \phi(s,a)^\top \left( \widehat{w}_{k,h} - w_{k,h} \right) \right| \leq \frac{H(C_\iota d B^{\max})^2}{\alpha} + \frac{\alpha}{4H} \| \phi(s,a) \|_{\Lambda_{k,h}^{-1}}^2$. Combining this with the definitions of $\widehat{B}_k(s,a)$ in Line 7 and $B_k(s,a)$ in Eq. (25), we get

$$\begin{aligned}
\widehat{B}_k(s,a) - B_k(s,a) &= \frac{\alpha}{4H} \| \phi(s,a) \|_{\Lambda_{k,h}^{-1}}^2 + \phi(s,a)^\top \left( \widehat{w}_{k,h} - w_{k,h} \right) \geq -\frac{H(C_\iota d B^{\max})^2}{\alpha} \\
\widehat{B}_k(s,a) - B_k(s,a) &\leq \frac{\alpha}{2H} \| \phi(s,a) \|_{\Lambda_{k,h}^{-1}}^2 + \frac{H(C_\iota d B^{\max})^2}{\alpha}.
\end{aligned}$$

With the two inequalities above, we have

**bias-3**

$$= \mathbb{E}\left[\sum_{k=1}^{K}\sum_{h=1}^{H}\mathbb{E}_h^\star\left[\left\langle \widehat{B}_k(s,\cdot) - B_k(s,\cdot), \pi_k(\cdot|s) - \pi^\star(\cdot|s)\right\rangle \mathbb{I}\{\mathcal{E}_h\}\right]\right]$$

$$\leq \mathbb{E}\left[\sum_{k=1}^{K}\sum_{h=1}^{H}\mathbb{E}_h^\star\mathbb{E}_{a\sim\pi_k(\cdot|s)}\left[\widehat{B}_k(s,a) - B_k(s,a)\right]\right] - \mathbb{E}\left[\sum_{k=1}^{K}\sum_{h=1}^{H}\mathbb{E}_h^\star\mathbb{E}_{a\sim\pi^\star(\cdot|s)}\left[\widehat{B}_k(s,a) - B_k(s,a)\right]\right]$$

$$\leq \widetilde{\mathcal{O}}\left(\frac{H^2(dB^{\max})^2}{\alpha}K\right) + \frac{\alpha}{2H}\left[\sum_{k=1}^{K}\sum_{h=1}^{H}\mathbb{E}_h^\star\mathbb{E}_{a\sim\pi_k(\cdot|s)}\left[\|\phi(s,a)\|_{\Lambda_{k,h}^{-1}}^2\mathbb{I}\{\mathcal{E}_h\}\right]\right].$$

$\square$

### D.3.2 BOUND **penalty** IN EQ. (36)

**Lemma 28.** penalty $\leq \frac{3d\tau\log(K)}{\eta}$

*Proof.* Since $\overline{\boldsymbol{H}}_\star(s) = \left(1 - \frac{1}{K^3}\right)\boldsymbol{H}_\star(s) + \frac{1}{K^3}\boldsymbol{H}_{\min}(s)$, we have $\overline{\boldsymbol{H}}_\star(s) \succeq \frac{1}{K^3}\boldsymbol{H}_{\min}(s)$. Then

$$\frac{\tau\left(F(\overline{\boldsymbol{H}}_\star(s)) - \min_{\boldsymbol{H}\in\mathcal{H}_s}F(\boldsymbol{H})\right)}{\eta} = \frac{\tau}{\eta}\log\frac{\det(\boldsymbol{H}_{\min}(s))}{\det(\overline{\boldsymbol{H}}_\star(s))} \leq \frac{3d\tau\log(K)}{\eta}$$

$\square$

### D.3.3 BOUND **error** IN EQ. (36)

**Lemma 29.** error $\leq \mathcal{O}(H)$.

*Proof.* By the choices of $\beta, \gamma, \alpha$, it holds that $\frac{\beta}{\gamma} + \alpha\rho^2 \leq \mathcal{O}(K)$ and $\frac{H}{\gamma} \leq \mathcal{O}(K)$. Let $\pi_{\min}$ be such that $\boldsymbol{H}_{\min}(s) = \mathbb{E}_{a\sim\pi_{\min}(\cdot|s)}\begin{bmatrix}\phi(s,a)\phi(s,a)^\top & \phi(s,a) \\ \phi(s,a)^\top & 1\end{bmatrix}$. For $s\in\mathcal{S}_h$, we have $\left|\widehat{Q}_k(s,a)\right| = |\phi(s,a)^\top\widehat{\mathrm{q}}_{k,h}| \leq \frac{H}{\gamma}$ by the definition of $\widehat{\mathrm{q}}_{k,h}$, and $\|\widehat{w}_{k,h}\|_2 \leq K^2$, which implies $\left|\widehat{B}_k(s,a)\right|\mathbb{I}\{s\in\mathcal{Z}_h\} \leq 2K^2$.

Therefore,

$$\mathbb{E}\left[\sum_{k=1}^{K}\sum_{h=1}^{H}\mathbb{E}_h^\star\left[\left\langle\widehat{\boldsymbol{\Gamma}}_{k,h} - \widehat{\boldsymbol{B}}_{k,h}, \overline{\boldsymbol{H}}_\star(s) - \boldsymbol{H}_\star(s)\right\rangle\mathbb{I}\{\mathcal{E}_h\}\right]\right]$$

$$= \frac{1}{K^3}\mathbb{E}\left[\sum_{k=1}^{K}\sum_{h=1}^{H}\mathbb{E}_h^\star\left[\left\langle\widehat{\boldsymbol{\Gamma}}_{k,h} - \widehat{\boldsymbol{B}}_{k,h}, \boldsymbol{H}_{\min}(s) - \boldsymbol{H}_\star(s)\right\rangle\mathbb{I}\{\mathcal{E}_h\}\right]\right] \quad \text{(by the definition of } \overline{\boldsymbol{H}}_\star(s)\text{)}$$

$$= \frac{1}{K^3}\mathbb{E}\left[\sum_{k=1}^{K}\sum_{h=1}^{H}\mathbb{E}_h^\star\left[\left\langle\widehat{Q}_k(s,\cdot) - \widehat{B}_k(s,\cdot), \pi_{\min}(\cdot|s) - \pi^\star(\cdot|s)\right\rangle\mathbb{I}\{\mathcal{E}_h\}\right]\right]$$

$$\leq \mathcal{O}(H)$$

$\square$

### D.3.4 BOUND **stability-1** IN EQ. (36)

To bound **stability-1**, we first introduce a useful identity in Lemma 30. This is first proposed in Zimmert and Lattimore (2022) and restated in Liu et al. (2023a).

**Lemma 30** (Lemma 25 in Liu et al. (2023a))**.** *Let* $\boldsymbol{G} = \begin{bmatrix}G + gg^\top & g \\ g^\top & 1\end{bmatrix}$ *and* $\boldsymbol{H} = \begin{bmatrix}H + hh^\top & h \\ h^\top & 1\end{bmatrix}$, *we have*

$$D_F(\boldsymbol{G}, \boldsymbol{H}) = D_F(G, H) + \|g - h\|_{H^{-1}}^2 \geq \|g - h\|_{H^{-1}}^2$$

**Lemma 31** (Lemma 12 in Liu et al. (2023a)). *Define* $\Sigma_{k,h} = \mathbb{E}_{s \sim \mu_h^k} \mathbb{E}_{a \sim \pi_k(\cdot|s)} \left[ \phi(s,a)\phi(s,a)^\top \right]$. *If* $\gamma = \frac{5d\log(6dHK/\delta)}{\tau}$, *for any* $k, h$, *with probability* $1 - \delta$, *we have*

$$\widehat{\Sigma}_{k,h} = \frac{1}{\tau} \sum_{(s,a,s') \in \mathcal{D}_{k,h}} \phi(s,a)\phi(s,a)^\top + \gamma I \succeq \frac{1}{2} \mathbb{E}_{s \sim \mu_h^k} \mathbb{E}_{a \sim \pi_k(\cdot|s)} \left[ \phi(s,a)\phi(s,a)^\top \right] = \frac{1}{2}\Sigma_{k,h}.$$

**Lemma 32.** *If* $\gamma = \frac{5d\log(6dHK/\delta)}{\tau}$, *then*

$$\textbf{stability-1} \leq \eta H^2 \mathbb{E} \left[ \sum_{k=1}^{K} \sum_{h=1}^{H} \mathbb{E}_h^\star \mathbb{E}_{a \sim \pi_k(\cdot|s)} \left[ \|\phi(s,a)\|_{\widehat{\Sigma}_{k,h}^{-1}}^2 \mathbb{I}\{\mathcal{E}_h\} \right] \right].$$

*Proof.* In this proof, we define

- $\phi(s,\pi) = \mathbb{E}_{a \sim \pi(\cdot|s)} \left[ \phi(s,a) \right]$

- $\overline{\mathrm{Cov}}(s,\pi) = \mathbb{E}_{a \sim \pi(\cdot|s)} \left[ \left( \phi(s,a) - \phi(s,\pi) \right) \left( \phi(s,a) - \phi(s,\pi) \right)^\top \right]$

- $\mathrm{Cov}(s,\pi) = \mathbb{E}_{a \sim \pi(\cdot|s)} \left[ \phi(s,a)\phi(s,a)^\top \right]$

Let $\mathbb{E}_k\left[\cdot\right]$ be the expectation conditioned on history up to episode $k - 1$. Consider a fixed $s \in \mathcal{S}_h$ and any policy $\pi$. Let

$$\boldsymbol{H}(s) = \mathbb{E}_{a \sim \pi(\cdot|s)} \begin{bmatrix} \phi(s,a)\phi(s,a)^\top & \phi(s,a) \\ \phi(s,a)^\top & 1 \end{bmatrix}.$$

We have

$$\mathbb{E}_k \left[ \left\langle \boldsymbol{H}_k(s) - \boldsymbol{H}(s), \widehat{\boldsymbol{\Gamma}}_{k,h} \right\rangle - \frac{D(\boldsymbol{H}(s), \boldsymbol{H}_k(s))}{2\eta} \right]$$

$$\leq \mathbb{E}_k \left[ \langle \phi(s,\pi_k) - \phi(s,\pi), \widehat{\mathrm{q}}_{k,h} \rangle - \frac{\|\phi(s,\pi_k) - \phi(s,\pi)\|_{\overline{\mathrm{Cov}}(s,\pi_k)^{-1}}^2}{2\eta} \right] \qquad \text{(Lemma 30)}$$

$$\leq \mathbb{E}_k \left[ \|\phi(s,\pi_k) - \phi(s,\pi)\|_{\overline{\mathrm{Cov}}(s,\pi_k)^{-1}} \|\widehat{\mathrm{q}}_{k,h}\|_{\overline{\mathrm{Cov}}(s,\pi_k)} - \frac{\|\phi(s,\pi_k) - \phi(s,\pi)\|_{\overline{\mathrm{Cov}}(s,\pi_k)^{-1}}^2}{2\eta} \right]$$

$$\leq \frac{\eta}{2} \mathbb{E}_k \left[ \|\widehat{\mathrm{q}}_{k,h}\|_{\overline{\mathrm{Cov}}(s,\pi_k)}^2 \right] \qquad \text{(AM-GM inequality)}$$

$$\leq \frac{\eta}{2} \mathbb{E}_k \left[ \left\| \widehat{\Sigma}_{k,h}^{-1} \phi(s_{k,h}, a_{k,h}) \sum_{t=h}^{H} \ell_t^k \right\|_{\mathrm{Cov}(s,\pi_k)}^2 \right] \qquad (\mathrm{Cov}(s,\pi) \succeq \overline{\mathrm{Cov}}(s,\pi))$$

$$\leq \frac{\eta H^2}{2} \mathbb{E}_k \left[ \phi(s_{k,h}, a_{k,h})^\top \widehat{\Sigma}_{k,h}^{-1} \mathrm{Cov}(s,\pi_k) \widehat{\Sigma}_{k,h}^{-1} \phi(s_{k,h}, a_{k,h}) \right]$$

$$= \frac{\eta H^2}{2} \mathbb{E}_k \left[ \mathrm{Tr} \left( \phi(s_{k,h}, a_{k,h}) \phi(s_{k,h}, a_{k,h})^\top \widehat{\Sigma}_{k,h}^{-1} \mathrm{Cov}(s,\pi_k) \widehat{\Sigma}_{k,h}^{-1} \right) \right]$$

$$= \frac{\eta H^2}{2} \mathrm{Tr} \left( \Sigma_{k,h} \widehat{\Sigma}_{k,h}^{-1} \mathrm{Cov}(s,\pi_k) \widehat{\Sigma}_{k,h}^{-1} \right)$$

$$\leq \eta H^2 \mathrm{Tr} \left( \mathrm{Cov}(s,\pi_k) \widehat{\Sigma}_{k,h}^{-1} \right) \qquad \text{(Lemma 31)}$$

$$= \eta H^2 \mathbb{E}_{a \sim \pi_k(\cdot|s)} \left[ \|\phi(s,a)\|_{\widehat{\Sigma}_{k,h}^{-1}}^2 \right]$$

Taking expectation and adding indicator for $s$, and then summing over all $k, h$ finish the proof. $\qquad \square$

Given $F(X) = -\log\det(X)$, $D^2 F(X) = X^{-1} \otimes X^{-1}$ where $\otimes$ is the Kronecker prod-

uct. For any matrix $A = \begin{bmatrix} a_1 & a_2 & \cdots & a_n \end{bmatrix}$, let $\mathrm{vec}(A) = \begin{bmatrix} a_1 \\ \vdots \\ a_n \end{bmatrix}$ which vectorizes ma-

trix $A$ to a column vector by stacking the columns $A$. The second order directional derivative for $F$ is $D^2 F(X)[A, A] = \mathrm{vec}(A)^\top \left(X^{-1} \otimes X^{-1}\right)\mathrm{vec}(A) = \mathrm{Tr}(A^\top X^{-1} A X^{-1})$. We define $\|A\|_{\nabla^2 F(X)} = \sqrt{\mathrm{Tr}(A^\top X^{-1} A X^{-1})}$ and $\|A\|_{\nabla^{-2} F(X)} = \sqrt{\mathrm{Tr}(A^\top X A X)}$. It is a pseudo-norm, and more discussion can be found in Appendix D of Zimmert et al. (2022). In the following analysis, we will only use one property of this pseudo-norm which is similar to the Holder inequality. It is standard and also appears as Lemma 8 in Liu et al. (2023a).

**Lemma 33.** *For any two symmetric matrices $A, B$ and positive definite matrix $X$,*

$$\langle A, B \rangle \le \|A\|_{\nabla^2 F(X)} \|B\|_{\nabla^{-2} F(X)}$$

*Proof.* Since $(X \otimes X)^{-1} = X^{-1} \otimes X^{-1}$, from Holder inequality, we have

$$\langle A, B \rangle = \langle \mathrm{vec}(A), \mathrm{vec}(B) \rangle \le \|\mathrm{vec}(A)\|_{X^{-1}\otimes X^{-1}} \|\mathrm{vec}(B)\|_{(X^{-1}\otimes X^{-1})^{-1}} = \|A\|_{\nabla^2 F(X)} \|B\|_{\nabla^{-2} F(X)}$$

$\square$

Lemma 34 gives a general argument to bound **stability-2** with arbitrary $\boldsymbol{B} \in \mathbb{R}^{(d+1)\times(d+1)}$. Similar theorems are also stated in Lemma 34 of Dann et al. (2023b) and Lemma 27 of Liu et al. (2023a).

**Lemma 34.** *For any matrix $\boldsymbol{B} \in \mathbb{R}^{(d+1)\times(d+1)}$, for any state $s$, given $\sqrt{\mathrm{Tr}(\boldsymbol{H}_k(s)\boldsymbol{B}\boldsymbol{H}_k(s)\boldsymbol{B})} \le m$, if $\eta \le \frac{1}{16m}$,*

$$\max_{\boldsymbol{H} \in \mathcal{H}_s} \langle \boldsymbol{H}_k(s) - \boldsymbol{H}, -\boldsymbol{B} \rangle - \frac{D_F(\boldsymbol{H}, \boldsymbol{H}_k(s))}{\eta} \le 8\eta\|\boldsymbol{B}\|^2_{\nabla^{-2} F(\boldsymbol{H}_k(s))} = 8\eta\,\mathrm{Tr}\left(\boldsymbol{H}_k(s)\boldsymbol{B}\boldsymbol{H}_k(s)\boldsymbol{B}\right).$$

*Proof.* For any $\boldsymbol{H} \in \mathcal{H}_s$, define

$$G(\boldsymbol{H}) = \langle \boldsymbol{H}_k(s) - \boldsymbol{H}, -\boldsymbol{B} \rangle - \frac{D_F(\boldsymbol{H}, \boldsymbol{H}_k(s))}{\eta}$$

and $\lambda = \|\boldsymbol{B}\|_{\nabla^{-2} F(\boldsymbol{H}_k(s))}$. Since $\sqrt{\mathrm{Tr}(\boldsymbol{H}_k(s)\boldsymbol{B}\boldsymbol{H}_k(s)\boldsymbol{B})} \le m$ and $\eta \le \frac{1}{16m}$, we have

$$\eta\lambda = \eta\|\boldsymbol{B}\|_{\nabla^{-2} F(\boldsymbol{H}_k(s))} = \eta\sqrt{\mathrm{Tr}(\boldsymbol{H}_k(s)\boldsymbol{B}\boldsymbol{H}_k(s)\boldsymbol{B})} \le \eta m \le \frac{1}{16}.$$

Let $\boldsymbol{H}'$ be the maximizer of $G$. Since $G(\boldsymbol{H}_k(s)) = 0$, we have $G(\boldsymbol{H}') \ge 0$. It suffices to show $\|\boldsymbol{H}' - \boldsymbol{H}_k(s)\|_{\nabla^2 F(\boldsymbol{H}_k(s))} \le 8\eta\lambda$ because from Lemma 33 it leads to

$$G(\boldsymbol{H}') \le \|\boldsymbol{H}_k(s) - \boldsymbol{H}'\|_{\nabla^2 F(\boldsymbol{H}_k(s))} \|\boldsymbol{B}\|_{\nabla^{-2} F(\boldsymbol{H}_k(s))} \le 8\eta\lambda\|\boldsymbol{B}\|_{\nabla^{-2} F(\boldsymbol{H}_k(s))} = 8\eta\|\boldsymbol{B}\|^2_{\nabla^{-2} F(\boldsymbol{H}_k(s))}$$

To show $\|\boldsymbol{H}' - \boldsymbol{H}_k(s)\|_{\nabla^2 F(\boldsymbol{H}_k(s))} \le 8\eta\lambda$, it suffices to show that for all $\boldsymbol{U}$ such that $\|\boldsymbol{U} - \boldsymbol{H}_k(s)\|_{\nabla^2 F(\boldsymbol{H}_k(s))} = 8\eta\lambda$, $G(\boldsymbol{U}) \le 0$. This is because given this condition, if $\|\boldsymbol{H}' - \boldsymbol{H}_k(s)\|_{\nabla^2 F(\boldsymbol{H}_k(s))} > 8\eta\lambda$, then there is a $\boldsymbol{U}$ in the line segment between $\boldsymbol{H}_k(s)$ and $\boldsymbol{H}'$ such that $\|\boldsymbol{U} - \boldsymbol{H}_k(s)\|_{\nabla^2 F(\boldsymbol{H}_k(s))} = 8\eta\lambda$. From the condition, $G(\boldsymbol{U}) \le 0 \le \min\{G(\boldsymbol{H}_k(s)), G(\boldsymbol{H}')\}$ which contradicts to the concavity of $G$.

Now consider any $\boldsymbol{U}$ such that $\|\boldsymbol{U} - \boldsymbol{H}_k(s)\|_{\nabla^2 F(\boldsymbol{H}_k(s))} = 8\eta\lambda$. By Taylor expansion, there exists $\boldsymbol{U}'$ in the line segment between $\boldsymbol{U}$ and $\boldsymbol{H}_k(s)$ such that

$$G(\boldsymbol{U}) \le \|\boldsymbol{U} - \boldsymbol{H}_k(s)\|_{\nabla^2 F(\boldsymbol{H}_k(s))} \|\boldsymbol{B}\|_{\nabla^{-2} F(\boldsymbol{H}_k(s))} - \frac{1}{2\eta}\|\boldsymbol{U} - \boldsymbol{H}_k(s)\|^2_{\nabla^2 F(\boldsymbol{U}')}$$

We have $\|\boldsymbol{U}' - \boldsymbol{H}_k(s)\|_{\nabla^2 F(\boldsymbol{H}_k(s))} \le \|\boldsymbol{U} - \boldsymbol{H}_k(s)\|_{\nabla^2 F(\boldsymbol{H}_k(s))} = 8\eta\lambda \le \frac{1}{2}$. From the Equation 2.2 in page 23 of Nemirovski (2004) (also appear in Eq.(5) of Abernethy et al. (2009)) and $\log\det$ is a self-concordant function, we have $\|\boldsymbol{U} - \boldsymbol{H}_k(s)\|_{\nabla^2 F(\boldsymbol{U}')}^2 \ge \frac{1}{4}\|\boldsymbol{U} - \boldsymbol{H}_k(s)\|_{\nabla^2 F(\boldsymbol{H}_k(s))}^2$. Thus, we have

$$G(\boldsymbol{U}) \le \|\boldsymbol{U} - \boldsymbol{H}_k(s)\|_{\nabla^2 F(\boldsymbol{H}_k(s))}\|\boldsymbol{B}\|_{\nabla^{-2} F(\boldsymbol{H}_k(s))} - \frac{1}{8\eta}\|\boldsymbol{U} - \boldsymbol{H}_k(s)\|_{(\boldsymbol{H}_k(s))^{-1}}^2 = 8\eta\lambda^2 - \frac{(8\eta\lambda)^2}{8\eta} = 0.$$

$\square$

**Lemma 35.** *Given* $B_k(s,a) = \beta\|\phi(s,a)\|_{\widehat{\Sigma}_{k,h}^{-1}}^2 + \alpha\left(1 - \frac{1}{4H}\right)\|\phi(s,a)\|_{\Lambda_{k,h}^{-1}}^2 + \phi(s,a)^\top w_{k,h}$ *defined in Eq.* (25) *for* $s \in \mathcal{S}_h$, *if* $\eta \le \frac{1}{3328 H^2\left(\frac{\beta}{\gamma} + \alpha\rho^2\right)}$, *we have*

$$\textbf{stability-2} \le \frac{1}{2H}\mathbb{E}\left[\sum_{k=1}^{K}\sum_{h=1}^{H}\mathbb{E}_h^\star\mathbb{E}_{a\sim\pi_k(\cdot|s)}\left[B_k(s,a)\mathbb{I}\{\mathcal{E}_h\}\right]\right] + \widetilde{\mathcal{O}}\left(\frac{(dB^{\max})^2}{\alpha}K\right).$$

*Proof.* We can decompose the bonus matrix in the following form and consider stability separately

$$\widehat{\boldsymbol{B}}_{k,h} = \begin{bmatrix} \beta\widehat{\Sigma}_{k,h}^{-1} + \alpha\Lambda_{k,h}^{-1} & \frac{1}{2}\widehat{w}_{k,h} \\ \frac{1}{2}\widehat{w}_h^{k\top} & 0 \end{bmatrix} = \underbrace{\begin{bmatrix} \beta\widehat{\Sigma}_{k,h}^{-1} + \alpha\Lambda_{k,h}^{-1} & 0 \\ 0 & 0 \end{bmatrix}}_{\widehat{\boldsymbol{B}}_{k,h}^1} + \underbrace{\begin{bmatrix} 0 & \frac{1}{2}\widehat{w}_{k,h} \\ \frac{1}{2}\widehat{w}_h^{k\top} & 0 \end{bmatrix}}_{\widehat{\boldsymbol{B}}_{k,h}^2}.$$

Then we have

$$\begin{aligned}
\textbf{stability-2} &= \mathbb{E}\left[\sum_{k=1}^{K}\sum_{h=1}^{H}\mathbb{E}_h^\star\left[\left(\max_{\boldsymbol{H}\in\mathcal{H}_s}\langle\boldsymbol{H}_k(s) - \boldsymbol{H}, -\widehat{\boldsymbol{B}}_{k,h}\rangle - \frac{D_F(\boldsymbol{H},\boldsymbol{H}_k(s))}{2\eta}\right)\mathbb{I}\{\mathcal{E}_h\}\right]\right] \\
&\le \mathbb{E}\left[\sum_{k=1}^{K}\sum_{h=1}^{H}\mathbb{E}_h^\star\left[\left(\max_{\boldsymbol{H}\in\mathcal{H}_s}\langle\boldsymbol{H}_k(s) - \boldsymbol{H}, -\widehat{\boldsymbol{B}}_{k,h}^1\rangle - \frac{D_F(\boldsymbol{H},\boldsymbol{H}_k(s))}{4\eta}\right)\mathbb{I}\{\mathcal{E}_h\}\right]\right] \\
&\quad + \mathbb{E}\left[\sum_{k=1}^{K}\sum_{h=1}^{H}\mathbb{E}_h^\star\left[\left(\max_{\boldsymbol{H}\in\mathcal{H}_s}\langle\boldsymbol{H}_k(s) - \boldsymbol{H}, -\widehat{\boldsymbol{B}}_{k,h}^2\rangle - \frac{D_F(\boldsymbol{H},\boldsymbol{H}_k(s))}{4\eta}\right)\mathbb{I}\{\mathcal{E}_h\}\right]\right]
\end{aligned}$$

For any matrix $A \in \mathbb{R}^{d\times d}$ with all non-negative eigenvalues, we have

$$\text{Tr}\left(A^2\right) = \sum_{i=1}^{d}\lambda_i(A^2) \le \left(\sum_{i=1}^{d}\lambda_i(A)\right)^2 = \text{Tr}(A)^2$$

Since both $\phi(s,a)\phi(s,a)^\top$ and $\beta\widehat{\Sigma}_{k,h}^{-1} + \alpha\Lambda_{k,h}^{-1}$ are positive semi-definite, the eigenvalues of $\phi(s,a)\phi(s,a)^\top\left(\beta\widehat{\Sigma}_{k,h}^{-1} + \alpha\Lambda_{k,h}^{-1}\right)$ are all non-negative. Thus, for any $s \in \mathcal{Z}_h$, we have

$$\begin{aligned}
\sqrt{\text{Tr}\left(\boldsymbol{H}_k(s)\widehat{\boldsymbol{B}}_{k,h}^1\boldsymbol{H}_k(s)\widehat{\boldsymbol{B}}_{k,h}^1\right)} &\le \sqrt{\text{Tr}\left(\left(\mathbb{E}_{a\sim\pi_k(\cdot|s)}\left[\phi(s,a)\phi(s,a)^\top\left(\beta\widehat{\Sigma}_{k,h}^{-1} + \alpha\Lambda_{k,h}^{-1}\right)\right]\right)^2\right)} \\
&\le \text{Tr}\left(\mathbb{E}_{a\sim\pi_k(\cdot|s)}\left[\phi(s,a)\phi(s,a)^\top\left(\beta\widehat{\Sigma}_{k,h}^{-1} + \alpha\Lambda_{k,h}^{-1}\right)\right]\right) \\
&= \mathbb{E}_{a\sim\pi_k(\cdot|s)}\left[\beta\|\phi(s,a)\|_{\widehat{\Sigma}_{k,h}^{-1}}^2 + \alpha\|\phi(s,a)\|_{\Lambda_{k,h}^{-1}}^2\right] \\
&\le \frac{\beta}{\gamma} + \alpha\rho^2. \qquad (\|\phi(s,a)\|_{\Lambda_{k,h}^{-1}} \le \rho \text{ for } s \in \mathcal{Z}_h)
\end{aligned}$$

Thus, from Lemma 34, if $\eta \le \frac{1}{64H\left(\frac{\beta}{\gamma} + \alpha\rho^2\right)}$, we have

$$\mathbb{E}\left[\sum_{k=1}^{K}\sum_{h=1}^{H}\mathbb{E}_h^\star\left[\left(\max_{\boldsymbol{H}\in\mathcal{H}_s}\langle\boldsymbol{H}_k(s) - \boldsymbol{H}, -\widehat{\boldsymbol{B}}_{k,h}^1\rangle - \frac{D_F(\boldsymbol{H},\boldsymbol{H}_k(s))}{4\eta}\right)\mathbb{I}\{\mathcal{E}_h\}\right]\right]$$

$$\leq 8\eta \sum_{k=1}^{K} \sum_{h=1}^{H} \mathbb{E}_h^{\star} \left[ \mathrm{Tr} \left( \boldsymbol{H}_k(s) \widehat{\boldsymbol{B}}_{k,h}^1 \boldsymbol{H}_k(s) \widehat{\boldsymbol{B}}_{k,h}^1 \right) \mathbb{I}\{\mathcal{E}_h\} \right]$$

$$\leq \frac{1}{8H} \sum_{k=1}^{K} \sum_{h=1}^{H} \mathbb{E}_h^{\star} \left[ \sqrt{\mathrm{Tr} \left( \boldsymbol{H}_k(s) \widehat{\boldsymbol{B}}_{k,h}^1 \boldsymbol{H}_k(s) \widehat{\boldsymbol{B}}_{k,h}^1 \right)} \mathbb{I}\{\mathcal{E}_h\} \right]$$

$$\leq \frac{1}{8H} \sum_{k=1}^{K} \sum_{h=1}^{H} \mathbb{E}_h^{\star} \mathbb{E}_{a \sim \pi_k(\cdot|s)} \left[ \left( \beta \|\phi(s,a)\|_{\widehat{\Sigma}_{k,h}^{-1}}^2 + \alpha \|\phi(s,a)\|_{\Lambda_{k,h}^{-1}}^2 \right) \mathbb{I}\{\mathcal{E}_h\} \right]. \tag{37}$$

Now consider $\widehat{\boldsymbol{B}}_{k,h}^2$, for any $s \in \mathcal{Z}_h$, we have

$$\sqrt{\mathrm{Tr} \left( \boldsymbol{H}_k(s) \widehat{\boldsymbol{B}}_{k,h}^2 \boldsymbol{H}_k(s) \widehat{\boldsymbol{B}}_{k,h}^2 \right)}$$

$$= \sqrt{2 \mathrm{Tr} \left( (\widehat{w}_{k,h})^\top \mathbb{E}_{a \sim \pi_k(\cdot|s)} [\phi(s,a)] \mathbb{E}_{a \sim \pi_k(\cdot|s)} [\phi(s,a)^\top] \widehat{w}_{k,h} + (\widehat{w}_{k,h})^\top \mathbb{E}_{a \sim \pi_k(\cdot|s)} [\phi(s,a)\phi(s,a)^\top] \widehat{w}_{k,h} \right)}$$

$$\leq 2 \sqrt{\mathbb{E}_{a \sim \pi_k(\cdot|s)} \left[ (\phi(s,a)^\top \widehat{w}_{k,h})^2 \right]} \leq 2 \left( 1 + \frac{1}{2H} \right) B_h^{\max} \leq 26H \left( \frac{\beta}{\gamma} + \alpha \rho^2 \right). \quad \text{(by Eq. (30))}$$

Similarly, from Lemma 34, if $\eta \leq \frac{1}{3328 H^2 \left( \frac{\beta}{\gamma} + \alpha \rho^2 \right)} \leq \frac{1}{256 H B^{\max}}$, then for all $h \in [H]$ and any state $s \in \mathcal{Z}_h$, we have

$$\max_{\boldsymbol{H} \in \mathcal{H}} \left\langle \boldsymbol{H}_k(s) - \boldsymbol{H}, -\widehat{\boldsymbol{B}}_{k,h}^2 \right\rangle - \frac{D_F(\boldsymbol{H}, \boldsymbol{H}_k(s))}{4\eta}$$

$$\leq 8\eta \, \mathrm{Tr} \left( \boldsymbol{H}_k(s) \widehat{\boldsymbol{B}}_{k,h}^2 \boldsymbol{H}_k(s) \widehat{\boldsymbol{B}}_{k,h}^2 \right)$$

$$\leq 32\eta \mathbb{E}_{a \sim \pi_k(\cdot|s)} \left[ \left( \phi(s,a)^\top \widehat{w}_{k,h} \right)^2 \right]$$

$$= 32\eta \mathbb{E}_{a \sim \pi_k(\cdot|s)} \left[ \left( \phi(s,a)^\top w_{k,h} + \phi(s,a)^\top (\widehat{w}_{k,h} - w_{k,h}) \right)^2 \right]$$

$$\leq 64\eta \mathbb{E}_{a \sim \pi_k(\cdot|s)} \left[ \left( \phi(s,a)^\top w_{k,h} \right)^2 \right] + 64\eta \mathbb{E}_{a \sim \pi_k(\cdot|s)} \left[ \left( \phi(s,a)^\top (\widehat{w}_{k,h} - w_{k,h}) \right)^2 \right]$$
$$((a+b)^2 \leq 2a^2 + 2b^2)$$

$$\leq \frac{1}{4H} \mathbb{E}_{a \sim \pi_k(\cdot|s)} \left[ \phi(s,a)^\top w_{k,h} \right] + \frac{1}{H} \mathbb{E}_{a \sim \pi_k(\cdot|s)} \left[ |\phi(s,a)^\top (\widehat{w}_{k,h} - w_{k,h})| \right]$$
$$\text{(see the explanation below)}$$

$$\leq \frac{(C_\iota d B^{\max})^2}{H\alpha} + \frac{1}{4H} \mathbb{E}_{a \sim \pi_k(\cdot|s)} \left[ \phi(s,a)^\top w_{k,h} \right] + \frac{1}{4H} \mathbb{E}_{a \sim \pi_k(\cdot|s)} \left[ \alpha \|\phi(s,a)\|_{\Lambda_{k,h}^{-1}}^2 \right].$$
$$\text{(Lemma 22 and AM-GM)}$$

where in the second-last inequality, we use the condition of $\eta$ and that

$$|\phi(s,a)^\top w_{k,h}| \leq \left( 1 + \frac{1}{H} \right) \sup_{s' \in \mathcal{S}_h} \widehat{W}(s') \mathbb{I}\{s' \in \mathcal{Z}_{h+1}\} \leq B^{\max}, \qquad \text{(by Eq. (31))}$$

$$|\phi(s,a)^\top (\widehat{w}_{k,h} - w_{k,h})| \leq |\phi(s,a)^\top \widehat{w}_{k,h}| + |\phi(s,a)^\top w_{k,h}| \leq \left( 2 + \frac{1}{2H} \right) B^{\max}. \quad \text{(by Eq. (30))}$$

Thus,

$$\mathbb{E} \left[ \sum_{k=1}^{K} \sum_{h=1}^{H} \mathbb{E}_h^{\star} \left[ \left( \max_{\boldsymbol{H} \in \mathcal{H}_s} \left\langle \boldsymbol{H}_k(s) - \boldsymbol{H}, -\widehat{\boldsymbol{B}}_{k,h}^2 \right\rangle - \frac{D_F(\boldsymbol{H}, \boldsymbol{H}_k(s))}{4\eta} \right) \mathbb{I}\{\mathcal{E}_h\} \right] \right]$$

$$\leq \frac{1}{4H} \left[ \sum_{k=1}^{K} \sum_{h=1}^{H} \mathbb{E}_h^{\star} \mathbb{E}_{a \sim \pi_k(\cdot|s)} \left[ \phi(s,a)^\top w_{k,h} \mathbb{I}\{\mathcal{E}_h\} \right] \right]$$

$$+ \frac{1}{4H} \left[ \sum_{k=1}^{K} \sum_{h=1}^{H} \mathbb{E}_h^{\star} \mathbb{E}_{a \sim \pi_k(\cdot|s)} \left[ \alpha \|\phi(s,a)\|_{\Lambda_{k,h}^{-1}}^2 \mathbb{I}\{\mathcal{E}_h\} \right] \right] + \widetilde{\mathcal{O}} \left( \frac{(d B^{\max})^2}{\alpha} K \right). \tag{38}$$

Combining Eq. (37) and Eq. (38), we see that if $\eta \leq \frac{1}{3328H^2\left(\frac{\beta}{\gamma}+\alpha\rho^2\right)}$, then

**stability-2**

$$\leq \frac{1}{8H}\mathbb{E}\left[\sum_{k=1}^{K}\sum_{h=1}^{H}\mathbb{E}_h^{\star}\mathbb{E}_{a\sim\pi_k(\cdot|s)}\left[\left(\beta\|\phi(s,a)\|_{\widehat{\Sigma}_{k,h}^{-1}}^2 + \alpha\|\phi(s,a)\|_{\Lambda_{k,h}^{-1}}^2\right)\mathbb{I}\{\mathcal{E}_h\}\right]\right]$$

$$+ \frac{1}{4H}\mathbb{E}\left[\sum_{k=1}^{K}\sum_{h=1}^{H}\mathbb{E}_h^{\star}\mathbb{E}_{a\sim\pi_k(\cdot|s)}\left[\phi(s,a)^{\top}w_{k,h}\mathbb{I}\{\mathcal{E}_h\}\right]\right]$$

$$+ \frac{1}{4H}\mathbb{E}\left[\sum_{k=1}^{K}\sum_{h=1}^{H}\mathbb{E}_h^{\star}\mathbb{E}_{a\sim\pi_k(\cdot|s)}\left[\alpha\|\phi(s,a)\|_{\Lambda_{k,h}^{-1}}^2\mathbb{I}\{\mathcal{E}_h\}\right]\right] + \widetilde{\mathcal{O}}\left(\frac{(dB^{\max})^2}{\alpha}K\right)$$

$$\leq \frac{1}{2H}\mathbb{E}\left[\sum_{k=1}^{K}\sum_{h=1}^{H}\mathbb{E}_h^{\star}\mathbb{E}_{a\sim\pi_k(\cdot|s)}\left[B_k(s,a)\mathbb{I}\{\mathcal{E}_h\}\right]\right] + \widetilde{\mathcal{O}}\left(\frac{(dB^{\max})^2}{\alpha}K\right).$$

$\square$

**Lemma 36.** *If* $\eta \leq \frac{1}{3228H^2\left(\frac{\beta}{\gamma}+\alpha\rho^2\right)}$ *and* $\gamma = \frac{5d\log(6dHK/\delta)}{\tau}$ *and* $B_h^{\max} \leq \frac{\alpha}{225\log(\frac{dK}{\delta})Hd^2}$ *and* $\eta H^2 \leq \frac{3}{4}\beta$, *then we have*

$$\mathbb{E}\left[\sum_{k=1}^{K}\sum_{h=1}^{H}\mathbb{E}_h^{\star}\left[\langle Q_k(s,\cdot) - B_k(s,a), \pi_k(\cdot|s) - \pi^{\star}(\cdot|s)\rangle\,\mathbb{I}\{\mathcal{E}_h\}\right]\right]$$

$$\leq \widetilde{\mathcal{O}}\left(\frac{d^2H^3}{\tau\beta}K + \frac{d^2H^2(B^{\max})^2}{\alpha}K + \frac{d\tau}{\eta}\right)$$

$$+ \sum_{k=1}^{K}\sum_{h=1}^{H}\mathbb{E}_h^{\star}\mathbb{E}_{a\sim\pi^{\star}(\cdot|s)}\left[b_k(s,a)\mathbb{I}\{\mathcal{E}_h\}\right] + \frac{1}{H}\sum_{k=1}^{K}\sum_{h=1}^{H}\mathbb{E}_h^{\star}\mathbb{E}_{a\sim\pi_k(\cdot|s)}\left[B_k(s,a)\mathbb{I}\{\mathcal{E}_h\}\right]$$

*where* $b_k(s,a)$ *is defined in Eq. (24).*

*Proof.* Since $\eta \leq \frac{1}{3328H^2\left(\frac{\beta}{\gamma}+\alpha\rho^2\right)}$, adding up the bound in Lemma 28, Lemma 29, Lemma 32, and Lemma 35 following the decomposition in Eq. (36), we get

$$\mathbf{ftrl} \leq \widetilde{\mathcal{O}}\left(\frac{d\tau}{\eta} + H + \frac{d^2(B^{\max})^2}{\alpha}K\right) + \eta H^2\sum_{k=1}^{K}\sum_{h=1}^{H}\mathbb{E}_h^{\star}\mathbb{E}_{a\sim\pi_k(\cdot|s)}\left[\|\phi(s,a)\|_{\widehat{\Sigma}_{k,h}^{-1}}^2\mathbb{I}\{\mathcal{E}_h\}\right]$$

$$+ \frac{1}{2H}\sum_{k=1}^{K}\sum_{h=1}^{H}\mathbb{E}_h^{\star}\mathbb{E}_{a\sim\pi_k(\cdot|s)}\left[B_k(s,a)\mathbb{I}\{\mathcal{E}_h\}\right]. \tag{39}$$

From the decomposition in Eq. (35) and Lemma 26, Lemma 27, and Eq. (39), under the specified conditions, we have

$$\mathbb{E}\left[\sum_{k=1}^{K}\sum_{h=1}^{H}\mathbb{E}_h^{\star}\left[\langle Q_k(s,\cdot) - B_k(s,a), \pi_k(\cdot|s) - \pi^{\star}(\cdot|s)\rangle\,\mathbb{I}\{\mathcal{E}_h\}\right]\right]$$

$$\leq \mathbf{bias\text{-}1} + \mathbf{bias\text{-}2} + \mathbf{bias\text{-}3} + \mathbf{ftrl}$$

$$\leq \widetilde{\mathcal{O}}\left(\frac{d^2H^3}{\tau\beta}K + \frac{d^2H^2(B^{\max})^2}{\alpha}K + \frac{d\tau}{\eta}\right)$$

$$+ \left(\frac{\beta}{4} + \eta H^2\right)\sum_{k=1}^{K}\sum_{h=1}^{H}\mathbb{E}_h^{\star}\mathbb{E}_{a\sim\pi^{\star}(\cdot|s)}\left[\|\phi(s,a)\|_{\widehat{\Sigma}_{k,h}^{-1}}^2\mathbb{I}\{s\in\mathcal{Z}_h\}\right]$$

$$+ \frac{1}{H}\sum_{k=1}^{K}\sum_{h=1}^{H}\mathbb{E}_h^{\star}\mathbb{E}_{a\sim\pi_k(\cdot|s)}\left[B_{k,h}(s,a)\mathbb{I}\{\mathcal{E}_h\}\right]$$

$$\leq \widetilde{\mathcal{O}}\left(\frac{d^2 H^3}{\tau\beta}K + \frac{d^2 H^2(B^{\max})^2}{\alpha}K + \frac{d\tau}{\eta}\right)$$

$$+ \sum_{k=1}^{K}\sum_{h=1}^{H}\mathbb{E}_h^\star\mathbb{E}_{a\sim\pi^\star(\cdot|s)}\left[b_k(s,a)\right] + \frac{1}{H}\sum_{k=1}^{K}\sum_{h=1}^{H}\mathbb{E}_h^\star\mathbb{E}_{a\sim\pi_k(\cdot|s)}\left[B_{k,h}(s,a)\mathbb{I}\{\mathcal{E}_h\}\right]$$

where in the last inequality we use $\frac{\beta}{4} + \eta H^2 \leq \beta$.

$\square$

### D.4 FINAL STEPS

**Lemma 37.** *Let $s \in \mathcal{S}_h$. We have*

$$B_k(s,a) \leq r_k(s,a) + \left(1 + \frac{1}{H}\right)\mathbb{E}_{s'\sim P(\cdot|s,a)}\mathbb{E}_{a'\sim\pi_k(\cdot|s')}\left[B_k(s',a')\mathbb{I}\{s'\in\mathcal{Z}_{h+1}\}\right]$$

*where we define*

$$r_k(s,a) = b_k(s,a) + \mathbb{E}_{s'\sim P(\cdot|s,a)}\mathbb{E}_{a'\sim\pi_k(\cdot|s')}\left[\alpha\|\phi(s',a')\|^2_{\Lambda_{k,h+1}^{-1}}\mathbb{I}\{s'\in\mathcal{Z}_{h+1}\}\right] + \frac{2(C_\iota dB^{\max})^2}{\alpha}.$$

*Proof.* Since $B_k(s,a) \geq 0$, we have

$$\left|\widehat{B}_k^+(s,a) - B_k(s,a)\right| \leq \left|\widehat{B}_k(s,a) - B_k(s,a)\right|$$
$$= \left|\frac{\alpha}{4H}\|\phi(s,a)\|^2_{\Lambda_{k,h}^{-1}} + \phi(s,a)^\top\left(\widehat{w}_{k,h} - w_{k,h}\right)\right|$$
$$\leq \frac{(C_\iota dB^{\max})^2}{\alpha} + \alpha\|\phi(s,a)\|^2_{\Lambda_{k,h}^{-1}}. \qquad \text{(by Lemma 22)}$$

Thus,

$$\phi(s,a)^\top w_{k,h}$$
$$= \left(1 + \frac{1}{H}\right)\mathbb{E}_{s'\sim P(\cdot|s,a)}\mathbb{E}_{a'\sim\pi_k(\cdot|s)}\left[\widehat{B}_k^+(s',a')\mathbb{I}\{s'\in\mathcal{Z}_{h+1}\}\right]$$
$$\leq \left(1 + \frac{1}{H}\right)\mathbb{E}_{s'\sim P(\cdot|s,a)}\mathbb{E}_{a'\sim\pi_k(\cdot|s')}\left[B_k(s',a')\mathbb{I}\{s'\in\mathcal{Z}_{h+1}\}\right]$$
$$+ \mathbb{E}_{s'\sim P(\cdot|s,a)}\mathbb{E}_{a'\sim\pi_k(\cdot|s')}\left[\alpha\|\phi(s',a')\|^2_{\Lambda_{k,h+1}^{-1}}\mathbb{I}\{s'\in\mathcal{Z}_{h+1}\}\right] + \frac{2(C_\iota dB^{\max})^2}{\alpha},$$

and

$$B_k(s,a)$$
$$= b_k(s,a) + \phi(s,a)^\top w_{k,h}$$
$$\leq b_k(s,a) + \left(1 + \frac{1}{H}\right)\mathbb{E}_{s'\sim P(\cdot|s,a)}\mathbb{E}_{a'\sim\pi_k(\cdot|s')}\left[B_k(s',a')\mathbb{I}\{s'\in\mathcal{Z}_{h+1}\}\right]$$
$$+ \mathbb{E}_{s'\sim P(\cdot|s,a)}\mathbb{E}_{a'\sim\pi_k(\cdot|s')}\left[\alpha\|\phi(s',a')\|^2_{\Lambda_{k,h+1}^{-1}}\mathbb{I}\{s'\in\mathcal{Z}_{h+1}\}\right] + \frac{2(C_\iota dB^{\max})^2}{\alpha}$$
$$= r_k(s,a) + \left(1 + \frac{1}{H}\right)\mathbb{E}_{s'\sim P(\cdot|s,a)}\mathbb{E}_{a'\sim\pi_k(\cdot|s')}\left[B_k(s',a')\mathbb{I}\{s'\in\mathcal{Z}_{h+1}\}\right].$$

$\square$

**Theorem 38.** *Suppose the parameters are properly chosen so that all conditions in Lemma 36 holds (see the proof for the final parameters). Then the regret of Algorithm 3 has the following guarantee*

$$\mathbb{E}\left[\mathcal{R}_K\right] \leq \widetilde{\mathcal{O}}\left(d^{\frac{3}{2}}H^3 K^{\frac{3}{4}}\right).$$

*Proof.* By Lemma 23, we have for $s \in \mathcal{S}_h$,

$$B_k(s,a) \geq b_k(s,a) + \left(1 + \frac{1}{H}\right) \mathbb{E}_{s' \sim P(\cdot|s,a)} \mathbb{E}_{a' \sim \pi_k(\cdot|s')} \left[B_k(s',a') \mathbb{I}\{s' \in \mathcal{Z}_{h+1}\}\right] - \widetilde{\mathcal{O}}\left(\frac{d^2 (B^{\max})^2}{\alpha}\right).$$

Combining this with Lemma 36, we see that the two conditions of Lemma 21 are satisfied with $f = \widetilde{\mathcal{O}}\left(\frac{d^2 (B^{\max})^2}{\alpha}\right)$ and $g = \widetilde{\mathcal{O}}\left(\frac{d^2 H^3}{\tau \beta} K + \frac{d^2 H^2 (B^{\max})^2}{\alpha} K + \frac{d\tau}{\eta}\right)$. Thus, by directly applying Lemma 21, we have

$$\textbf{reg-term} \leq \widetilde{\mathcal{O}}\left(\frac{d^2 H^3}{\tau \beta} K + \frac{d^2 H^2 (B^{\max})^2}{\alpha} K + \frac{d\tau}{\eta}\right) + \left(1 + \frac{1}{H}\right) \mathbb{E}\left[\sum_{k=1}^{K} \mathbb{E}_{a \sim \pi_k(\cdot|s_1)} \left[B_k(s_1, a)\right]\right]$$

To bound the last term, below we use induction to show that for $s \in \mathcal{S}_h$, the following holds:

$$\mathbb{E}_{a \sim \pi_k(\cdot|s)} \left[B_k(s,a)\right] \leq \left(1 + \frac{1}{H}\right)^{H-h} V^{\pi_k}(s; r_k)$$

for the $r_k$ defined in Lemma 37.

**Base case (step $H$).** for any $s \in \mathcal{S}_H$, we have

$$\mathbb{E}_{a \sim \pi_k(\cdot|s)} \left[B_k(s,a)\right] = \mathbb{E}_{a \sim \pi_k(\cdot|s)} \left[b_k(s,a)\right] \leq V^{\pi_k}(s; r_k)$$

**Induction.** Assume that for any $s \in \mathcal{S}_{h+1}$,

$$\mathbb{E}_{a \sim \pi_k(\cdot|s)} \left[B_k(s,a)\right] \leq \left(1 + \frac{1}{H}\right)^{H-h-1} V^{\pi_k}(s; r_k).$$

Then for any $s \in \mathcal{S}_h$, we have

$$\mathbb{E}_{a \sim \pi_k(\cdot|s)} \left[B_k(s,a)\right]$$
$$\leq \mathbb{E}_{a \sim \pi_k(\cdot|s)} \left[r_k(s,a) + \left(1 + \frac{1}{H}\right) \mathbb{E}_{s' \sim P(\cdot|s,a)} \mathbb{E}_{a' \sim \pi_k(\cdot|s')} \left[B_k(s',a')\right]\right] \qquad \text{(Lemma 37)}$$
$$\leq \mathbb{E}_{a \sim \pi_k(\cdot|s)} \left[r_k(s,a) + \left(1 + \frac{1}{H}\right)^{H-h} \mathbb{E}_{s' \sim P(\cdot|s,a)} \left[V^{\pi_k}(s'; r_k)\right]\right] \qquad \text{(induction hypothesis)}$$
$$\leq \left(1 + \frac{1}{H}\right)^{H-h} \mathbb{E}_{a \sim \pi_k(\cdot|s)} \left[r_k(s,a) + \mathbb{E}_{s' \sim P(\cdot|s,a)} \left[V^{\pi_k}(s'; r_k)\right]\right] \qquad (r_k(s,a) \geq 0)$$
$$= \left(1 + \frac{1}{H}\right)^{H-h} V^{\pi_k}(s; r_k).$$

Since $\left(1 + \frac{1}{H}\right)^H < e < 3$, we have

$$\left(1 + \frac{1}{H}\right) \sum_{k=1}^{K} \mathbb{E}_{a \sim \pi_k(\cdot|s_1)} \left[B_k(s_1, a)\right]$$
$$\leq 3 \sum_{k=1}^{K} V^{\pi_k}(s_1; r_k)$$
$$= \widetilde{\mathcal{O}}\left(\sum_{k=1}^{K} \sum_{h=1}^{H} \mathbb{E}_{s \sim \mu_h^k} \mathbb{E}_{a \sim \pi_k(\cdot|s)} \left[\beta \|\phi(s,a)\|_{\widehat{\Sigma}_{k,h}^{-1}}^2 + \alpha \|\phi(s,a)\|_{\Lambda_{k,h}^{-1}}^2\right] + \frac{(d B^{\max})^2}{\alpha} K\right)$$
$$\leq \widetilde{\mathcal{O}}\left(\beta d H K + \alpha d H + \frac{(d B^{\max})^2}{\alpha} K\right).$$

Given that $B_h^{\max} = 4H \left(1 + \frac{1}{H}\right)^{2(H-h+1)} \left(\frac{\beta}{\gamma} + \alpha \rho^2\right)$, we have $B^{\max} \leq 36H \left(\frac{\beta}{\gamma} + \alpha \rho^2\right)$. Thus,

$$\textbf{reg-term} \leq \widetilde{\mathcal{O}}\left(\frac{d^2 H^3}{\tau \beta} K + \frac{d^2 H^4 \beta^2}{\alpha \gamma^2} K + d^2 H^4 \alpha \rho^4 K + \frac{d\tau}{\eta} + \beta d H K + \alpha d H\right)$$

We pick $\rho = H^{-\frac{1}{2}} d^{-\frac{1}{4}} K^{-\frac{1}{4}}$, $\beta = \sqrt{d} K^{-\frac{1}{4}}$, $\alpha = H K^{\frac{3}{4}}$, $\tau = K^{\frac{1}{2}}$, $\delta = \frac{1}{K^3}$, $\gamma = \frac{5d \log\left(6dHK^4\right)}{\tau}$, $\eta = \frac{K^{-\frac{1}{4}}}{3328\sqrt{d}H^2}$. In that case, if $\sqrt{K} \geq 16200 d^{\frac{3}{2}} H \log\left(dK^4\right) = \widetilde{\Omega}\left(d^{\frac{3}{2}} H\right)$, all conditions in [Lemma 36](#) are satisfied and **reg-term** $\leq \widetilde{\mathcal{O}}(d^{\frac{3}{2}} H^3 K^{\frac{3}{4}})$.

By [Lemma 15](#), the initial pure exploration phase takes

$$K_0 = \widetilde{\mathcal{O}}\left(\frac{\frac{dH}{\rho^2} + d^4 H^4}{\epsilon_{\text{cov}}}\right) = \widetilde{\mathcal{O}}\left(d^{\frac{3}{2}} H^2 K^{\frac{3}{4}} + d^4 H^4 K^{\frac{1}{4}}\right)$$

episodes, which contributes to an additional regret of $HK_0 = \widetilde{\mathcal{O}}(d^{\frac{3}{2}} H^3 K^{\frac{3}{4}})$ (omitting lower-order terms). Finally, the cost of ignoring states outside of $\mathcal{Z}$ is $H^3 K^{-\frac{3}{4}}$ as calculated in [Eq. (20)](#).

Combining all parts of regret finishes the proof.

$\square$

## E  AUXILARY LEMMAS

### E.1  UNIFORM CONCENTRATION VIA COVERING

Consider policy class

$$\mathbf{P}(s) = \left\{ p : \ \widehat{\text{Cov}}(s, p) = \underset{\boldsymbol{H} \in \mathcal{H}_s}{\arg\min} \left\{ \langle \boldsymbol{H}, \boldsymbol{Z} \rangle + F(\boldsymbol{H}) \right\}, \text{for } \boldsymbol{Z} \in \mathcal{Z} \right\} \tag{40}$$

where $\mathcal{Z} = [-K^3, K^3]^{(d+1) \times (d+1)} \cap \mathbb{S}$ with $\mathbb{S}$ denoting the set of symmetric matrices. We define the following function class.

**Definition 39.** *For any $h$ and any $s \in \mathcal{S}_h$,*

$$V_h\left(s; \Sigma, \Lambda, w, p\right) = \left(1 + \frac{1}{H}\right) \mathbb{E}_{a \sim p} \left[ \left[ \beta \|\phi(s,a)\|^2_{\Sigma^{-1}} + \phi^\top(s,a) w + 2\alpha \|\phi(s,a)\|^2_{\Lambda^{-1}} \right]^+ \mathbb{I}\{s \in \mathcal{Z}_h\} \right],$$

$$\mathcal{V}_h = \{V\left(s\,; \Sigma, \Lambda, w, p\right) \mid \lambda_{\min}\left(\Sigma\right) \geq \gamma, \lambda_{\min}\left(\Lambda\right) \geq 1, \|w\| \leq K^2, p \in \mathbf{P}(s)\}.$$

*where $\mathbf{P}(s)$ is defined in [Eq. (40)](#).*

We propose the following two covering lemma. [Lemma 40](#) is standard which argues the upper bound of the cover number of a Euclidian ball. [Lemma 41](#) inherits from Lemma 15 in [Liu et al. (2023a)](#).

**Lemma 40** (Cover number of Euclidian Ball)**.** *For any $\epsilon > 0$, the $\epsilon$-covering of the Euclidean ball in $\mathbb{R}^d$ with radius $R > 0$ is upper bounded by $\left(1 + \frac{2R}{\epsilon}\right)^d$.*

**Lemma 41** (Covering for logdet policy class, Lemma 15 in [Liu et al. (2023a)](#))**.** *For any $s$, there exists an $\epsilon$-cover $\mathbf{P}'(s)$ of $\mathbf{P}(s)$ with size $\log|\mathbf{P}'(s)| = (d+1)^2 \log \frac{24(d+1)^2}{\epsilon}$ such that for any $p \in \mathbf{P}(s)$, there exists an $p' \in \mathbf{P}'(s)$ satisfying*

$$\left\| \widehat{\text{Cov}}(s, p) - \widehat{\text{Cov}}(s, p') \right\|_{\text{F}} \leq \epsilon.$$

[Lemma 42](#) gives the covering number of function class $\mathcal{V}_h$.

**Lemma 42.** *Let $\mathcal{N}_\epsilon(\mathcal{V}_h)$ be the $\|\cdot\|_\infty$ $\epsilon$-covering number of function class $\mathcal{V}_h$, for any $h$, we have*

$$\log\left(\mathcal{N}_\epsilon(\mathcal{V}_h)\right) \leq d \log\left(1 + \frac{16K^2}{\epsilon}\right) + d^2 \log\left(1 + \frac{16\sqrt{d}\beta}{\epsilon\gamma}\right) + d^2 \log\left(1 + \frac{16\sqrt{d}\alpha}{\epsilon}\right)$$

$$+ (d+1)^2 \log\left(\frac{96(d+1)^2\left(2\beta\gamma^{-1} + 2\alpha + K^2\right)}{\epsilon}\right).$$

*If $\frac{\beta}{\gamma} + 2\alpha \leq K^2$, then*

$$\log\left(\mathcal{N}_\epsilon(\mathcal{V}_h)\right) \leq 4(d+1)^2 \log\left(\frac{400(d+1)^2 K^2}{\epsilon}\right).$$

*Proof.* Define

$$B\left(s, a; D, E, w\right) = \|\phi(s, a)\|_D^2 + \|\phi(s, a)\|_E^2 + \phi^\top(s, a)w$$

and consider the following function classes

$$\mathcal{B} = \left\{ B\left(s, a; D, E, w\right) \mid \|D\|_2 \leq 2\beta\gamma^{-1}, \|E\|_2 \leq 2\alpha, \|w\|_2 \leq 2K^2 \right\},$$

$$\widetilde{\mathcal{V}} = \left\{ \mathbb{E}_{a \sim p}\left[B\left(s, a; D, E, w\right)\right] \mid B\left(s, a; D, E, w\right) \in \mathcal{B}, p \in \mathbf{P}(s) \right\}.$$

For any $V_1 = \mathbb{E}_{a \sim p_1}\left[B(s, a; D_1, E_1, w_1)\right]$ and $V_2 = \mathbb{E}_{a \sim p_2}\left[B(s, a; D_2, E_2, w_2)\right]$, it holds that

$$\begin{aligned}
|V_1 - V_2| &= \left|\mathbb{E}_{a \sim p_1}\left[B(s, a; D_1, E_1, w_1)\right] - \mathbb{E}_{a \sim p_2}\left[B(s, a; D_2, E_2, w_2)\right]\right| \\
&= \left|\mathbb{E}_{a \sim p_1}\left[B(s, a; D_1, E_1, w_1)\right] - \mathbb{E}_{a \sim p_1}\left[B(s, a; D_2, E_2, w_2)\right]\right| \\
&\quad + \left|\mathbb{E}_{a \sim p_1}\left[B(s, a; D_2, E_2, w_2)\right] - \mathbb{E}_{a \sim p_2}\left[B(s, a; D_2, E_2, w_2)\right]\right|.
\end{aligned}$$

On the one hand, we have

$$\begin{aligned}
&|B\left(s, a; D_1, E_1, w_1\right) - B\left(s, a; D_2, E_2, w_2\right)| \\
&= \left|\|\phi(s, a)\|_{D_1}^2 - \|\phi(s, a)\|_{D_2}^2\right| + \left|\phi^\top(s, a)\left(w_1 - w_2\right)\right| + \left|\|\phi(s, a)\|_{E_1}^2 - \|\phi(s, a)\|_{E_2}^2\right| \\
&= \left|\phi(s, a)^\top\left(D_1 - D_2\right)\phi(s, a)\right| + \left|\phi^\top(s, a)\left(w_1 - w_2\right)\right| + \left|\phi(s, a)^\top\left(E_1 - E_2\right)\phi(s, a)\right| \\
&\leq \|D_1 - D_2\|_2 + \|w_1 - w_2\|_2 + \|E_1 - E_2\|_2 \qquad\qquad (\|\phi(s, a)\|_2 \leq 1) \\
&\leq \|D_1 - D_2\|_{\mathrm{F}} + \|w_1 - w_2\|_2 + \|E_1 - E_2\|_{\mathrm{F}}.
\end{aligned}$$

Since for any matrix $A \in \mathbb{R}^{d \times d}$, $\|A\|_{\mathrm{F}} \leq \sqrt{d}\|A\|_2$, we consider a $\frac{\epsilon}{4}$ net on $\{D \in \mathbb{R}^{d \times d} \mid \|D\|_{\mathrm{F}} \leq 2\sqrt{d}\beta\gamma^{-1}\}$, a $\frac{\epsilon}{4}$ net on $\{w \in \mathbb{R}^d \mid \|w\|_2 \leq 2K^2\}$, a $\frac{\epsilon}{4}$ net on $\{E \in \mathbb{R}^{d \times d} \mid \|E\|_{\mathrm{F}} \leq 2\sqrt{d}\alpha\}$. From Lemma 40, the log size of these nets is

$$d\log\left(1 + \frac{16K^2}{\epsilon}\right) + d^2\log\left(1 + \frac{16\sqrt{d}\beta}{\epsilon\gamma}\right) + d^2\log\left(1 + \frac{16\sqrt{d}\alpha}{\epsilon}\right).$$

On the other hand, define $\boldsymbol{B}_2 = \begin{bmatrix} D_2 + E_2 & \frac{1}{2}w_2 \\ \frac{1}{2}w_2^\top & 0 \end{bmatrix}$, we have $\|\boldsymbol{B}_2\|_2 \leq 2\beta\gamma^{-1} + 2\alpha + K^2$ and

$$\begin{aligned}
&\left|\mathbb{E}_{a \sim p_1}\left[B(s, a; D_2, E_2, w_2)\right] - \mathbb{E}_{a \sim p_2}\left[B(s, a; D_2, E_2, w_2)\right]\right| \\
&= \left|\left\langle \widehat{\mathrm{Cov}}(s, p_1) - \widehat{\mathrm{Cov}}(s, p_2), \boldsymbol{B}_2 \right\rangle\right| \\
&\leq \left\|\widehat{\mathrm{Cov}}(s, p_1) - \widehat{\mathrm{Cov}}(s, p_2)\right\|_2 \|\boldsymbol{B}_2\|_2 \\
&\leq \left(2\beta\gamma^{-1} + 2\alpha + K^2\right)\left\|\widehat{\mathrm{Cov}}(s, p_1) - \widehat{\mathrm{Cov}}(s, p_2)\right\|_{\mathrm{F}}.
\end{aligned}$$

Moreover, we construct a $\frac{\epsilon}{4(2\beta\gamma^{-1} + 2\alpha + K^2)}$ net on policy class $\mathbf{P}(s)$ based on Frobenius norm. From Lemma 41, the log size of this net is

$$(d+1)^2\log\left(\frac{96(d+1)^2\left(2\beta\gamma^{-1} + 2\alpha + L\right)}{\epsilon}\right).$$

Since clipping and adding more constraints will not increase the cover number, for any $h$, we have

$$\begin{aligned}
\log\mathcal{N}_\epsilon(\mathcal{V}_h) \leq \log\mathcal{N}_\epsilon(\widetilde{\mathcal{V}}) &\leq d\log\left(1 + \frac{16K^2}{\epsilon}\right) + d^2\log\left(1 + \frac{16\sqrt{d}\beta}{\epsilon\gamma}\right) + d^2\log\left(1 + \frac{16\sqrt{d}\alpha}{\epsilon}\right) \\
&\quad + (d+1)^2\log\left(\frac{96(d+1)^2\left(2\beta\gamma^{-1} + 2\alpha + K^2\right)}{\epsilon}\right).
\end{aligned}$$

$\square$

Lemma 43 shows the uniform concentration of all functions in $\mathcal{V}$. It also appears as Lemma D.4 of Jin et al. (2020b), Lemma D.7 of Sherman et al. (2023b) and Lemma 24 of Sherman et al. (2023a).

**Lemma 43.** *Let $\{x_\tau\}$ be a stochastic process on state space $\mathcal{S}$ with corresponding filtration $\{\mathcal{F}_\tau\}_{\tau=1}^\infty$. Let $\{\phi_\tau\}$ be an $\mathbb{R}^d$-valued stochastic process where $\phi_\tau \in \mathcal{F}_\tau$, and $\|\phi_\tau\| \leq 1$. Further, let $\Lambda_n = \lambda I + \sum_{\tau=1}^n \phi_\tau \phi_\tau^\top$. Then for any $\delta > 0$, with probability at least $1 - \delta$, for all $n \geq 1$ and any $V \in \mathcal{V}$ such that $\|V\|_\infty \leq D$, we have*

$$\left\| \sum_{\tau=1}^n \phi_\tau \left( V\left(x_\tau\right) - \mathbb{E}\left[V(x_\tau | \mathcal{F}_{\tau-1})\right]\right)\right\|_{\Lambda_n^{-1}}^2 \leq 4D^2 \left(\frac{d}{2}\log\left(\frac{n+\lambda}{\lambda}\right) + \log\frac{\mathcal{N}_\epsilon(\mathcal{V})}{\delta}\right) + \frac{8n^2\epsilon^2}{\lambda}$$

*where $\mathcal{N}_\epsilon(\mathcal{V})$ is $\|\cdot\|_\infty$ $\epsilon$- covering number of $\mathcal{V}$ with difference $\epsilon$.*

**Lemma 44** (Lemma D.4 in Sherman et al. (2023b))**.** *Let $\{\phi_i\}_{i=1}^n \in \mathbb{R}^d, \{y_i\}_{i=1}^n \in \mathbb{R}, \lambda \in \mathbb{R}$ and set $\Lambda = \sum_{i=1}^N \phi_i\phi_i^\top + \lambda I$, and $\widehat{w} = \Lambda^{-1}\sum_{i=1}^N \phi_i y_i$. Then for any $w^\star \in \mathbb{R}^d$*

$$\|\widehat{w} - w^\star\|_\Lambda \leq \left\| \sum_{i=1}^N \phi_i\left(y_i - \phi_i w^\star\right)\right\|_{\Lambda^{-1}} + \sqrt{\lambda}\|w^\star\|$$

### E.2 FTRL REGRET BOUNDS

**Lemma 45** (Standard FTRL bound)**.** *Let $\Omega \subset \mathbb{R}^d$ be a convex set, $g_1, \ldots, g_T \in \mathbb{R}^d$, and $\eta > 0$. Then the FTRL update*

$$w_t = \operatorname*{argmin}_{w\in\Omega}\left\{\left\langle w, \sum_{\tau=1}^{t-1} g_\tau\right\rangle + \frac{1}{\eta}\psi(w)\right\}$$

*ensures for any $u \in \Omega$ and $\eta_0 > 0$,*

$$\sum_{t=1}^T \langle w_t - u, g_t\rangle \leq \underbrace{\frac{\psi(u) - \min_{w\in\Omega}\psi(w)}{\eta}}_{\text{Penalty}} + \underbrace{\sum_{t=1}^T \left(\max_{w\in\Omega}\langle w_t - w, g_t\rangle - \frac{D_\psi(w, w_t)}{\eta}\right)}_{\text{Stability}}.$$

Since we do not use standard FRTL but run the same policy $\pi$ in $2\tau$ episodes. We will introduce a blocked FTRL regret bound in Lemma 46.

**Lemma 46.** *Let $K \in \mathbb{Z}_+, \tau \leq K, J = \lceil\frac{K}{\tau}\rceil$, and set $T_j = \{\tau(j-1)+1, \cdots, \tau j\}$ for all $j \in [J]$. Assume $\eta > 0$, let $g_k$ be a sequence of input, define*

$$g_{(j)} = \frac{1}{\tau}\sum_{k\in T_j} g_k, \forall j \in [J]$$

$$w_{(j+1)} = \operatorname*{argmin}_{w\in\Omega}\left\{\left\langle w, \sum_{\tau=1}^j g_{(\tau)}\right\rangle + \frac{1}{\eta}\psi(w)\right\}$$

*Then if $w_k \in \Omega$ are such that $w_k = w_{(j)}$ for all $k \in T_j, j \in [J]$, for any $u \in \Omega$ we have*

$$\sum_{k=1}^K \langle g_k, w_k - u\rangle \leq \frac{\tau(\psi(u) - \min_{w\in\Omega}\psi(w))}{\eta} + \sum_{k=1}^K \left(\max_{w\in\Omega}\langle w_k - w, g_k\rangle - \frac{D_\psi(w, w_k)}{\eta}\right)$$

*Proof.* By applying Lemma 45 on $g_{(j)}, x_{(j)}$, we get

$$\sum_{j=1}^J \langle g_{(j)}, w_{(j)} - u\rangle \leq \frac{\psi(u) - \min_{w\in\Omega}\psi(w)}{\eta} + \sum_{j=1}^J \left(\max_{w\in\Omega}\langle w_{(j)} - w, g_{(j)}\rangle - \frac{D_\psi(w, w_{(j)})}{\eta}\right)$$

In addition,

$$\sum_{j=1}^J \langle g_{(j)}, w_{(j)} - u\rangle = \sum_{j=1}^J \left\langle \frac{1}{\tau}\sum_{k\in T_j} g_k, w_k - u\right\rangle = \frac{1}{\tau}\sum_{j=1}^J\sum_{k\in T_j}\langle g_k, w_k - u\rangle = \frac{1}{\tau}\sum_{k=1}^K\langle g_k, w_k - u\rangle$$

On the other hand,

$$\sum_{j=1}^{J}\left(\max_{w\in\Omega}\langle w_{(j)}-w, g_{(j)}\rangle - \frac{D_\psi(w, w_{(j)})}{\eta}\right) \leq \sum_{j=1}^{J}\left(\max_{w\in\Omega}\left\langle w_{(j)}-w, \frac{1}{\tau}\sum_{k\in T_j} g_k\right\rangle - \frac{D_\psi(w, w_{(j)})}{\eta}\right)$$

$$\leq \sum_{j=1}^{J}\left(\max_{w\in\Omega}\frac{1}{\tau}\sum_{k\in T_j}\langle w_k - w, g_k\rangle - \frac{1}{\tau}\sum_{k\in T_j}\frac{D_\psi(w, w_k)}{\eta}\right)$$

$$\leq \frac{1}{\tau}\sum_{j=1}^{J}\sum_{k\in T_j}\left(\max_{w\in\Omega}\langle w_k - w, g_k\rangle - \frac{D_\psi(w, w_k)}{\eta}\right)$$

$$= \frac{1}{\tau}\sum_{k=1}^{K}\left(\max_{w\in\Omega}\langle w_k - w, g_k\rangle - \frac{D_\psi(w, w_k)}{\eta}\right)$$

Thus, we have

$$\sum_{k=1}^{K}\langle g_k, w_k - u\rangle \leq \frac{\tau(\psi(u) - \min_{w\in\Omega}\psi(w))}{\eta} + \sum_{k=1}^{K}\left(\max_{w\in\Omega}\langle w_k - w, g_k\rangle - \frac{D_\psi(w, w_k)}{\eta}\right)$$

$\square$

### E.3  OTHER TECHNICAL LEMMAS

**Lemma 47.** *Let $x_i$ be a sequence of vectors, $p_i$ a probability distribution and $a_i$ arbitrary scalars, then*

$$\left\|\sum_i p_i a_i x_i\right\|^2 \leq \left(\sum_i p_i \|x_i\|^2\right)\left(\sum_j p_j a_j^2\right).$$

*Proof.*

$$\left\|\sum_i p_i a_i x_i\right\|^2 = \left\|\sum_i p_i a_i^2 \frac{x_i}{a_i}\right\|^2 = \left\|\sum_i \frac{p_i a_i^2}{\sum_j p_j a_j^2}\frac{x_i}{a_i}\right\|^2\left(\sum_j p_j a_j^2\right)^2$$

$$\leq \sum_i \frac{p_i a_i^2}{\sum_j p_j a_j^2}\left\|\frac{x_i}{a_i}\right\|^2\left(\sum_j p_j a_j^2\right)^2 \qquad\text{(Jensen's)}$$

$$= \left(\sum_i p_i \|x_i\|^2\right)\left(\sum_j p_j a_j^2\right).$$

$\square$

