# OpenReview forum: "Towards Optimal Regret in Adversarial Linear MDPs with Bandit Feedback"
_ICLR.cc/2024/Conference — ICLR 2024 spotlight_

### Official Review · Reviewer_eGft · 2023-10-27

**Soundness:** 4 excellent
**Presentation:** 3 good
**Contribution:** 4 excellent
**Rating:** 8
**Confidence:** 3

**Summary:**

The paper studies the problem of learning in online adversarial linear Markov decision processes in the presence of partial feedback. The authors propose two algorithms. The first one achieves optimal $\sqrt{K}$ regret bound while being computationally inefficient. The second one achieves a regret bound of order $K^\frac{3}{4}$, while being computationally efficient.

**Strengths:**

The paper is clear and well-written. The setting the paper studies is of interest both from a theoretical and a practical perspective, and it has gained lots of attention in the last few years. The theoretical results, both for the efficient setting and the inefficient one, are surely improving the state-of-the-art. Indeed, the paper answers different questions raised by prior work (e.g., whether the $\sqrt{K}$ regret bound was achievable). From an algorithmic perspective, even if the authors employ many existing techniques such as dilated bonus and FTRL with logdet barrier, the novelty is clear. The theoretical analysis is interesting and non-trivial, even if part of it is partially adapted from existing work.

**Weaknesses:**

- The MDP has a layered structure. Indeed, this is standard in the literature of online learning in Markov decision process and it is without loss of generality. Nevertheless, the dependency on the decision space could be worse than the one presented in the paper for not loop-free MDPs.
- The dependency of the regret bound on the horizon and the feature vector dimension is far from being good.

**Questions:**

I am interested in understanding why the adversary is assumed to be oblivious. In online episodic adversarial MDP research area, the adversary can be adaptive.

---

> ### Author Response · Authors · 2023-11-18
>
> Thank you for the support and the valuable feedback. Please see our response below.
>
> **Q1:** Why the adversary is assumed to be oblivious?
>
> **Reply:** This is mostly for simplicity. Our analysis can directly obtain a *pseudo regret* bound for adaptive adversary, where pseudo regret is defined as
> $$\mathbb{E}\left[\sum_{k=1}^K V^{\pi_k}\left(s_1, \ell_k\right)\right] - \min_{\pi} \mathbb{E}\left[\sum_{k=1}^K V^{\pi}(s_1; \ell_k)\right]. $$
> If we would like to obtain the stronger *expected regret*:
> $$\mathbb{E}\left[\sum_{k=1}^K V^{\pi_k}\left(s_1, \ell_k\right)\right] - \mathbb{E}\left[\min_{\pi} \sum_{k=1}^K V^{\pi}(s_1; \ell_k)\right]$$
> for adaptive adversary, then the standard technique is to first obtain a *high probability* bound for the algorithm, and then take expectations [1]. Overall, to obtain an expected regret bound for adaptive adversary, it suffices to prove a high-probability bound for the algorithm.
>
> The standard technique to obtain a high-probability bound is to add a bonus in the update to compensate the deviation in the concentration bound. This is based on the same idea of our current algorithm (which already incorporates some bonus term to compensate the bias), but may require extra bonus terms. We do not pursue in this direction to focus the exposition more on our key contributions. We point out that the regret notion in our work is the same as those in previous work (Luo et al. 2021, Dai et al. 2023, Sherman et al. 2023b, Kong et al. 2023), and the cases the algorithms can be applied to are also the same.
>
> [1] Julian Zimmert and Tor Lattimore. Return of the bias: Almost minimax optimal high probability bounds for adversarial linear bandits. In Conference on Learning Theory, pages 3285–3312. PMLR, 2022

---

### Official Review · Reviewer_B4ir · 2023-10-29

**Soundness:** 3 good
**Presentation:** 3 good
**Contribution:** 4 excellent
**Rating:** 8
**Confidence:** 3

**Summary:**

This work establishes the first rate-optimal algorithm for adversarial linear MDPs with bandit feedback, though it is computationally inefficient. Besides, this work also provides a computationally efficient policy optimization (PO)-based algorithm with $\widetilde{O}(K^{3/4})$ regret, improving previous SOTA result of order $\widetilde{O}(K^{6/7})$.

**Strengths:**

1. Results: The problem of rate optimal results for learning adversarial linear MDPs with bandit feedback has been open since [1]. It is exciting to see an algorithm, though not computationally efficient, obtaining the rate optimal result for this challenging problem.
2. Novelty: The rate optimal algorithm takes the same viewpoint that reducing the adversarial linear MDP problem as an adversarial linear bandit problem as [2], but new algorithmic designs are proposed to achieve the rate optimal result, which might be of independent interest. I think the combination of existing techniques to devise a computationally efficient PO-based algorithm with $\widetilde{O}(K^{3/4})$ regret also has its merits.
3. Writing: In general, this paper is well-written, with sufficient discussions on the algorithm designs.

[1] Luo et al. Policy optimization in adversarial mdps: Improved exploration via dilated bonuses. NeurIPS, 2021.

[2] Kong et al. Improved regret bounds for linear adversarial mdps via linear optimization. arXiv, 2023.

**Weaknesses:**

1. In this work, Algorithm 1 used to estimate $ \hat{\mu}^\pi $ for policy $\pi $ needs to solve a complicated optimization problem, which might be less appealing for practitioners in the RL community. However, I think this is not a fatal weakness, given both the hardness of this problem and the technical novelty in the design of the first algorithm.
2. Giving a table that shows the comparisons of regret bounds with most related works might further benefit the readers, in my opinion.

**Questions:**

1. Can the authors give a brief discussion about the possibility or the main barriers to achieving the rate-optimal result for PO-based methods?

---

> ### Author Response · Authors · 2023-11-18
>
> Thank you for the support and the valuable feedback. We have added a table to compare the regret bounds. Please see Appendix A. Other questions are addressed below.
>
> **Q1:** Can the authors give a brief discussion about the possibility or the main barriers to achieving the rate-optimal result for PO-based methods?
>
> **Reply:** We believe that it is possible to achieve optimal rate with PO-based methods. Currently, the main challenge lies in how to efficiently reuse previous data to create an covariance matrix estimator $\hat{\Sigma}_k$, and bound the bias of the loss estimator due to the error in $\hat{\Sigma}_k$. Below we give a high-level explanation on why our inefficient exponential weight algorithm is able to achieve an optimal rate, while the PO approach still faces difficulties.
>
> Notice that the main difference between the inefficient exponential weight algorithm and the PO algorithm is that, the former runs a "global" linear bandit algorithm over all policies, while the latter runs a "local" linear bandit algorithm on every state. This difference leads to a difference in their loss estimator construction. The loss estimator in the exponential weight case is
>
> $$M_k^{-1}\hat{\phi} _k^{\pi_k}L _k$$
>
> where
>
> $$M_k = \mathbb{E}_{\pi\sim q_k}[\hat{\phi}_k^\pi\hat{\phi}_k^{\pi\top}],$$
>
> while the loss estimator for the PO case is
>
> $\hat{\Sigma}\_{k,h}^{-1} \phi (s\_{k,h}, a\_{k,h}) L\_{k,h}$
>
>
> where
>
> $$\hat{\Sigma}\_{k,h} = \lambda I + \hat{\mathbb{E}}\_{s\_h\sim \pi\_k}\left[\sum\_a \pi\_k(a|s\_h)\phi(s\_h,a)\phi(s\_h,a)^\top \right]$$
>
> and $\hat{\mathbb{E}}\_{s\_h\sim \pi\_k}$ means that we have to use empirical data to estimate the state distribution of $s_h$ under $\pi_k$. For exponential weights, the bias comes from the error of $\hat{\phi}^\pi_k$, while its covariance matrix $M_k$ can be computed exactly (since $q_k$ is known to the learner). For PO, on the other hand, the bias mainly comes from the error in the covariance matrix $\hat{\Sigma}\_{k,h}$. In exponential weights, bounding the error $\hat{\phi}^\pi\_k - \phi^\pi\_k$ is relative easier, while in PO, bounding the error $\hat{\Sigma}\_{k,h}^{-1} - \Sigma\_{k,h}^{-1}$ is more challenging because concentration inequalities for inverse matrices are more difficult to establish. Currently, we use fresh on-policy samples to estimate $\hat{\Sigma}\_{k,h}$ in epoch $k$, which is expensive. A more sample efficient option would be to reuse previously collected data to estimate $\hat{\Sigma}\_{k,h}$ in an off-policy manner. However, currently we do not know how to bound the estimation error $\hat{\Sigma}\_{k,h}^{-1} - \Sigma\_{k,h}^{-1}$ tightly if adopt this approach.

---

> > ### Comment · Reviewer_B4ir · 2023-11-22
> > **Reply**
> >
> > Thanks for the detailed response. I have no further questions.

---

### Official Review · Reviewer_QJAu · 2023-11-01

**Soundness:** 3 good
**Presentation:** 4 excellent
**Contribution:** 2 fair
**Rating:** 8
**Confidence:** 3

**Summary:**

The paper studies the linear Markov decision processes(MDP) with adversarial losses and bandit feedback. This paper first 1) introduces a computationally inefficient algorithm that achieves a regret of $\tilde{O}(\sqrt{K})$, for $K$ as the number of episode in this MDP, and then 2) introduces a second algorithm that is computationally efficient and achieves $\tilde{O}(K^{3/4})$ regret. The first algorithm is nearly optimal, and the second algorithm significantly improves the state-of-the-art.

**Strengths:**

- The presentation of this paper is good, and display a relatively clean and timely literature review.
- This paper clearly presents the intuition behind the results presented in this paper, as well as the similarities and differences between the algorithm and the state-of-the-art (SOTA), and where SOTA offers further improvement on regret, e.g. paragraphs in section 3.1
- The result of this paper is a significant improvement of state-of-the-art.

**Weaknesses:**

- The pseudo-code of algorithm 1 lacks explanations.
- This paper is purely theoretical, and hence doesn't have empirical evaluations.

**Questions:**

- What is the practical motivation of this problem?
- With the help of simulators, K^{2/3} regret can be obtained, what is changed when the simulator of the environment is available?

---

> ### Author Response · Authors · 2023-11-18
>
> Thank you for the support and the valuable feedback. Please see our response below.
>
> **Q1:** The pseudo-code of algorithm 1 lacks explanations.
>
> **Reply:** The explanation of Algorithm 1 is provided in Page 5 between Eq.(6) and Lemma 5. Please feel free to let us know if the exposition there requires further improvement.
>
> **Q2:** What is the practical motivation of this problem?
>
> **Reply:** Our work follows the line of research by (Luo et al. 2021, Dai et al. 2023, Sherman et al. 2023b, Kong et al. 2023) that investigate the intersection of function approximation, non-stationary reward, and the design of exploration bonus on top of them. The main goal of this line of research is to improve the weak part of traditional policy-based algorithms --- having low sample efficiency, only performing local policy search --- through the use of exploration bonus, while keeping its strength of being robust to non-stationarity. Indeed, simultaneously ensuring sample efficiency and robustness in changing environments is crucial for fields such as robotics. We agree that currently this line of research is theory-oriented, but some techniques that have been developed (e.g., the use of the initial exploration phase to reduce the required exploration bonus, and the recursive construction of the exploration bonus) could provide insights for future empirical study.
>
> **Q3:** With the help of simulators, $K^{2/3}$ regret can be obtained, what is changed when the simulator of the environment is available?
>
> **Reply:** We point out that the state-of-the-art regret bound with a simulator is $\sqrt{K}$ by Algorithm 1 in [1].
>
> With the help of a simulator, the learner can learn the transition directly from the simulator without interacting with the environments (assuming the simulator is unbiased). This saves the number of interactions with the environments. Specifically, for our Algorithm 2, if there is a simulator, the estimation of the covariance matrix can be free and with low error. This also eliminates the need for the initial exploration phase because the original goal of the initial phase is to reduce the bias caused by the covariance matrix estimation error. Overall, if there is a simulator, our policy optimization algorithm can actually achieve a $\sqrt{K}$ regret, like in [1].
>
> [1] Yan Dai, Haipeng Luo, Chen-Yu Wei, and Julian Zimmert. Refined regret for adversarial mdps with linear function approximation. In International Conference on Machine Learning, 2023.

---

> > ### Comment · Reviewer_QJAu · 2023-11-20
> >
> > Thanks for the feedback, I raised my score accordingly.

---

### Official Review · Reviewer_vRkG · 2023-11-04

**Soundness:** 3 good
**Presentation:** 4 excellent
**Contribution:** 3 good
**Rating:** 6
**Confidence:** 4

**Summary:**

This work studies RL on adversarial MDPs with bandit feedback. In detail, the authors proposed the first algorithms for linear MDPs with the standard $\sqrt{K}$ regret. Although with a $\sqrt{K}$ regret, such algorithms are not computationally efficient since the computation complexity depends on the state complexity $|\cS|$. The authors also proposed computationally efficient algorithms with a $K^{3/4}$ regret, which is also the SOTA result.

**Strengths:**

The improvement of the existing regret results for linear MDP with adversarial loss is very important for the research community. The presentation of this paper is also very clear.

**Weaknesses:**

I do not find any obvious weaknesses in this work.

**Questions:**

1. Can the authors discuss the possibility of designing an algorithm with a minimax-optimal regret guarantee? Such a result has already been established in [1,2] for linear mixture MDPs with/without adversarial losses, with the full information feedback. [2] also established a lower bound of regret which depends on $\log|\cS|$ and $\log|\cA|$. Does the same lower bound also hold for the bandit feedback setting?

[1] Zhou, Dongruo, Quanquan Gu, and Csaba Szepesvari. "Nearly minimax optimal reinforcement learning for linear mixture markov decision processes." Conference on Learning Theory. PMLR, 2021.
[2] Ji, Kaixuan, et al. "Horizon-free Reinforcement Learning in Adversarial Linear Mixture MDPs." arXiv preprint arXiv:2305.08359 (2023).

---

> ### Author Response · Authors · 2023-11-18
>
> Thank you for the support and the valuable feedback. Please see our response below.
>
> **Q1:** Can the authors discuss the possibility of designing an algorithm with a minimax-optimal regret guarantee? Such a result has already been established in [1,2] for linear mixture MDPs with/without adversarial losses, with the full information feedback. [2] also established a lower bound of regret which depends on $\log{|\mathcal{S}|}$ and $\log{|\mathcal{A}|}$. Does the same lower bound also hold for the bandit feedback setting?
>
>
>
> **Reply:** The main differences between our setting and those of [1, 2] are the following:
>
> (1)  They study full-information loss feedback, while we study bandit loss feedback;
>
> (2) Their *linear mixture MDP* only assumes linear structure on the transition ($\mathbb{P}(s'|s,a) = \phi(s'|s,a)^\top \psi$), while our *linear MDP* assumes linear structures both on the transition ($\mathbb{P}(s'|s,a) = \phi(s,a)^\top \psi(s')$) and the loss ($\ell_k(s,a) = \phi(s,a)^\top \theta_k$). Notice that the assumptions on the transition are different -- in linear mixture MDP, it's possible to use a model-based algorithm to estimate the the transition, while in linear MDP this is not allowed.
>
> The algorithmic techniques required for linear mixture MDPs and linear MDPs are quite different due to the different assumptions. Therefore, the bounds obtained in one setting cannot be translated or compared to the other setting in general. We note that a concurrent work [2] studied a more comparable setting of adversarial linear MDPs with full-information loss feedback. They obtained a regret of $\sqrt{d^4H^6 K}$ (also the first rate-optimal result), while ours in the bandit setting is $\sqrt{d^7H^7 K}$. Clearly, in both works, there is still a significant gap between the upper bounds and the currently best lower bound $dH\sqrt{K}$. We believe that the lower bound is tight. Potential ways to improve our upper bound include:
>
> - In Algorithm 1, try NOT to control the estimation error for all functions in the function set $\mathcal{F}_\pi$ (which requires an union bound over an exponentially large function set); instead, just control the error for those $f$ that are relevant to the estimation of $\phi^\pi$.
>
> - Improve the upper bound for the bias term in Lemma 10 -- we believe that a more refined and complicated analysis could improve the $d$ dependence there.
>
> To implement these ideas, significant changes in the analysis and the algorithm may be required, so we leave it as future work.
>
> Next, we remark on the $\log|\mathcal{S}|$ and $\log|\mathcal{A}|$ lower bound established in [2]. The key reason that they have explicit $|\mathcal{S}|$ and $|\mathcal{A}|$ dependencies is that in their linear mixture MDP model, there is NO structural assumption on the loss function $\ell_k(s,a)$ (see (2) above). Therefore, their $\ell_k(s,a), \ell_k(s',a')$ can be arbitrarily different even if $\phi(s,a)=\phi(s',a')$. This makes the intrinsic dimension of their $\ell_k(s,a)$ scales with $|\mathcal{S}||\mathcal{A}|$ (though the full-information assumption make their regret dependency only logarithmic). On the other hand, in our linear MDP model, $\ell_k(s,a)$ can be represented as a  $d$ dimensional linear function. Therefore, our bound can only depend on $d$ but not on $|\mathcal{S}|$ or $|\mathcal{A}|$.
>
> [1] Zhou, Dongruo, Quanquan Gu, and Csaba Szepesvari. Nearly minimax optimal reinforcement learning for linear mixture markov decision processes. Conference on Learning Theory. PMLR, 2021.
>
> [2] Ji, Kaixuan, et al. Horizon-free Reinforcement Learning in Adversarial Linear Mixture MDPs. arXiv preprint arXiv:2305.08359 (2023).
>
> [3] Uri Sherman, Alon Cohen, Tomer Koren, Yishay Mansour. Rate-Optimal Policy Optimization for Linear Markov Decision Processes. arXiv preprint arXiv:2308.14642 (2023).

---

### Author Response · Authors · 2023-11-18

We thank all the reviewers for their valuable feedback. Below are the changes in the revised version:

1)  After the paper submission and before the rebuttal phase, we have improved the writing of the paper, fixed some small error in the analysis, and performed better parameter choices. They led to some minor improvement in the bounds (still of order $\sqrt{K}$ and $K^{3/4}$ though). We highlight the new regret bounds in red.

2)  We move OBME (current Algorithm 4) from the appendix to the main text, while moving the related work section from the main text to the appendix. We also added a table to compare regret bounds with previous works, as suggested by Reviewer B4ir.

---

### Meta-Review · Area_Chair_7Cmz · 2023-12-04

**Metareview:**

This paper advances the theoretical SOTA on no-regret learning in adversarial linear MDPs with bandit feedback. They present a computationally inefficient alg that achieves $\tilde O(\sqrt{K})$ regret and a computationally efficient alg that achieves $\tilde O(K^{3/4})$ regret, both of which significantly outperforms the state of the art.

**Justification For Why Not Higher Score:**

Relatively niche.

**Justification For Why Not Lower Score:**

Strong results.

---

### Decision · Program_Chairs · 2024-01-16

Accept (spotlight)